# Learning Universal Adversarial Perturbations for Ordered Top-K Targeted Attacks

## Abstract

Universal adversarial perturbations (UAPs) have deepen concerns regarding the vulnerability of Deep Neural Networks (DNNs) under the white-box attack setting. While most success with UAPs has been observed in untargeted attack settings, achieving effective top-1 targeted UAPs has proven challenging. In this paper, we address this challenge by demonstrating that ordered Top-K targeted UAPs can be learned aggressively along the label target axis (tested up to Top-6), and transfer very well along the data axis (i.e., across images from the seen training images to the unseen test images). They also show strong double-transferability across unseen test models and unseen test images, when learned from an ensemble of disparate train models. Our method, named **AllAttacK**, simultaneously targets three axes: images, models, and label targets, and is posed as a maximum satisfiability (MAXSAT) problem. We evaluate AllAttacK on the ImageNet-1k classification task using 27 diverse models with more than 500 UAPs learned, showing that the resulting perturbations not only exhibit strong transferability but also display intriguing, interpretable characteristics.

## 1 Introduction

Visual perception is robust with human vision, and is aimed to be similarly, if not more, robust with computer vision (Palmer, 1999). Computer vision has witnessed remarkable progress by end-to-end representation learning using Deep Neural Networks (DNNs) (LeCun et al., 1998; Krizhevsky et al., 2012; He et al., 2016; Huang et al., 2017; Dosovitskiy et al., 2020). However, adversarial attacks can easily fool well trained DNNs (e.g., classify a dog image as a cat) by adding visually-imperceptible perturbations (Nguyen et al., 2015; Szegedy et al., 2014; Athalye & Sutskever, 2017; Carlini & Wagner, 2016; Goodfellow et al., 2015; Kannan et al., 2018; Madry et al., 2017; Xie et al., 2019; Madry et al., 2018). Initially perceived as mere anomalies, adversarial attacks have rapidly evolved, posing increasingly intricate challenges (Geirhos et al., 2020) for the reliability and trustworthiness of DNNs, especially in high-stake applications.

Universal Adversarial Perturbations (UAPs) have heightened concerns regarding the vulnerability of DNNs since they are doubly transferable across DNNs and images. UAPs are often *quasi-imperceptible* with higher perturbation energy than perturbations optimized for a single image using a single model. While most success with UAPs has been observed in untargeted attack settings (Moosavi-Dezfooli et al., 2017; Shafahi et al., 2020), **achieving effective top-1 targeted UAPs has proven challenging** (Liu et al., 2016; Zhang et al., 2020b; Weng et al., 2023). Furthermore, the prior art of learning UAPs is mainly tested with convolutional neural networks, including CaffeNet (Jia et al., 2014), VGGNets (Chatfield et al., 2014; Simonyan & Zisserman, 2015), GoogLeNet (Szegedy et al., 2015) and ResNets (He et al., 2016). With the recent development of new neural architectures such as Vision Transformers (Dosovitskiy et al., 2020), ConvNeXt (Woo et al., 2023) and MLP-Mixers (Tolstikhin et al., 2021), and with new and more powerful training recipes such as the contrastive language-image pretraining (CLIP) (Radford et al., 2021) and further combined with masked image modeling (MIM) as in the EVA2 model (Fang et al., 2023), **it is unclear whether targeted UAPs can retain their attacking power for ensembles of those disparate DNNs, as well as adversarially-robustified counterparts** (Croce et al., 2020).

More recently, the guidelines of the National Institute of Standards and Technology (NIST SP800-226, March 2025) (Near et al., 2025) frames robustness/utility checks around whether the *entire ordered set* of the highest-scoring items is preserved under noise—not just the single best (Durfee & Rogers,

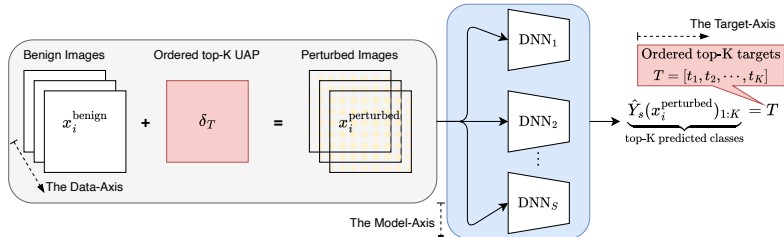

Figure 1: Illustration of the proposed AllAttacK for learning ordered top-K UAPs, $\delta_T$, along three axes: the data-axis, the model-axis and the target-axis. See text for details.

2019)—underscoring regulators' need for *top-K mis-ranking tests*. For example, safety-critical systems (face unlock, medical triage, content moderation) reason over *entire ranked lists*. An attacker dictating *all* top predictions obtains finer control and evades simple "top-1 changed" detectors. In practice, well-trained DNNs often output semantically similar classes in the top predictions for benign images. top-1 targeted attacks often break the semantic coherence between the top-K predictions of adversarial examples, making them relatively easier to detect by checking the top-K semantic similarities. Top-K targeted attacks can select semantically coherent adversarial targets in learning attacks, addressing the issue with top-1 targeted attacks. **Learning ordered top-K targeted UAPs has not been studied in the literature, however.**

**Our aim** is to revisit and expand targeted UAPs to enable the target labels transferrable. More specifically, we aim to learn targeted UAPs simultaneously along three axes:

*The target-axis:* Going beyond top-1 targets, how many top-K targets can be attacked, and can they be attacked with respect to any given orders as stress-tests to reveal deeper vulnerabilities? We integrate single-model-single-image ordered top-K targeted adversarial attacks studied in (Zhang & Wu, 2020; Paniagua et al., 2023), which provide specific targeted top-K classes in order and the top-K predicted classes after attack must match this exact order. (Paniagua et al., 2023) provides convincing justifications for why ordered top-K attacks matter. As stress-tests, we will randomly sample ordered top-K targets, not just the semantically coherent ones (that are often easier to attack as shown in our experiments).

*The model-axis:* How many different types of DNNs (e.g., convolutional neural networks, Transformer models and all-MLP models), and how many different models of each type can be attacked, simultaneously? Furthermore, can perturbations learned from an ensemble of training models generalize to unseen models that are of very different architectures than those in training? We test an ensemble of disparate models such as different types of Convolutional Neural Networks and their adversarially-robustified versions, Vision Transformers, CLIP vision encoders, and MLP-Mixers.

*The data-axis:* How many of training images (that are used in learning UAPs) can be attacked, and how many unseen images can the same UAPs transfer to? We learn ordered top-K targeted UAPs using 1000 `train` images in ImageNet-1k (Russakovsky et al., 2015), and also test them using 1000 `val` images that are unseen both by the DNNs themselves and in learning the UAPs.

We present **AllAttacK** that shows the ordered top-K target labels can transfer (for $K \geq 1$), as illustrated in Fig. 1. Fig. 2 shows some examples of learned UAPs. The existence of such targeted UAPs could indicate that there were aspects that have been overlooked in a systematic way in DNNs. Seeking quantitative analyses of AllAttacK will facilitate us understanding the adversarial vulnerability at the fundamental level better (e.g., how aggressive can adversarial attacks be?). Studying models' vulnerability under such attacks reveals an upper bound on its susceptibility, and often provides insights that inform better defenses. Many black-box attacks succeed by approximating white-box gradients, either through query-based methods or surrogate models. By identifying failure points through white-box attacks, we gain critical insights into potential black-box vulnerabilities.

**Our Contributions.** This paper makes two main contributions to the field of learning white-box targeted adversarial attacks: (i) It presents, to our knowledge, the first large-scale study of learning ordered Top-K targeted UAPs that are both model-agnostic (up to 18 disparate DNNs in training) and image-agnostic (at the ImageNet-1k scale), with strong results obtained. (ii) It proposes two optimization formulations in learning AllAttacK, built on previous single-model-single-image ordered Top-K attack work, with a proposed stochastic mini-data-batch and mini-model-batch optimization strategy for practicality and generalizability.

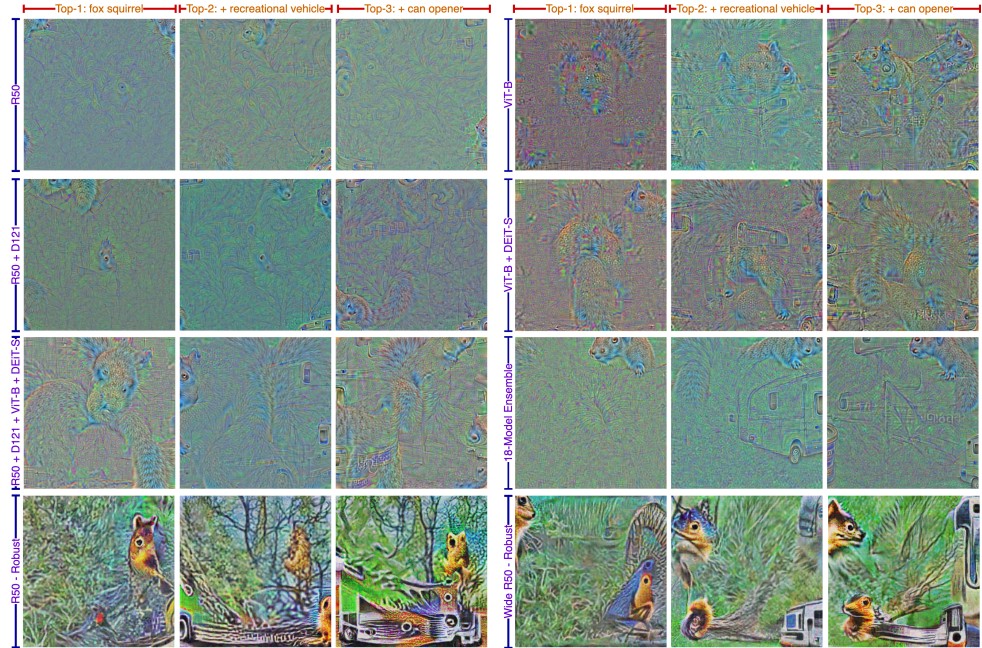

Figure 2: Examples of AllAttacK UAPs ($K = 1, 2, 3$) using Eqn. 7 and 1000 training images in ImageNet-1k (Russakovsky et al., 2015). The models include: ResNets-50 (R50) (He et al., 2016) and two adversarially-trained variants (Engstrom et al., 2019; Salman et al., 2020), DenseNet-121 (D121) (Huang et al., 2017), ViT-Base (Dosovitskiy et al., 2020), DEiT-Small (Touvron et al., 2022), and an 18-model ensemble. See text for details. **Please check all UAPs via a HTML visualization interface (see Fig. 4 in the Appendix) in the supplementary.**

## 2  PROBLEM FORMULATION OF ALLATTACK

We consider image classification with the label set $\mathcal{Y}$ (e.g., 1000 classes in ImageNet (Russakovsky et al., 2015)). Let $F(\cdot)$ be a DNN (e.g., ResNet-50 (He et al., 2016)) trained on the ImageNet, which consists of the feature backbone $f_\theta(\cdot) \in \mathbb{R}^d$ (where $\theta$ collects all backbone parameters), and a linear head classifier $h(\cdot; W, b) \in \mathbb{R}^{|\mathcal{Y}|}$ computing the logits (where $W \in \mathbb{R}^{d \times |\mathcal{Y}|}$ and $b \in \mathbb{R}^{|\mathcal{Y}|}$). For an RGB image $x \in [0, 1]^{3 \times H \times W}$ where $H$ and $W$ are image height and width respectively, we have,

$$\text{Logits: } F(x) = h\big(f(x; \theta); W, b\big) = f(x; \theta) \cdot W + b, \qquad (1)$$

$$\text{Probabilities: } \hat{P}(x) = \text{Softmax}\big(F(x)\big), \qquad (2)$$

$$\text{Sorted Label Indexes: } \hat{Y}(x) = \arg \text{sort}\big(\hat{P}(x)\big), \qquad (3)$$

where $\hat{Y}(x)$ are the predicted class indexes in the descending order of predicted probabilities. The top-K predicted classes are denoted by $\hat{Y}(x)_{1:K}$. As illustrated in Fig. 1, our proposed AllAttacK aims to address the challenges along the three axes as follows,

- **The Model Axis.** Denoted by $\Theta = (\theta, W, b)$ a trained DNN, and we have a training set of DNNs, $\Theta_s \in \mathcal{M}^{\text{train}}$ used in learning UAPs, and a testing set of DNNs, $\Theta_u \in \mathcal{M}^{\text{test}}$ in evaluating the model-transferability of learned UAPs. We select a set of disparate DNNs in experiments, in which some testing DNNs have very different neural architectures than those training DNNs.
- **The Data Axis.** Denote by $\mathcal{D}^{\text{train}}$ and $\mathcal{D}^{\text{test}}$ the training and testing sets for AllAttacK. $\mathcal{D}^{\text{train}}$ is sampled from the ImageNet-1k train set each sample of which can be correctly classified by all the DNNs in $\mathcal{M}^{\text{train}} \cup \mathcal{M}^{\text{test}}$. $\mathcal{D}^{\text{test}}$ is sampled from the ImageNet-1k validation set which is not only unseen in training AllAttacK, but also unseen by all the DNNs in their training stages (if they are trained from scratch on ImageNet-1k). We sample one image per category for both $\mathcal{D}^{\text{train}}$ and $\mathcal{D}^{\text{test}}$, resulting 1000 images, 1 for each class in ImageNet-1k.
- **The Target Axis.** Denote by $T = [t_1, \cdots, t_K]$ a list of ordered top-K adversarial targets sampled from $\mathcal{Y}$, where $t_k \in \mathcal{Y}$. Since we sample one image per category, we remove the image whose ground-truth label is the same as the target $t_1 \in T$. For the remaining images, we also verify that the sampled $T$ is not a segment of any of the model predictions, i.e., $T \not\subset \hat{Y}_\Theta(x_i)$, for $K \geq 2$, $\forall \Theta \in \mathcal{M}^{\text{train}} \cup \mathcal{M}^{\text{test}}$ and $\forall x_i \in \mathcal{D}^{\text{train}} \cup \mathcal{D}^{\text{test}}$.

**Denote by** $\delta_T$ **the AllAttacK UAP** learned for the target-model-data triplet, $(T, \mathcal{M}^{\text{train}}, \mathcal{D}^{\text{train}})$. The objective is to solve the maximum satisfiability (MAXSAT) problem,

$$\underset{\delta_T \in \mathbb{R}^{3 \times H \times W}}{\text{minimize}} \quad \|\delta_T\|_p^2, \tag{4}$$

$$\text{subject to} \quad F_{\Theta_s}(x_i^{\text{perturb}})_{t_k} > F_{\Theta_s}(x_i^{\text{perturb}})_{t_{k+1}}, \quad \forall k \in [1, K-1], \quad t_k \in T$$

$$F_{\Theta_s}(x_i^{\text{perturb}})_{t_K} > F_{\Theta_s}(x_i^{\text{perturb}})_c, \quad \forall c \in \mathcal{Y} \setminus T, \quad t_K \in T, \quad \forall \Theta_s \in \mathcal{M}^{\text{train}}$$

$$x_i^{\text{perturb}} = \text{Clamp}(x_i^{\text{benign}} + \delta_T), \quad \forall x_i^{\text{benign}} \in \mathcal{D}^{\text{train}},$$

where $\| \cdot \|_p$ is the $\ell_p$ norm (e.g., $\ell_2$), and $\text{Clamp}(\cdot)$ is to clip the input element-wisely to $[0, 1]$. The number of (non-linear) constraints is very large, e.g., 1000 images $\times 4$ models $\times |\mathcal{Y}| - 1)$ logit-orders, which enables the satisfaction, $T = \hat{Y}_{\Theta_s}(x_i^{\text{perturb}})_{1:K}$.

**The Challenge of AllAttacK.** The goal of AllAttacK is to seek an energy-minimized point $\delta_T$ in the data space, which once added to an input benign image can "shut it off" and "steer it towards the adversarial targets $T$" for any training or testing DNNs, no matter what the benign top-K predictions are. Put in other words, $\delta_T$ **will induce benign images as "negligible noise" to DNNs.** The existence of such UAPs (see examples in Fig. 2) clearly shows that those DNNs might still have "shallow and fragile understanding" of the structure of the data space. As we observed in experiments, the learned universal top-K perturbations alone indeed can fool most of DNNs.

## 3 Learning AllAttacK UAPs

Our AllAttacK is built on two single-model-single-image ordered top-K attack methods, the KL divergence based adversarial distillation method (Zhang & Wu, 2020) and the more recently proposed quadratic programming based method, QuadAttacK (Paniagua et al., 2023). We propose a joint mini-data-batch and mini-model-batch optimization strategy in learning UAPs for a large number of models (e.g., up to 18 disparate DNNs) and a large number of images (e.g., 1000 images).

### 3.1 AllKLAttacK: Minimizing a Surrogate Loss Function

To reformulate Eqn. 4 as an unconstrained optimization problem, we seek some surrogate loss functions, $\mathcal{L}(x_i^{\text{perturb}}, T; \Theta_s)$, such that the ordered top-K attack constraint $T = \hat{Y}_{\Theta_s}(x_i^{\text{perturb}})_{1:K}$ is satisfied if and only if $\mathcal{L}(x_i^{\text{perturb}}, T; \Theta_s) \leq 0$. We have,

$$\underset{\delta_T}{\text{minimize}} \quad \|\delta_T\|_p^2 + \lambda \cdot \frac{1}{S \cdot I} \sum_{i=1}^{I} \sum_{s=1}^{S} \mathcal{L}(x_i^{\text{perturb}}, T; \Theta_s), \tag{5}$$

where $x_i^{\text{perturb}} = \text{Clamp}(x_i^{\text{benign}} + \delta_T)$, $I = |\mathcal{D}^{\text{train}}|$, and $S = |\mathcal{M}^{\text{train}}|$. $\lambda$ is a trade-off parameter in optimization between the energy of learned perturbation and the attack success rate. In this paper, we build on the adversarial distillation (AD) loss function proposed in (Zhang & Wu, 2020), which has shown state-of-the-art performance in learning model-/instance-specific top-K targeted attacks under the unconstrained optimization formulation. The AD loss function is based on the Kullback-Leiber (KL) divergence between the predicted probability distribution $\hat{P}_{\Theta_s}(x_i^{\text{perturb}})$ and a top-down designed target distribution $P^{AD}(T)$,

$$\mathcal{L}(x_i^{\text{perturb}}, T; \Theta_s) = \text{KL}\big(P^{AD}(T)\|\hat{P}_{\Theta_s}(x_i^{\text{perturb}})\big), \tag{6}$$

where $P^{AD}(T)$ maintains the ordered top-K targets $T$, $P^{AD}(T)_{t_u} > P^{AD}(T)_{t_v}, \forall u < v$, and exploits the label distance between labels using the Glove embedding (Pennington et al., 2014). Please refer to (Zhang & Wu, 2020) for details. In Eqn. 6, $\text{KL}\big(P^{AD}(T)\|\hat{P}_{\Theta_s}(x_i^{\text{perturb}})\big) \geq 0$, and it equals zero if and only if the two distributions exactly match, $\hat{P}_{\Theta_s}(x_i^{\text{perturb}}) \equiv P^{AD}(T)$.

### 3.2 AllQuadAttacK: The Quadratic Programming Formulation

In the surrogate KL-divergence loss function (Eqn. 6), the design of $P^{AD}(T)$ has more than needed information (i.e., the probability differences between different categories) in addition to maintaining the top-K order of targets. As pointed by a recently proposed QuadAttacK (Paniagua et al., 2023) method, eliminating those unnecessary constraints and directly maintaining the order of the top-K targets as linear constraints facilitate a Quadratic Programming (QP) solution with significantly better performance in learning single-model-single-image ordered top-K targeted attacks.

Built on QuadAttacK (Paniagua et al., 2023), given the current perturbation $\delta_T$, our AllAttacK is learned by first solving the QP problem,

$$\underset{z_{i,s} \in \mathbb{R}^{d_s}}{\text{minimize}} \frac{1}{S \cdot I} \sum_{i=1}^{I} \sum_{s=1}^{S} \beta_s \cdot ||z_{i,s} - f(x_i^{\text{perturb}}; \theta_s)||_2^2, \quad (7)$$

$$\text{subject to } F_{t_k}^{i,s} > F_{t_{k+1}}^{i,s}, \quad \forall k \in [1, K-1], \quad t_k \in T$$

$$F_{t_K}^{i,s} > F_c^{i,s}, \quad \forall c \in \mathcal{Y} \setminus T, \quad t_K \in T,$$

$$F^{i,s} = h(z_{i,s}; W_s, b_s) = z_{i,s} \cdot W_s + b_s, \quad \forall (\theta_s, W_s, b_s) \in \mathcal{M}^{\text{train}},$$

$$x_i^{\text{perturb}} = \text{Clamp}(x_i^{\text{benign}} + \delta_T), \quad \forall x_i^{\text{benign}} \in \mathcal{D}^{\text{train}},$$

where $d_s$ is the output feature dimension of a DNN backbone, $f_{\theta_s}(\cdot)$, and let $d_{\text{mean}}$ the mean feature dimension of the $S$ DNNs. $\beta_s = \sqrt{\frac{d_{\text{mean}}}{d_s}}$ is introduced to normalize the $\ell_2$ distances which exhibit large variations among different DNNs.

With the optimized $z_{i,s}^*$, we update the UAP $\delta_T$ by one-step gradient descent,

$$\delta_T \Leftarrow \delta_T - \gamma \cdot \frac{\partial}{\partial \delta} \Big( \frac{\lambda}{S \cdot I} \sum_{i,s} \beta_s \cdot ||z_{i,s}^* - f(x_i^{\text{benign}} + \delta; \theta_s)||_2^2 + ||\delta||_2^2 \Big)|_{\delta = \delta_T}, \quad (8)$$

where $\lambda$ is the trade-off parameter and $\gamma$ the step size.

### 3.3 Stochastic Mini-Batch-of-Data-and-Model Optimization

In practice, when the training dataset $\mathcal{D}^{\text{train}}$ and/or the training model ensemble $\mathcal{M}^{\text{train}}$ are large, we can not afford the full-batch optimization, even for a single large model, due to the GPU memory constraint. To handle this, we resort to stochastic mini-batch and mini-model learning. During each iteration in the optimization, we sample a mini-batch of training images with a predefined batch size (e.g., 64), and sample a number of models (e.g., 4) to fit the GPU memory constraint. **This is critical for learning UAPs from a large ensemble of models.**

**Remarks: AllAttacK as Potential Analytical Tools.** We emphasize that learning AllAttacK UAPs is not merely to "break" models, but to potentially use them as *analytical instruments*—tools for uncovering latent weaknesses and generating training signals to address them. So, we choose to focus on a large-scale analyses of AllAttacK UAPs while extending existing single-model-single-image ordered top-K attack methods in solving Eqn. 4.

## 4 Experiments

In this section, we test our proposed AllAttacK in the ImageNet-1k benchmark (Russakovsky et al., 2015) with strong performance obtained. **Our PyTorch code will be released.** Implementation details are in Appendix B with runtime results in Appendix C.

**Models.** We use 27 models in learning and testing AllAttacK UAPs.

- **ImageNet-1k trained models:** `ResNets`-(18, 34, 50, 101) (He et al., 2016) (with two differently trained checkpoints of ResNet-50), `DenseNets`-(121, 161, 169, 201) (Huang et al., 2017), `HRNet`-(W18, W30) (Wang et al., 2020), `ConvNeXt`-(Tiny, Small, Base) (Liu et al., 2022), `DEiT`-Small (Touvron et al., 2021), `DEiT3`-(small, medium) (Touvron et al., 2022), `ViT`-Base (Dosovitskiy et al., 2020), `MLPMixer`-Base (Tolstikhin et al., 2021), `ConvMixer`-768 (isotropic architecture) (Trockman & Kolter, 2023) and `Swin`-Base (Liu et al., 2021). Their pretrained checkpoints are souced from the `timm` package (Wightman, 2019).
- **Models with large-scale pretraining and ImageNet-1k fine-tuning:** `ConvNeXtV2`-H (Woo et al., 2023) pretrained by Masked Image Modeling (MIM) on ImageNet-21k, OpenAI `CLIP ViT`-B (Radford et al., 2021) pretrained by contrastive language image pretraining (CLIP) with a massive number ($\sim$400M) of proprietary image-caption pairs, and `EVA2 ViT`-B (Fang et al., 2023) pretrained by combining MIM and CLIP.
- **Adversarially-trained models:** We also test three adversariall-trained robust models (whose Top-1 accuracy on benign data are significantly sacrificed): ResNet-50$_{robust}^1$ (Engstrom et al., 2019), ResNet-50$_{robust}^2$ and WideResNet-50$_{robust}$ (Salman et al., 2020), sourced from the Robust-Bench (Croce et al., 2020).

Table 1: The `mean` ASRs of learned AllAttacK UAPs for a 18-model ensemble. $\ell_2$ norms are in Table 12 in the Appendix, due to space limit.

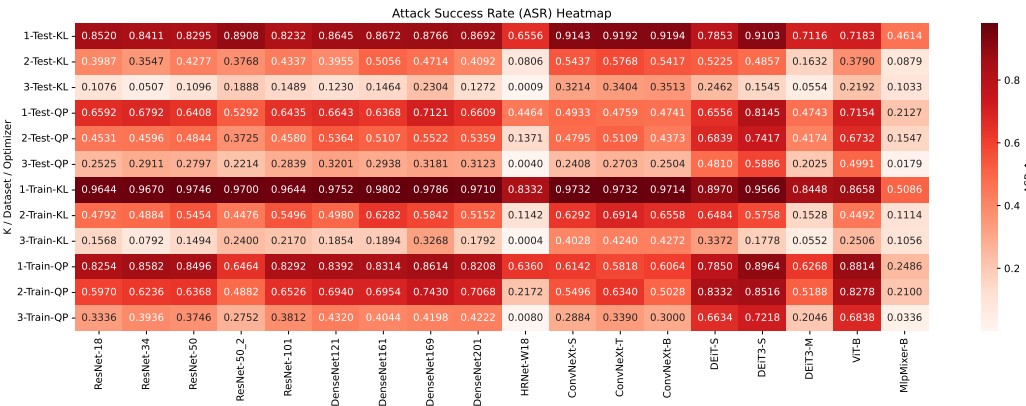

 

Figure 3: Examples of learned 18-model UAPs for the targets sequentially, $T = [\text{Coffeepot}, \text{Foxhound}, \text{Tub}]$. The left/right three UAPs are learned by QP/KL.

**Metrics.** We evaluate our method using attack success rates (ASRs) and $\ell_1, \ell_2$ and $\ell_\infty$ norms of perturbations. For each number of targets, $K$ (e.g., $K = 1, \cdots, 6$), we randomly sample 5 lists of ordered top-$K$ targets. For each ordered top-$K$ target list $T$, we learn the UAP $\delta_T$ using both KL and QP. We use different seeds in optimization in learning each of the universal perturbations. We compute the ASR with respect to the `Best`, `Worst` and `Mean` protocols. By `Best`, it means we call it a success attack if any of the 5 samplings does so for an image. By `Worst`, it means we call it a failure if any of the 5 samplings does so for an image. By `Mean`, we use the mean success rate among the 5 samplings for an image. Then, the ASRs of a method are computed by the average over the set of data $\mathcal{D}^{\text{train}}$ or $\mathcal{D}^{\text{test}}$. **Due to space limit, we report the results using the** `Mean` **ASRs and $\ell_2$ norms in the main text, and provide full results in Appendix F.**

Table 2: The 18-model ensemble learned UAPs are transferrable to unseen testing model on unseen testing images.

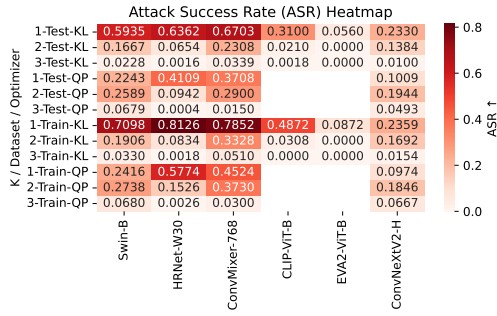

### 4.1 DOUBLY-TRANSFERRABLE ALLATTACK UAPs

We first learn UAPs using a 18-model ensemble for $K = 1, 2, 3$, and then test the learned UAPs on 6 DNNs which are not included in the 18-model ensemble. Table 1 shows ASRs for the 18 training models on both training and testing images. Fig. 3 shows examples of learned UAPs.

- On training images, top-1 UAPs by KL (i.e., 1-Train-KL) can achieve consistently high ASRs, greater than 83%, except for MlpMixer-B (50.9%). UAPs by QP for $K = 2, 3$ remain promising ASRs on most models.
- On testing images, top-1 UAPs by KL (i.e., 1-Test-KL) remains highly transferrable, mostly achieving more than 70% ASRs, except for HRNet-W18 and MlpMixer-B. UAPs by QP for $K = 2, 3$ also have promising transferrability.
- For Top-1 perturbations, the KL formulation works better than the QP formulation. When $K > 1$, the QP formulation is significantly better, which is consistently observed throught all experiments.
- Fooling a disparate ensemble of DNNs entailing "tricky yet meaningful" signals that respect and resemble the targets. As pointed out in (Park & Kim, 2021), ConvNets tend to capture more high-

frequency texture features, while ViTs tends to capture more low-frequency shape related features. So, fooling them all enforces the learned UAPs not only to respect those spectrum information in isolation, but also to "shut off" information of those images. **"Intepretable" perturbations thus emerge** (see Fig. 2 and Fig. 3).

Table 2 shows the testing model transferability of the 18-model ensemble learned UAPs:

• Top-1 UAPs by KL (i.e., 1-Train-KL and 1-Test-KL) achieve strong transferrability. For models with similar architectures such as HRNet-W30 (similar to HRNet-W18 in the 18-model ensemble), the ASR on training images is 81.3%, and the ASR on testing images can still reach 63.6%. The ASRs on Swin-B and ConvMixer-768 that are different from models in training are also quite high. More impressively, for the CLIP-ViT-B, the ASRs can reach 48.7% and 31.0% on training and testing images respectively. The EVA2-ViT-B shows the highest resistance to the UAPs.

• Top-2 UAPs have promising ASRs on Swin-B, ConvMixer-768 and ConvNeXtV2-H.

**These results, especially the double-transferability across testing models and testing images, represent a significant leap forward from the prior art, as well as broader impacts as dicussed in Appendix A**.

### 4.2 DATA TRANSFERRABLE SINGLE-MODEL ALLATTACK UAPS

We conduct three groups of experiments. **Single-model AllAttacK UAPs are aggressive along the label target axis (tested up to Top-6), and transfer very well along the data axis** (i.e., across both training images and testing images).

Table 3: The `mean` ASRs and $\ell_2$ norms of single-model AllAttacK UAPs for four models.

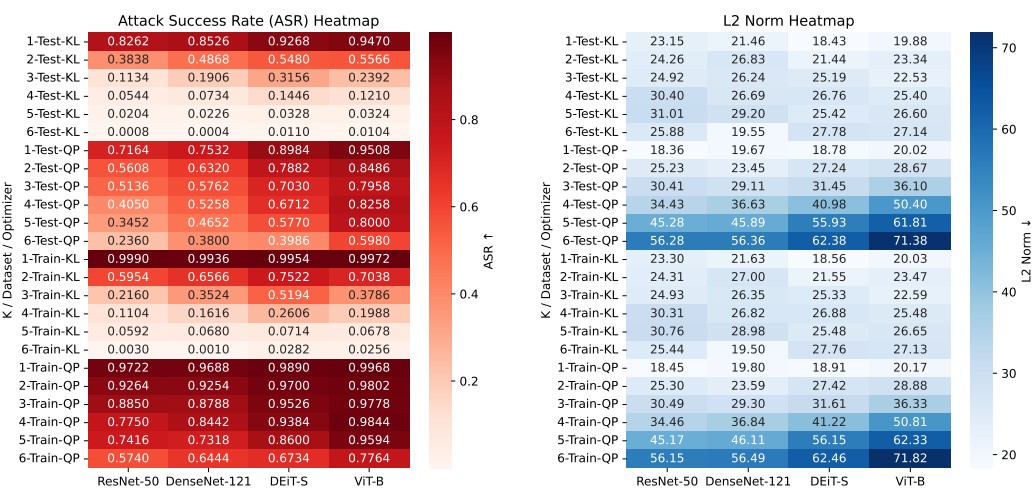

Table 3 shows results for 4 representative and commonly used DNNs: ResNet-50, DenseNet-121, ViT-Base and DEiT-Small, with $K = 1, 2, \cdots, 6$. For example, we obtain about 60% ASR on testing images even for ordered Top-6 targeted UAPs for ViT-B (Dosovitskiy et al., 2020).

Table 4 shows results for 6 relatively larger DNNs: Swin-B, HRNet-W30, ConvMixer-768, CLIP-ViT-B, EVA2-ViT-B and ConvNeXtV2-H, with $K = 1, 2, 3$. We obtain much higher ASRs across the board at the expense of increasing the perturbation energies compared with Table 3, which shows those models are still vulnerable once the $\ell_2$ energy of perturbation is slightly increased.

Table 5 shows results for 3 adversarially-trained ResNet-50 models. Similarly, those robustified models are still vulnerable at the expense of even further increased perturbation energies.

### 4.3 DATA & TRAINING-MODEL-ENSEMBLE TRANSFERRABLE ALLATTACK UAPS

We perform three groups of experiments using three combinations of four DNNs: ResNet-50 (R50), DenseNet-121 (D121), ViT-Base (V-B) and DEiT-Small (D-S), with $K = 1, \cdots, 6$.

Table 6 shows the results. The ASRs on testing images for $K \leq 3$ remains reasonably high.

• **R50+D121**: we obtain around 37% ASR for $K = 3$ by QP.

Table 4: The `mean` ASRs and $\ell_2$ norms of single-model AllAttacK UAPs for the 6 models in Table 2.

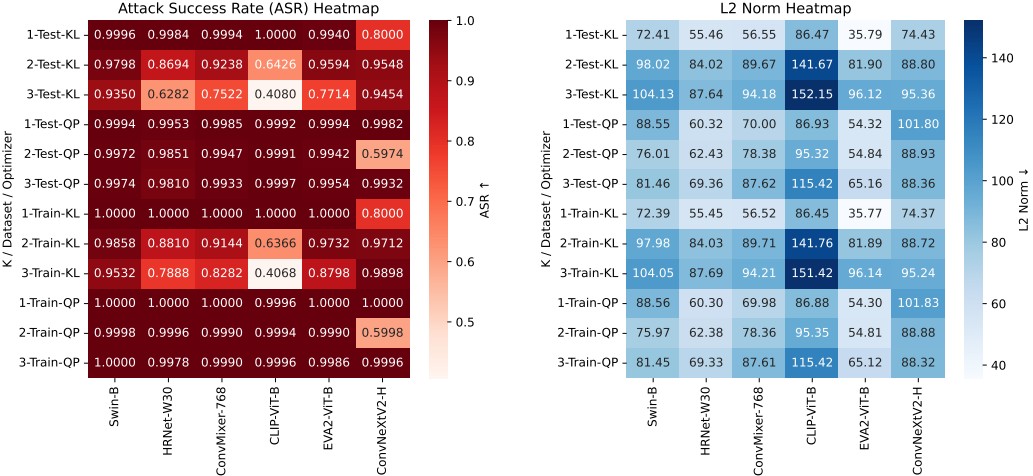

Table 5: The `mean` ASRs and $\ell_2$ norms of single-model AllAttacK UAPs for three adversarially-trained ResNet-50 models.

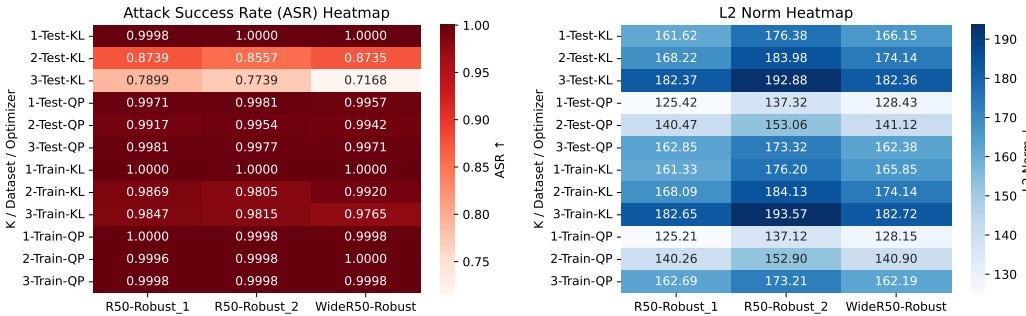

- **V-B+D-S**: we obtain around 67% ASR for $K = 3$, and around 19% ASR for $K = 6$, both by QP.
- **R50+D121+V-B+D-S**: we obtain around 21% ASR for $K = 3$ by QP.

When $K > 3$, the ASRs drop quickly for the ensembles with R50+D121, but remains promising for the V-B+D-S ensemble with around 18% for Top-6.

### 4.4 ALLATTACK UAPS INDUCE BENIGN IMAGES AS "NEGLIGIBLE NOISES" TO DNNS

We compute the Pearson Correlation Coefficient (PCC) between the logits (Eqn. 1) of an AllAttacK UAP $\delta_T$ itself and the logits of an image (benign or perturbed). The PCC can show if they are positively correlated (if the PCC $\in (0, 1]$), not linearly correlated (if the PCC $= 0$), or negatively correlated (if the PCC $\in [-1, 0)$).

$$\text{mPCC}(\delta, \text{ref}) = \frac{1}{K \cdot I \cdot S} \sum_{k,i,s} \text{PCC}(F(\delta_{T_k}; \Theta_s), F(x_i^{\text{ref}}; \Theta_s)), \tag{9}$$

where ref $\in \{\text{benign, perturbed}\}$, $K$ the total number of top-k ($k \in [1, \cdots, K]$), $I$ is the number of testing images that are successfully attacked, and $S$ the number of models.

As shown in Table 7, we can see the effects of learned AllAttacK UAPs, "steering the logits of DNNs regardless of the benign images".

### 4.5 SEMANTICALLY COHERENT ORDERED TOP-K TARGETS ARE EASIER TO ATTACK

In our main experiments, the random sampling strategy serves as a fair criterion benchmarking the general attacking power of our AllAttacK. To show that **semantically coherent top-K targets are easier to attack**, we conduct one experiment using the four-model combination setting in Table 6. Following the observations in (Zhang & Wu, 2020), we exploit the label similarities in constructing a list of Top-$K$ targets. A Top-1 target is first randomly sampled, and the remaining $K$-1 targets are the nearest neighbors to the Top-1 target based on the Glove (Pennington et al., 2014) embedding

Table 6: The `mean` ASRs (left) and $\ell_2$ (right) of learned AllAttacK UAPs for 3 different combinations of 4 DNNs. $\ell_2$ norms are in Table 14 in the Appendix, due to space limit.

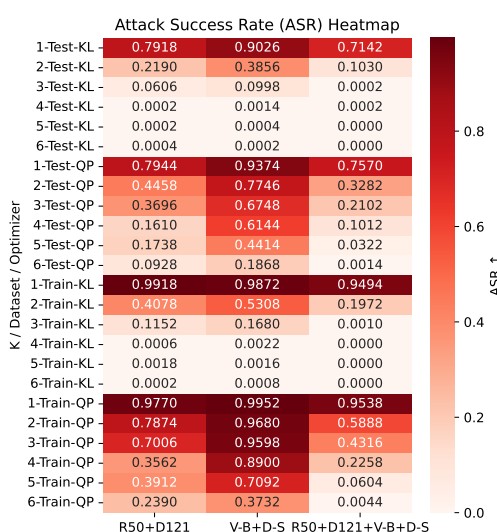

Table 7: The mPCC (Eqn. 9) comparisons.

| Models | $x^{benign}$ | $x^{perturbed}$ |
|---|---|---|
| ResNet-50 | 0.06 | 0.85 |
| DenseNet-121 | 0.09 | 0.84 |
| DEiT-S | 0.01 | 0.79 |
| ViT-B | 0.00 | 0.81 |
| R50+D121 | 0.10 | 0.83 |
| V-B+D-S | 0.00 | 0.80 |
| R50+D121+V-B+D-S | 0.04 | 0.78 |

Table 8: Results of ordered Top-5 UAPs with semantically-coherent targets for the four-model ensemble in Table 6.

| $K$ / Opt. | Data | ASR↑ | $\ell_2 \downarrow$ |
|---|---|---|---|
| 5 / QP | Test | $0.0322 \rightarrow 0.3342$ | $69.65 \rightarrow 88.35$ |
| | Train | $0.0604 \rightarrow 0.7495$ | $69.63 \rightarrow 87.15$ |

similarities. Table 8 shows the results. **In practice, adversary often chooses semantically coherent attack targets, so the vulnerability reported based on randomly sampled targeted represents the lower-bound, and the concerns in practice are significantly higher.** See Appendix D for more details.

## 5 RELATED WORK

**Universal Adversarial Perturbations (UAPs).** One remarkable discovery in (Szegedy et al., 2013) was that adversarial attacks have the ability to transfer to models trained with different hyperparameters or training sets than those the adversarial attack was generated with. (Liu et al., 2016) later investigates the transferability of adversarial examples among different neural network architectures, differentiating between targeted and non-targeted attacks. (Liu et al., 2016) introduces an ensemble-based attack method similar in spirit to our own AllAttack, enabling successful targeted adversarial attacks. This further extended to Universal Adversarial Perturbations (Hendrik Metzen et al., 2017; Moosavi-Dezfooli et al., 2017; Shafahi et al., 2020). (Moosavi-Dezfooli et al., 2017) and (Shafahi et al., 2020) achieve a single perturbation that can be applied to a large number of images for a model and prevent correct classification, While (Hendrik Metzen et al., 2017) extended the concept of universal attacks to the domain of segmentation. (Zhang et al., 2020a) first observes the possibility of common class-specific "features" across universal perturbations. (Benz et al., 2020) extends the specificity of UAPs to only change predictions for one specified "source" class and change them to a prescribed "sink" class, leaving all other classes unchanged.

## 6 CONCLUSION

This paper studies the problem of learning ordered top-K targeted Universal Adversarial Perturbations (UAPs). Learning Top-1 targeted UAPs has been proven challenging, and learning ordered top-K UAPs has not been studied in the literature. This paper makes a significant leap forward by showing ordered Top-$K$ target labels can transfer (for $K \geq 1$) across both data and models. Especially, ordered top-K UAPs can transfer across both unseen testing images and unseen testing models, when they are optimized using a large ensemble of disparate models. This paper presents AllAttacK, revisiting and expanding targeted UAPs along three axes (data, model and target labels). AllAttacK is built on two state-of-the-art single-model-single-image ordered top-K targeted attack methods, Adversarial Distillation and QuadAttacK. The proposed AllAttacK is thoroughly evaluated in experiments with more than 500 UAPs learned for an ensemble of 27 diverse models, showing that the resulting perturbations not only exhibit strong transferability but also display intriguing, interpretable characteristics.

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

## A  BROADER IMPACTS

The promising transferability of the learned ordered top-K UAPs to unseen models, especially those at the so-called foundation model level (e.g., the CLIP ViT-B in Table 2), might be exploited in a harmful way for applications built on those models. Powerful defense methods should be studied, which we will investigate in our future work. We will also release our source code to encourage more research on studying defense methods against the proposed AllAttacK.

## B  DETAILS OF OPTIMIZATION

We build on the released code of QuadAttacK (Paniagua et al., 2023). In all experiments, we use the *AdamW* optimizer with a learning rate of 0.002 to minimize our presented objectives (Eqn. 5 and Eqn. 7 in the main paper). We run all configurations for 50 epochs on the training images and models.

In optimization, both the QP and KL methods require choosing a hyperparameter $\lambda$ for the loss term focused on satisfying the Ordered Top-K constraint (Eqn. 5 and Eqn. 8 in the main paper). There is no "optimal" value for this parameter, it is a trade-off parameter that selects a point on the ASR vs Energy tradeoff curve. To facilitate finding successful attacks on more challenging cases (e.g., the 18-model ensemble attack), we perform our optimization for multiple $\lambda$ values and select the smallest energy that obtained a non-negligible ASR. For QP we search in $\lambda \in \{100, 150\}$ and for KL we search in $\lambda \in \{1000, 1500\}$, where we choose different magnitudes for QP and KL due to the different spaces that losses operate in.

We use 1 Nvidia A100 80G GPU in all our experiments. We run multiple configurations (e.g., different $K$'s and DNN combinations) in parallel across 4-8 GPUs on our server.

## C  RUNTIMES

To better understand the complexity of our method, we have profiled the *QP* attack on 4 different configurations. We report the recorded runtimes of our method **per-epoch** in Table 9. Every attack configuration in this paper runs for *50 epochs*. Each configuration has been profiled on a single Nvidia A100 80G GPU and averaged over the course of 12 epochs. We note runtime does not seem to be affected significantly by the choice of $K$ in the attack, but is more directly affected by the models being attacked.

Table 9: Average epoch runtime in seconds on 4 different configurations averaged over 12 trials.

| Model / K | K=2 | K=5 |
|---|---|---|
| **ResNet-50** | 58s | 55s |
| **4 Model Ensemble** | 164s | 164s |

## D  EXPLICIT ATTACK TARGET SAMPLING

In the results presented in the main paper, we have chosen to sample *ordered Top-K* classes for each attack trial at random, as stress-tests and the fair criteria benchmarking the underlying vulnerability of DNNs. Here we include extended results showing the effect of sampling *similar* and *contrasting* classes within the desired *Top-K* targets. We extend these results by showing attacks on *similar* classes in Table 10 (extending Table 8 in the main paper) and *contrasting* classes in Table 11. We sample these classes through a class similarity matrix generated by the word embeddings used in the *KL* method (Zhang & Wu, 2020). In this extended sampling we first sample the Top-1 class at random, and pick the rest of the $K - 1$ classes by determining the most similar or dissimilar classes to the Top-1 class based on the class similarity matrix.

## E  ALL LEARNED PERTURBATIONS

We develop a HTML based interactive visualization tool (Fig. 4). **Please check the** index.html **for browsing all the perturbations in this supplementary material.** Due to the file size limit

Table 10: Results of our *similar* sampling strategy attacking the 4-model ensemble (ResNet-50, DenseNet-121, DEiT-S, and ViT-B).

| Protocol | Attack Method | ResNet-50 \| DenseNet121 \| ViT-B \| DEiT-S | | | | | | | | | | | |
| --- | --- | --- | --- | --- | --- | --- | --- | --- | --- | --- | --- | --- | --- |
| | | Best | | | | Mean | | | | Worst | | | |
| | | ASR↑ | $\ell_1\downarrow$ | $\ell_2\downarrow$ | $\ell_\infty\downarrow$ | ASR↑ | $\ell_1\downarrow$ | $\ell_2\downarrow$ | $\ell_\infty\downarrow$ | ASR↑ | $\ell_1\downarrow$ | $\ell_2\downarrow$ | $\ell_\infty\downarrow$ |
| Top-3 | $QP_{test}$ | **0.9520** | 22654.61 | 73.98 | **0.8688** | **0.9444** | 26898.47 | 87.19 | 0.9368 | 0.9300 | 29744.05 | 95.52 | **0.9536** |
| | $QP_{train}$ | **1.0000** | 22549.70 | 73.81 | 0.8703 | **0.9970** | 26803.68 | 87.01 | 0.9376 | **0.9920** | 29669.37 | 95.38 | **0.9535** |
| | $KL_{test}$ | 0.4050 | 29010.95 | 92.71 | 0.9301 | 0.3044 | 32099.89 | 102.19 | 0.9546 | 0.1880 | **29475.97** | **95.01** | 0.9578 |
| | $KL_{train}$ | 0.9100 | 28633.45 | 92.09 | 0.9373 | 0.7654 | 31727.67 | 101.54 | 0.9626 | 0.5640 | **29215.48** | **94.58** | 0.9650 |
| Top-2 | $QP_{test}$ | **0.9710** | 27444.99 | 89.03 | 0.9429 | **0.9652** | 24968.47 | 81.26 | **0.9182** | **0.9540** | 22925.88 | 74.64 | **0.8917** |
| | $QP_{train}$ | **1.0000** | 23775.09 | 77.31 | 0.8970 | **0.9994** | 24915.94 | 81.17 | 0.9194 | **0.9980** | 27329.03 | 88.79 | 0.9447 |
| | $KL_{test}$ | 0.6440 | 27966.56 | 90.51 | 0.9556 | 0.5766 | 29138.69 | 93.47 | 0.9493 | 0.4350 | 29567.84 | 94.82 | 0.9548 |
| | $KL_{train}$ | 0.9380 | 27243.88 | 88.04 | 0.9477 | 0.8650 | 28999.14 | 93.20 | 0.9533 | 0.7730 | 29523.81 | 94.69 | 0.9577 |

Table 11: Results of our *contrasting* sampling strategy attacking the 4-model ensemble (ResNet-50, DenseNet-121, DEiT-S, and ViT-B).

| Protocol | Attack Method | ResNet-50 \| DenseNet121 \| ViT-B \| DEiT-S | | | | | | | | | | | |
| --- | --- | --- | --- | --- | --- | --- | --- | --- | --- | --- | --- | --- | --- |
| | | Best | | | | Mean | | | | Worst | | | |
| | | ASR↑ | $\ell_1\downarrow$ | $\ell_2\downarrow$ | $\ell_\infty\downarrow$ | ASR↑ | $\ell_1\downarrow$ | $\ell_2\downarrow$ | $\ell_\infty\downarrow$ | ASR↑ | $\ell_1\downarrow$ | $\ell_2\downarrow$ | $\ell_\infty\downarrow$ |
| Top-3 | $QP_{test}$ | 0.8720 | 28438.82 | 92.01 | 0.9593 | 0.8408 | 28431.03 | 91.67 | 0.9484 | 0.7820 | 28058.32 | 90.07 | 0.9303 |
| | $QP_{train}$ | 0.9930 | 28221.33 | 91.70 | 0.9603 | 0.9882 | 28205.78 | 91.29 | 0.9521 | 0.9830 | 27630.56 | 89.26 | 0.9375 |
| | $KL_{test}$ | 0.3010 | 35066.20 | 111.85 | 0.9718 | 0.2380 | 34849.41 | 110.98 | 0.9717 | 0.1600 | 34652.71 | 111.12 | 0.9785 |
| | $KL_{train}$ | 0.6250 | 34828.42 | 111.51 | 0.9766 | 0.4902 | 34749.13 | 110.87 | 0.9758 | 0.3440 | 34427.40 | 111.05 | 0.9829 |
| Top-2 | $QP_{test}$ | 0.8470 | 25908.68 | 83.80 | 0.9354 | 0.7834 | 25880.11 | 83.53 | 0.9254 | 0.7340 | 27065.69 | 86.79 | 0.9332 |
| | $QP_{train}$ | 0.9950 | 24909.85 | 81.07 | 0.9309 | 0.9648 | 25636.74 | 83.08 | 0.9298 | 0.9020 | 26840.08 | 86.39 | 0.9376 |
| | $KL_{test}$ | 0.7470 | 33048.54 | 105.23 | 0.9687 | 0.6354 | 31940.24 | 102.15 | 0.9678 | 0.4310 | 31334.47 | 101.32 | 0.9729 |
| | $KL_{train}$ | 0.8620 | 32614.94 | 103.73 | 0.9696 | 0.7428 | 31989.64 | 102.20 | 0.9675 | 0.4860 | 31572.03 | 101.63 | 0.9698 |

of supplementary material (100M), we include UAPs learned with 3 seeds. We will release all the learned UAPs, together with the source code after the review process.

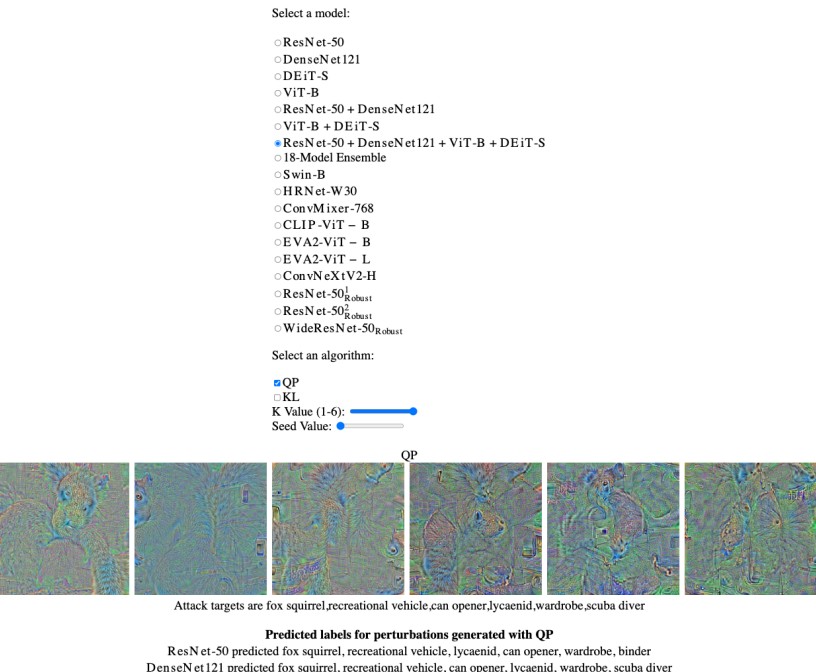

Figure 4: The HTML visualizer interface.

To illustrate the quasi-imperceptibility of UAPs, Fig. 5 shows some examples.

# F  DETAILED QUANTITATIVE RESULTS

In the main paper, due to space limit, we report results using mean ASRs and $\ell_2$ norms. In this section, we report the full results in terms of Best, Mean and Worst ASRs and $\ell_1$, $\ell_2$ and $\ell_\infty$ norms.

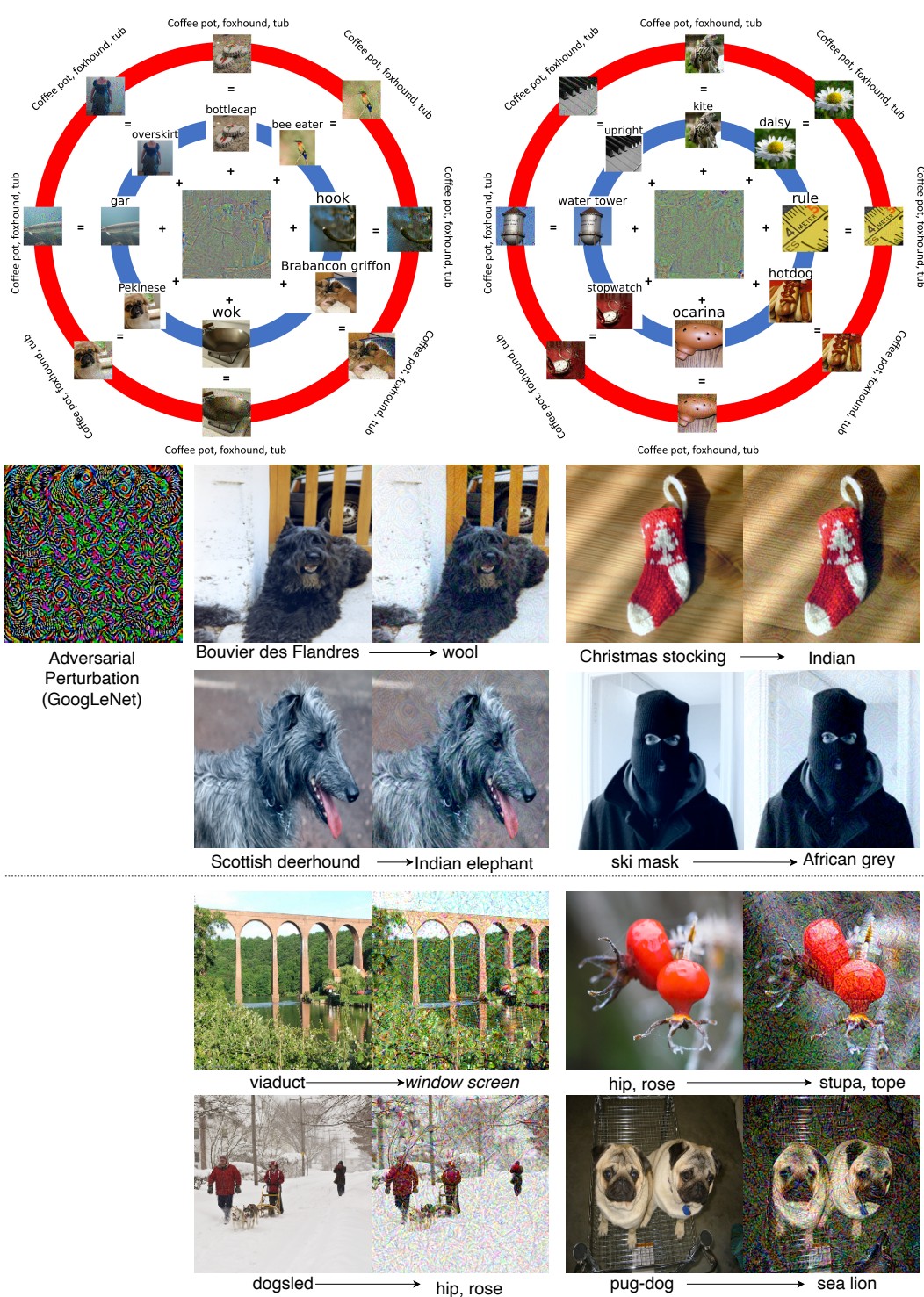

Figure 5: **Top:** Examples of learned ordered top-3 UAPs with $T = [\texttt{Coffee pot, Foxhound, Tub}]$. *Top-Left*: attacking the 4-model ensemble (ResNet-50, DenseNet-121, ViT-B and DEiT-S) and *Top-Right:* attacking DEiT-S. The center shows the perturbations normalized to [0,1] for the sake of visualization. On the inner blue circle are clean images with their ground-truth label. On the outer red circle are perturbed images that successfully fool the model(s). **Middle**: Non-targeted UAPs using GoogLeNet for images in ImageNet-1000 validation in (Moosavi-Dezfooli et al., 2017). **Bottom**: Top-1 targeted UAPs learned using VGG-16 for images in ImageNet-1000 validation in (Liu et al., 2016). **We note that quasi-imperceptible perturbations are the common practice in learning UAPs.**

Table 12: The `mean` ASRs and $\ell_2$ norms of learned AllAttacK UAPs for a 18-model ensemble, extending Table 1 in the main paper.

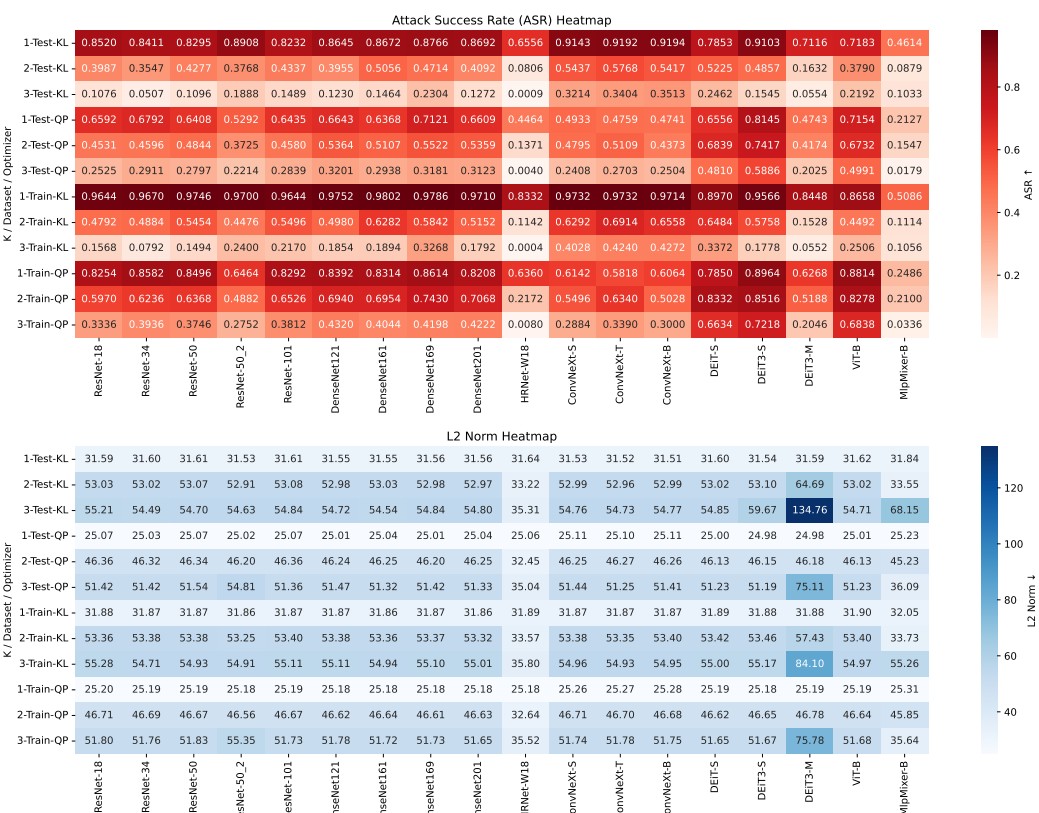

Table 13: The `mean` ASRs and $\ell_2$ norms of applying the 18-model AllAttacK UAPs for 6 testing models, extending Table 2 in the main paper.

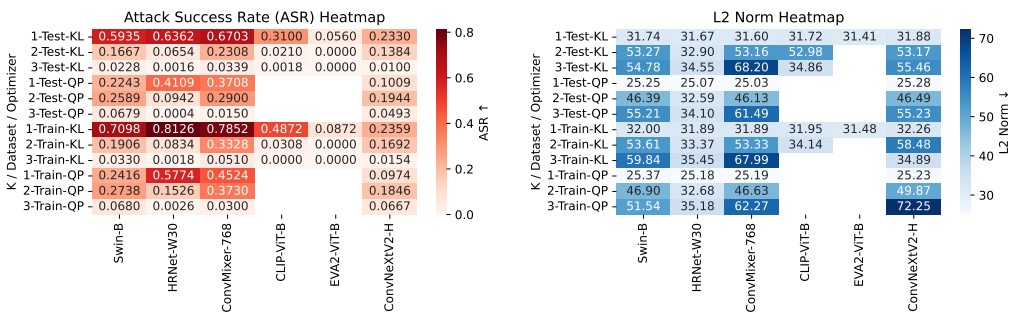

Table 14: The `mean` ASRs and $\ell_2$ norms of learned AllAttacK UAPs for a 18-model ensemble, extending Table 6 in the main paper.

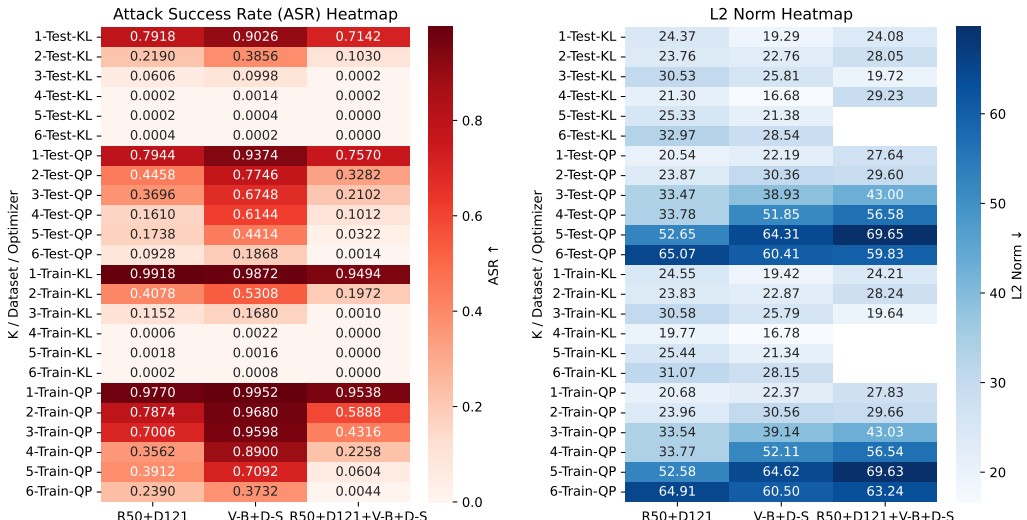

Table 15: Full results for ResNet-50 and DenseNet-121 in Table 3 in the main paper.

| | | ResNet-50 | | | | | | | | | | | |
|---|---|---|---|---|---|---|---|---|---|---|---|---|---|
| Protocol | Attack Method | Best | | | | Mean | | | | Worst | | | |
| | | ASR↑ | $\ell_1$↓ | $\ell_2$↓ | $\ell_\infty$↓ | ASR↑ | $\ell_1$↓ | $\ell_2$↓ | $\ell_\infty$↓ | ASR↑ | $\ell_1$↓ | $\ell_2$↓ | $\ell_\infty$↓ |
| Top-6 | $KL_{test}$ | 0.0020 | 7423.81 | 25.01 | 0.6499 | 0.0008 | 7534.61 | 25.88 | 0.5851 | 0.0000 | - | - | - |
| | $KL_{train}$ | 0.0100 | 7129.05 | 24.65 | 0.5437 | 0.0030 | 7341.37 | 25.44 | 0.5781 | 0.0000 | - | - | - |
| | $QP_{test}$ | **0.2880** | 17380.21 | 56.73 | 0.7784 | **0.2360** | 17082.95 | 56.28 | 0.7882 | **0.1980** | 19745.88 | 64.20 | 0.8107 |
| | $QP_{train}$ | **0.6190** | 15870.16 | 52.67 | 0.7924 | **0.5740** | 17006.52 | 56.15 | 0.7928 | **0.5020** | 17019.15 | 56.37 | 0.8018 |
| Top-5 | $KL_{test}$ | 0.0430 | 9103.39 | 31.60 | 0.6410 | 0.0204 | 9036.74 | 31.01 | 0.6483 | 0.0060 | 8569.89 | 29.76 | 0.7003 |
| | $KL_{train}$ | 0.0830 | 8816.55 | 30.35 | 0.7230 | 0.0592 | 8937.48 | 30.76 | 0.6534 | 0.0220 | 9191.38 | 31.68 | 0.6515 |
| | $QP_{test}$ | **0.4020** | 12784.94 | 43.21 | 0.7235 | **0.3452** | 13484.07 | 45.28 | 0.7624 | **0.2950** | 12049.47 | 41.35 | 0.7621 |
| | $QP_{train}$ | **0.8030** | 13297.80 | 44.62 | 0.7875 | **0.7416** | 13420.23 | 45.17 | 0.7695 | **0.6740** | 15200.24 | 50.91 | 0.7959 |
| Top-4 | $KL_{test}$ | 0.1130 | 8609.84 | 30.10 | 0.6598 | 0.0544 | 8860.27 | 30.40 | 0.6863 | 0.0190 | 9231.51 | 31.40 | 0.6954 |
| | $KL_{train}$ | 0.1750 | 8498.26 | 29.82 | 0.6726 | 0.1104 | 8820.95 | 30.31 | 0.6995 | 0.0370 | 9195.29 | 31.28 | 0.7181 |
| | $QP_{test}$ | **0.4260** | 8754.73 | 31.12 | 0.7082 | **0.4050** | 9966.86 | 34.43 | 0.7224 | **0.3750** | 11121.31 | 37.48 | 0.7065 |
| | $QP_{train}$ | **0.8480** | 8734.67 | 31.09 | 0.7066 | **0.7750** | 9965.70 | 34.46 | 0.7257 | **0.7150** | 11122.73 | 37.53 | **0.7094** |
| Top-3 | $KL_{test}$ | 0.1380 | 7820.22 | 27.03 | 0.6499 | 0.1134 | 7189.22 | 24.92 | 0.5911 | 0.0810 | 6669.33 | 23.69 | 0.6076 |
| | $KL_{train}$ | 0.2710 | 7418.35 | 25.69 | 0.6184 | 0.2160 | 7180.67 | 24.93 | 0.5934 | 0.1320 | 6655.45 | 23.70 | 0.6125 |
| | $QP_{test}$ | **0.5300** | 9423.68 | 32.46 | 0.6738 | **0.5136** | 8798.63 | 30.41 | 0.6679 | **0.4780** | 8258.90 | 28.91 | 0.6242 |
| | $QP_{train}$ | **0.9090** | 8244.32 | 28.89 | 0.6281 | **0.8850** | 8811.83 | 30.49 | 0.6706 | **0.8640** | 9455.38 | 32.60 | 0.6750 |
| Top-2 | $KL_{test}$ | 0.4540 | 7050.01 | 24.58 | 0.5399 | 0.3838 | 6961.05 | 24.26 | 0.5593 | 0.3100 | 6742.45 | 23.83 | 0.5456 |
| | $KL_{train}$ | 0.7400 | 7080.44 | 24.70 | **0.5413** | 0.5954 | 6972.23 | 24.31 | 0.5599 | 0.4980 | 6727.38 | 23.80 | 0.5459 |
| | $QP_{test}$ | **0.6120** | 7598.11 | 26.47 | 0.5779 | **0.5608** | 7264.53 | 25.23 | 0.5983 | **0.5200** | 7599.71 | 26.09 | 0.6272 |
| | $QP_{train}$ | **0.9620** | 6744.20 | 23.78 | 0.5437 | **0.9264** | 7280.19 | 25.30 | 0.5976 | **0.9040** | 7612.42 | 26.15 | 0.6162 |
| Top-1 | $KL_{test}$ | **0.8490** | 7236.13 | 24.65 | 0.6471 | **0.8262** | 6724.56 | 23.15 | **0.5118** | **0.7810** | 6284.51 | 22.04 | **0.5044** |
| | $KL_{train}$ | **1.0000** | 6601.40 | 22.58 | **0.4444** | **0.9990** | 6772.62 | 23.30 | **0.5129** | **0.9980** | 7297.60 | 24.83 | 0.6539 |
| | $QP_{test}$ | 0.7930 | **4525.26** | **16.60** | 0.5165 | 0.7164 | **5220.64** | **18.36** | 0.5151 | 0.6460 | **5486.59** | **18.87** | 0.5320 |
| | $QP_{train}$ | 0.9830 | **4551.12** | **16.70** | 0.5169 | 0.9722 | **5245.27** | **18.45** | 0.5202 | 0.9640 | **5505.92** | **18.94** | 0.5392 |

| | | DenseNet121 | | | | | | | | | | | |
|---|---|---|---|---|---|---|---|---|---|---|---|---|---|
| Protocol | Attack Method | Best | | | | Mean | | | | Worst | | | |
| | | ASR↑ | $\ell_1$↓ | $\ell_2$↓ | $\ell_\infty$↓ | ASR↑ | $\ell_1$↓ | $\ell_2$↓ | $\ell_\infty$↓ | ASR↑ | $\ell_1$↓ | $\ell_2$↓ | $\ell_\infty$↓ |
| Top-6 | $KL_{test}$ | 0.0010 | **5610.15** | 19.50 | 0.3955 | 0.0004 | 5560.38 | 19.55 | 0.4853 | 0.0000 | - | - | - |
| | $KL_{train}$ | 0.0040 | **5490.69** | 19.55 | 0.5415 | 0.0010 | 5542.69 | 19.50 | 0.4952 | 0.0000 | - | - | - |
| | $QP_{test}$ | **0.4130** | 15012.97 | 50.37 | 0.7441 | **0.3800** | 17065.69 | 56.36 | 0.7497 | **0.3260** | 19773.50 | 64.14 | 0.7576 |
| | $QP_{train}$ | **0.7040** | 14884.52 | 49.91 | 0.7066 | **0.6444** | 17093.06 | 56.49 | 0.7500 | **0.5300** | 17430.55 | 57.84 | 0.7887 |
| Top-5 | $KL_{test}$ | 0.0300 | **8175.26** | 28.92 | 0.6362 | 0.0226 | **8359.95** | 29.20 | 0.5907 | 0.0170 | **8400.00** | 29.52 | 0.5774 |
| | $KL_{train}$ | 0.1000 | **8007.07** | 28.47 | 0.6436 | 0.0680 | **8273.76** | 28.98 | 0.5960 | 0.0420 | **8601.66** | 29.80 | 0.5629 |
| | $QP_{test}$ | **0.5120** | 11333.09 | 39.52 | 0.6606 | **0.4652** | 13640.25 | 45.89 | 0.7054 | **0.4040** | 15879.71 | 52.21 | 0.7253 |
| | $QP_{train}$ | **0.8240** | 11425.21 | 39.78 | 0.6625 | **0.7318** | 13697.98 | 46.11 | 0.7055 | **0.6640** | 15673.12 | 51.92 | 0.7065 |
| Top-4 | $KL_{test}$ | 0.1010 | **7698.26** | 27.10 | 0.5438 | 0.0734 | **7652.10** | 26.69 | 0.5683 | 0.0440 | **7935.90** | 27.24 | 0.6076 |
| | $KL_{train}$ | 0.2820 | **7692.78** | 27.10 | 0.5445 | 0.1616 | **7673.86** | 26.82 | 0.5690 | 0.0810 | 8002.22 | 27.54 | 0.6011 |
| | $QP_{test}$ | **0.6130** | 12777.94 | 43.11 | 0.6665 | **0.5258** | 10638.53 | 36.63 | 0.6645 | **0.4040** | 9645.29 | 33.60 | 0.6178 |
| | $QP_{train}$ | **0.8790** | 9132.35 | 32.28 | 0.6909 | **0.8442** | 10704.41 | 36.84 | 0.6632 | **0.8040** | 9645.13 | 33.61 | 0.6188 |
| Top-3 | $KL_{test}$ | 0.2190 | **7419.83** | 25.96 | 0.5917 | 0.1906 | **7532.61** | 26.24 | 0.5883 | 0.1750 | **7703.96** | 26.57 | 0.5890 |
| | $KL_{train}$ | 0.4870 | 7422.59 | 25.99 | 0.5933 | 0.3524 | **7560.13** | 26.35 | 0.5909 | 0.2810 | **7708.60** | 26.66 | 0.5859 |
| | $QP_{test}$ | **0.6040** | 7830.06 | 27.29 | 0.6664 | **0.5762** | 8349.86 | 29.11 | 0.6174 | **0.5330** | 8505.32 | 29.35 | 0.6079 |
| | $QP_{train}$ | **0.9340** | 7169.80 | 25.63 | 0.5658 | **0.8788** | 8403.05 | 29.30 | 0.6171 | **0.8500** | 9492.00 | 33.09 | 0.6308 |
| Top-2 | $KL_{test}$ | 0.5390 | 7128.62 | 25.38 | 0.6042 | 0.4868 | 7699.19 | 26.83 | 0.6241 | 0.4420 | 8062.35 | 28.15 | 0.6419 |
| | $KL_{train}$ | 0.8030 | 7177.23 | 25.53 | 0.6035 | 0.6566 | 7747.83 | 27.00 | 0.6265 | 0.5650 | 8138.65 | 28.19 | 0.6343 |
| | $QP_{test}$ | **0.6990** | 6306.09 | 22.50 | 0.5974 | **0.6320** | 6650.82 | 23.45 | 0.5752 | **0.5790** | 6674.58 | 23.25 | 0.5708 |
| | $QP_{train}$ | **0.9470** | 6676.32 | 23.66 | 0.5703 | **0.9254** | 6690.53 | 23.59 | 0.5762 | **0.9060** | 7556.91 | 26.24 | 0.5922 |
| Top-1 | $KL_{test}$ | **0.8900** | 6641.33 | 23.32 | 0.5219 | **0.8526** | 6109.33 | 21.46 | 0.5047 | **0.7900** | 6218.27 | 21.50 | 0.4227 |
| | $KL_{train}$ | **0.9970** | 5403.39 | 19.48 | 0.5228 | **0.9936** | 6164.11 | 21.63 | 0.5036 | **0.9890** | 6274.64 | 21.68 | 0.4237 |
| | $QP_{test}$ | 0.8240 | **6154.92** | 20.90 | 0.4555 | 0.7532 | **5650.84** | 19.67 | 0.4818 | 0.7130 | **5598.51** | 19.37 | 0.3887 |
| | $QP_{train}$ | 0.9810 | **4980.71** | 17.74 | 0.5246 | 0.9688 | **5695.26** | 19.80 | 0.4816 | 0.9440 | 6359.25 | 21.77 | 0.4912 |

Table 16: Full results for DEiT-S and ViT-B in Table 3 in the main paper.

**DEiT-S**

| Protocol | Attack Method | Best ASR↑ | $\ell_1$↓ | $\ell_2$↓ | $\ell_\infty$↓ | Mean ASR↑ | $\ell_1$↓ | $\ell_2$↓ | $\ell_\infty$↓ | Worst ASR↑ | $\ell_1$↓ | $\ell_2$↓ | $\ell_\infty$↓ |
|---|---|---|---|---|---|---|---|---|---|---|---|---|---|
| Top-6 | $KL_{test}$ | 0.0370 | **7554.69** | **26.38** | **0.4735** | 0.0110 | **7884.40** | **27.78** | 0.5097 | 0.0010 | **7937.90** | 28.28 | 0.4796 |
| | $KL_{train}$ | 0.0800 | **7682.20** | **26.70** | **0.4777** | 0.0282 | **7902.49** | **27.76** | 0.5139 | 0.0060 | **7946.10** | 28.14 | **0.4714** |
| | $QP_{test}$ | **0.4910** | 20282.67 | 66.02 | 0.8132 | **0.3986** | 19017.19 | 62.38 | 0.8147 | **0.3320** | 20806.12 | 67.77 | 0.8452 |
| | $QP_{train}$ | **0.8100** | 20412.34 | 66.36 | 0.8144 | **0.6734** | 19016.97 | 62.46 | 0.8180 | **0.5960** | 20666.03 | 67.65 | 0.8526 |
| Top-5 | $KL_{test}$ | 0.0550 | **7107.77** | 24.79 | 0.5279 | 0.0328 | **7245.84** | 25.42 | 0.5014 | 0.0090 | **7181.36** | 25.27 | 0.4964 |
| | $KL_{train}$ | 0.1280 | **7240.35** | 25.80 | 0.5283 | 0.0714 | **7245.06** | 25.48 | 0.5031 | 0.0140 | **7154.82** | 25.34 | 0.5033 |
| | $QP_{test}$ | **0.6380** | 16398.45 | 54.80 | 0.7661 | **0.5770** | 16873.89 | 55.93 | 0.7629 | **0.5490** | 16274.34 | 53.26 | 0.7298 |
| | $QP_{train}$ | **0.9000** | 16384.61 | 54.83 | 0.7658 | **0.8600** | 16937.20 | 56.15 | 0.7634 | **0.8060** | 17530.56 | 58.99 | 0.8019 |
| Top-4 | $KL_{test}$ | 0.2220 | **7709.48** | 26.37 | 0.4959 | 0.1446 | **7681.20** | 26.76 | 0.5082 | 0.0540 | **7333.33** | 25.90 | 0.4734 |
| | $KL_{train}$ | 0.4170 | **7793.41** | 26.63 | 0.4990 | 0.2606 | **7717.38** | 26.88 | 0.5105 | 0.1190 | **7766.74** | 27.17 | 0.5705 |
| | $QP_{test}$ | **0.6930** | 11896.16 | 39.47 | 0.6518 | **0.6712** | 12140.38 | 40.98 | 0.6472 | **0.6320** | 13606.76 | 45.91 | 0.6304 |
| | $QP_{train}$ | **0.9540** | 11263.74 | 38.46 | 0.6589 | **0.9384** | 12220.24 | 41.22 | 0.6489 | **0.9150** | 13697.70 | 46.20 | 0.6306 |
| Top-3 | $KL_{test}$ | 0.4150 | **7385.16** | 25.97 | 0.5270 | 0.3156 | **7202.23** | 25.19 | 0.5018 | 0.2330 | **7164.11** | 25.11 | 0.4955 |
| | $KL_{train}$ | 0.6900 | **7434.29** | 26.13 | 0.5300 | 0.5194 | **7251.29** | 25.33 | 0.5040 | 0.3270 | **7221.73** | 25.28 | 0.4971 |
| | $QP_{test}$ | **0.7320** | 9604.55 | 32.99 | **0.5252** | **0.7030** | 9123.45 | 31.45 | 0.5950 | **0.6760** | 9173.88 | 31.89 | 0.6579 |
| | $QP_{train}$ | **0.9830** | 8491.79 | 28.95 | 0.6332 | **0.9526** | 9173.53 | 31.61 | 0.5979 | **0.9190** | 10105.96 | 34.59 | 0.5834 |
| Top-2 | $KL_{test}$ | 0.6230 | **6063.17** | 21.83 | 0.4454 | 0.5480 | **6082.46** | 21.44 | 0.4553 | 0.4750 | **6062.32** | 21.31 | 0.5933 |
| | $KL_{train}$ | 0.7850 | **6473.64** | 22.79 | 0.4700 | 0.7522 | **6119.41** | 21.55 | 0.4566 | 0.6720 | **6115.11** | 21.46 | 0.5974 |
| | $QP_{test}$ | **0.8170** | 8180.33 | 28.01 | 0.5119 | **0.7882** | 7959.05 | 27.24 | 0.5070 | **0.7370** | 8235.29 | 28.09 | 0.5074 |
| | $QP_{train}$ | **0.9830** | 8470.84 | 28.62 | 0.6039 | **0.9700** | 8020.04 | 27.42 | 0.5081 | **0.9650** | 8037.13 | 27.93 | 0.4882 |
| Top-1 | $KL_{test}$ | **0.9410** | 5426.09 | 19.07 | 0.3806 | **0.9268** | 5250.73 | 18.43 | 0.3864 | **0.9160** | 4674.76 | 16.71 | 0.4135 |
| | $KL_{train}$ | **0.9960** | 5632.64 | 19.94 | 0.4101 | **0.9954** | 5298.05 | 18.56 | 0.3882 | **0.9940** | 5672.40 | 19.92 | 0.4023 |
| | $QP_{test}$ | 0.9220 | **5340.78** | **17.95** | **0.2677** | 0.8984 | 5426.08 | 18.78 | 0.3371 | 0.8740 | 5448.42 | 19.34 | 0.3336 |
| | $QP_{train}$ | 0.9930 | **5409.02** | 18.92 | 0.4047 | 0.9890 | 5474.60 | 18.91 | **0.3378** | 0.9870 | **5489.21** | 19.45 | 0.3345 |

**ViT-B**

| Protocol | Attack Method | Best ASR↑ | $\ell_1$↓ | $\ell_2$↓ | $\ell_\infty$↓ | Mean ASR↑ | $\ell_1$↓ | $\ell_2$↓ | $\ell_\infty$↓ | Worst ASR↑ | $\ell_1$↓ | $\ell_2$↓ | $\ell_\infty$↓ |
|---|---|---|---|---|---|---|---|---|---|---|---|---|---|
| Top-6 | $KL_{test}$ | 0.0200 | **7426.01** | 26.81 | 0.5382 | 0.0104 | **7715.54** | 27.14 | 0.5070 | 0.0020 | 8439.95 | 29.45 | 0.5522 |
| | $KL_{train}$ | 0.0490 | **7496.11** | 26.89 | 0.5470 | 0.0256 | **7691.11** | 27.13 | 0.5092 | 0.0060 | 8257.53 | 29.00 | 0.5546 |
| | $QP_{test}$ | **0.7920** | 22361.08 | 73.51 | 0.8747 | **0.5980** | 21834.39 | 71.38 | 0.8337 | **0.4380** | 21886.20 | 71.66 | 0.8199 |
| | $QP_{train}$ | **0.9530** | 22695.33 | 74.28 | 0.8664 | **0.7764** | 22003.51 | 71.82 | 0.8317 | **0.5840** | 21893.14 | 71.70 | 0.8211 |
| Top-5 | $KL_{test}$ | 0.0820 | **7765.80** | 27.17 | 0.5143 | 0.0324 | **7587.33** | 26.60 | 0.5008 | 0.0070 | **7685.13** | 26.69 | 0.4624 |
| | $KL_{train}$ | 0.1720 | **7758.33** | 27.20 | 0.5127 | 0.0678 | **7575.71** | 26.65 | 0.5154 | 0.0150 | **7666.16** | 26.89 | 0.5264 |
| | $QP_{test}$ | **0.8300** | 19601.43 | 64.32 | 0.7475 | **0.8000** | 18771.50 | 61.81 | 0.7630 | **0.7560** | 20701.62 | 68.08 | 0.7808 |
| | $QP_{train}$ | **0.9780** | 19887.48 | 64.98 | 0.7459 | **0.9594** | 18975.86 | 62.33 | 0.7608 | **0.9300** | 20918.99 | 68.66 | 0.7791 |
| Top-4 | $KL_{test}$ | 0.1910 | **7298.67** | 25.79 | 0.4530 | 0.1210 | **7186.62** | 25.40 | 0.5047 | 0.0610 | **7383.93** | 25.95 | 0.5470 |
| | $KL_{train}$ | 0.3120 | **7209.95** | 25.74 | 0.4520 | 0.1988 | **7209.97** | 25.48 | 0.5059 | 0.0840 | **7434.38** | 26.15 | 0.5557 |
| | $QP_{test}$ | **0.8540** | 14920.53 | 49.80 | 0.6701 | **0.8258** | 15222.82 | 50.40 | 0.6790 | **0.7930** | 16756.79 | 55.47 | 0.7011 |
| | $QP_{train}$ | **0.9960** | 15109.16 | 50.28 | 0.6698 | **0.9844** | 15378.26 | 50.81 | 0.6785 | **0.9600** | 16951.09 | 56.02 | 0.7053 |
| Top-3 | $KL_{test}$ | 0.3380 | **6830.77** | 23.87 | 0.4913 | 0.2392 | **6393.40** | 22.53 | 0.4995 | 0.1680 | **6184.27** | 22.15 | 0.4857 |
| | $KL_{train}$ | 0.5160 | **6821.05** | 23.90 | 0.4939 | 0.3786 | **6411.75** | 22.59 | 0.5029 | 0.2300 | **6213.00** | 22.27 | 0.4822 |
| | $QP_{test}$ | **0.8280** | 10761.71 | 35.86 | 0.5478 | **0.7958** | 10730.18 | 36.10 | 0.5687 | **0.7470** | 10393.48 | 34.96 | 0.5189 |
| | $QP_{train}$ | **0.9870** | 10382.41 | 34.61 | 0.5879 | **0.9778** | 10814.79 | 36.33 | 0.5702 | **0.9640** | 11479.07 | 38.85 | 0.5651 |
| Top-2 | $KL_{test}$ | 0.6310 | **6390.68** | 22.27 | 0.4495 | 0.5566 | 6707.49 | 23.34 | 0.4802 | 0.3640 | 6905.25 | 23.79 | 0.5055 |
| | $KL_{train}$ | 0.8090 | **6883.84** | 23.97 | 0.4651 | 0.7038 | 6749.49 | 23.47 | 0.4822 | 0.5570 | 6940.94 | 23.91 | 0.5112 |
| | $QP_{test}$ | **0.8700** | 8503.30 | 29.21 | 0.6450 | **0.8486** | 8423.92 | 28.67 | 0.5291 | **0.8250** | 8135.73 | 27.64 | 0.5062 |
| | $QP_{train}$ | **0.9920** | 8168.44 | 27.73 | 0.5069 | **0.9802** | 8501.22 | 28.88 | 0.5316 | **0.9530** | 8690.05 | 29.08 | 0.4654 |
| Top-1 | $KL_{test}$ | 0.9690 | 6275.22 | 21.36 | **0.3308** | 0.9470 | 5775.37 | 19.88 | 0.3598 | 0.9230 | 5264.73 | 18.41 | 0.3299 |
| | $KL_{train}$ | **0.9990** | 5307.63 | 18.54 | 0.3318 | **0.9972** | 5830.60 | 20.03 | 0.3609 | **0.9950** | 6040.47 | 20.75 | 0.3667 |
| | $QP_{test}$ | 0.9740 | 6202.25 | 21.27 | 0.3790 | 0.9508 | 5846.16 | 20.02 | 0.3751 | 0.9360 | 5355.73 | 18.78 | 0.3877 |
| | $QP_{train}$ | **0.9990** | 6240.86 | 21.19 | 0.4201 | **0.9968** | 5904.46 | 20.17 | 0.3771 | **0.9950** | 5403.55 | **18.92** | 0.3893 |

Table 17: Full results for R50+D121, V-B+D-S, and R50+D121+V-B+D-S in Table 6 in the main paper.

**ResNet-50 | DenseNet121**

| Protocol | Attack Method | Best ASR↑ | $\ell_1\downarrow$ | $\ell_2\downarrow$ | $\ell_\infty\downarrow$ | Mean ASR↑ | $\ell_1\downarrow$ | $\ell_2\downarrow$ | $\ell_\infty\downarrow$ | Worst ASR↑ | $\ell_1\downarrow$ | $\ell_2\downarrow$ | $\ell_\infty\downarrow$ |
|---|---|---|---|---|---|---|---|---|---|---|---|---|---|
| Top-6 | $KL_{test}$ | 0.0010 | 9356.27 | 32.56 | **0.6355** | 0.0004 | 9461.32 | 32.97 | 0.6443 | 0.0000 | - | - | - |
| | $KL_{train}$ | 0.0010 | **9017.74** | 31.07 | **0.6355** | 0.0002 | **9017.74** | 31.07 | **0.6355** | 0.0000 | - | - | - |
| | $QP_{test}$ | **0.1210** | 17638.06 | 58.25 | 0.7922 | **0.0928** | 20118.39 | 65.07 | 0.7873 | **0.0650** | 23475.77 | 75.12 | 0.8147 |
| | $QP_{train}$ | 0.2940 | 17570.06 | 58.05 | 0.8000 | 0.2390 | 20018.87 | 64.91 | 0.7926 | 0.1830 | 23331.62 | 74.88 | 0.8219 |
| Top-5 | $KL_{test}$ | 0.0010 | 7125.80 | 25.33 | 0.7229 | 0.0002 | 7125.80 | 25.33 | 0.7229 | 0.0000 | - | - | - |
| | $KL_{train}$ | 0.0060 | **7030.86** | 24.92 | **0.6691** | 0.0018 | 7127.44 | 25.44 | **0.6366** | 0.0000 | - | - | - |
| | $QP_{test}$ | 0.2340 | 14378.07 | 48.17 | 0.7385 | 0.1738 | 15894.99 | 52.65 | 0.7600 | 0.1230 | 18130.94 | 59.36 | 0.7935 |
| | $QP_{train}$ | **0.4870** | 14256.11 | 48.10 | 0.7472 | **0.3912** | 15851.45 | 52.58 | 0.7597 | **0.3140** | 16845.02 | 55.31 | 0.7768 |
| Top-4 | $KL_{test}$ | 0.0010 | 5967.68 | 21.30 | **0.5511** | 0.0002 | 5967.68 | 21.30 | **0.5511** | 0.0000 | - | - | - |
| | $KL_{train}$ | 0.0010 | **5438.07** | 19.75 | 0.5655 | 0.0006 | **5582.88** | 19.77 | 0.5557 | 0.0000 | - | - | - |
| | $QP_{test}$ | 0.2240 | 8699.91 | 31.07 | 0.6478 | 0.1610 | 9689.51 | 33.78 | 0.6581 | 0.1230 | 11153.02 | 38.31 | 0.6791 |
| | $QP_{train}$ | **0.4620** | 8702.42 | 31.08 | 0.6505 | **0.3562** | 9675.40 | 33.77 | 0.6597 | **0.2530** | 11066.06 | 38.16 | 0.6812 |
| Top-3 | $KL_{test}$ | 0.1080 | **8405.48** | 28.78 | **0.6556** | 0.0606 | 8816.56 | 30.53 | 0.6789 | 0.0190 | 8798.46 | 30.67 | 0.6542 |
| | $KL_{train}$ | 0.2180 | 9253.28 | 32.00 | 0.6815 | 0.1152 | **8819.96** | 30.58 | 0.6754 | 0.0310 | **8705.31** | 30.52 | 0.6471 |
| | $QP_{test}$ | **0.4000** | 8553.18 | 30.55 | 0.6594 | **0.3696** | 9650.25 | 33.47 | **0.6676** | **0.3480** | 10869.45 | 37.58 | 0.7066 |
| | $QP_{train}$ | **0.7990** | 8593.43 | 30.69 | 0.6618 | **0.7006** | 9666.14 | 33.54 | **0.6702** | **0.5950** | 10919.67 | 37.77 | 0.7156 |
| Top-2 | $KL_{test}$ | 0.2490 | 6559.76 | 23.54 | 0.6312 | 0.2190 | 6753.34 | 23.76 | 0.6168 | 0.1700 | **6671.20** | 23.67 | 0.6138 |
| | $KL_{train}$ | 0.5020 | 6572.20 | 23.60 | **0.6262** | 0.4078 | 6764.51 | 23.83 | 0.6162 | 0.3520 | 6954.27 | 24.26 | 0.5883 |
| | $QP_{test}$ | 0.5220 | **6247.31** | 22.69 | 0.6281 | 0.4458 | 6692.53 | 23.87 | 0.6200 | 0.3570 | 7388.11 | 25.65 | **0.5609** |
| | $QP_{train}$ | **0.8800** | 6267.12 | 22.77 | 0.6287 | **0.7874** | 6714.57 | 23.96 | 0.6224 | **0.6850** | 7418.76 | 25.76 | **0.5606** |
| Top-1 | $KL_{test}$ | 0.8120 | 6290.36 | 22.15 | 0.6565 | 0.7918 | 7004.87 | 24.37 | 0.5542 | 0.7740 | 7426.65 | 26.04 | 0.6312 |
| | $KL_{train}$ | **0.9950** | 7282.12 | 24.80 | **0.4442** | **0.9918** | 7061.92 | 24.55 | 0.5554 | **0.9870** | 7487.19 | 26.25 | 0.6386 |
| | $QP_{test}$ | 0.8170 | 5148.40 | 18.89 | 0.5308 | 0.7944 | 5755.58 | 20.54 | 0.5232 | 0.7720 | **6271.25** | 22.25 | 0.5924 |
| | $QP_{train}$ | **0.9800** | **5580.75** | 20.44 | 0.5485 | **0.9770** | 5799.87 | 20.68 | 0.5282 | **0.9690** | 6314.22 | 22.41 | 0.6101 |

**ViT-B | DEiT-S**

| Protocol | Attack Method | Best ASR↑ | $\ell_1\downarrow$ | $\ell_2\downarrow$ | $\ell_\infty\downarrow$ | Mean ASR↑ | $\ell_1\downarrow$ | $\ell_2\downarrow$ | $\ell_\infty\downarrow$ | Worst ASR↑ | $\ell_1\downarrow$ | $\ell_2\downarrow$ | $\ell_\infty\downarrow$ |
|---|---|---|---|---|---|---|---|---|---|---|---|---|---|
| Top-6 | $KL_{test}$ | 0.0010 | 8290.64 | 28.54 | **0.5086** | 0.0002 | 8290.64 | 28.54 | **0.5086** | 0.0000 | - | - | - |
| | $KL_{train}$ | 0.0040 | **8058.14** | 28.15 | 0.6344 | 0.0008 | **8058.14** | 28.15 | 0.6344 | 0.0000 | - | - | - |
| | $QP_{test}$ | 0.2460 | 17062.54 | 55.65 | 0.7065 | 0.1868 | 18507.49 | 60.41 | 0.7371 | 0.1210 | 20619.11 | 67.31 | 0.7552 |
| | $QP_{train}$ | **0.4890** | 17157.28 | 55.90 | 0.7045 | **0.3732** | 18500.28 | 60.50 | 0.7383 | **0.2270** | 18118.13 | 59.71 | 0.7322 |
| Top-5 | $KL_{test}$ | 0.0010 | 6009.86 | 21.23 | 0.4327 | 0.0004 | 6039.43 | 21.38 | 0.4500 | 0.0000 | - | - | - |
| | $KL_{train}$ | 0.0050 | **6015.96** | 21.36 | **0.4315** | 0.0016 | 6026.06 | 21.34 | 0.4411 | 0.0000 | - | - | - |
| | $QP_{test}$ | 0.5350 | 20824.67 | 67.14 | 0.7966 | 0.4414 | 19738.61 | 64.31 | 0.7613 | 0.3070 | 22552.63 | 73.01 | 0.8024 |
| | $QP_{train}$ | **0.8080** | 20986.76 | 67.54 | 0.7942 | **0.7092** | 19845.15 | 64.62 | 0.7611 | **0.5460** | 22755.40 | 73.54 | 0.7993 |
| Top-4 | $KL_{test}$ | 0.0050 | 4648.75 | 17.03 | 0.4104 | 0.0014 | 4655.57 | 16.68 | 0.3594 | 0.0000 | - | - | - |
| | $KL_{train}$ | 0.0110 | **4525.51** | 16.78 | 0.4107 | 0.0022 | **4525.51** | 16.78 | 0.4107 | 0.0000 | - | - | - |
| | $QP_{test}$ | 0.6690 | 15054.55 | 49.36 | 0.6590 | 0.6144 | 15706.20 | 51.85 | 0.6878 | 0.5120 | 18329.80 | 60.00 | 0.7367 |
| | $QP_{train}$ | **0.9240** | 15182.73 | 49.71 | 0.6624 | **0.8900** | 15796.71 | 52.11 | 0.6889 | **0.7880** | 18372.43 | 60.20 | 0.7375 |
| Top-3 | $KL_{test}$ | 0.1430 | 7745.26 | 27.07 | 0.5694 | 0.0998 | 7399.86 | 25.81 | 0.5240 | 0.0640 | 7610.32 | 26.10 | 0.4659 |
| | $KL_{train}$ | 0.2580 | **7478.58** | 26.50 | **0.4787** | 0.1680 | 7377.14 | 25.79 | 0.5236 | 0.0970 | **6580.82** | 23.09 | 0.5219 |
| | $QP_{test}$ | 0.7130 | 10873.59 | 36.36 | 0.7110 | 0.6748 | 11564.26 | 38.93 | 0.6013 | 0.6280 | 11112.21 | 37.90 | 0.5776 |
| | $QP_{train}$ | **0.9780** | 10982.58 | 36.64 | 0.7116 | **0.9598** | 11632.23 | 39.14 | 0.6030 | **0.9400** | 13007.96 | 43.76 | 0.5835 |
| Top-2 | $KL_{test}$ | 0.4650 | 6871.62 | 23.75 | **0.4210** | 0.3856 | 6460.49 | 22.76 | 0.4715 | 0.2670 | 6423.73 | 22.16 | 0.4382 |
| | $KL_{train}$ | 0.6350 | 6884.71 | 23.82 | **0.4220** | 0.5308 | 6494.80 | 22.87 | 0.4759 | 0.3730 | 6464.91 | 22.29 | 0.4407 |
| | $QP_{test}$ | 0.8140 | 9282.50 | 31.73 | 0.5481 | 0.7746 | 8927.34 | 30.36 | 0.5374 | 0.7010 | 8824.44 | 29.74 | 0.5456 |
| | $QP_{train}$ | **0.9780** | 8881.52 | 30.04 | 0.5869 | **0.9680** | 8995.02 | 30.56 | 0.5403 | **0.9380** | 8866.49 | 29.88 | 0.5492 |
| Top-1 | $KL_{test}$ | 0.9130 | 6514.93 | 22.55 | 0.4603 | 0.9026 | 5556.07 | 19.29 | 0.3588 | 0.8850 | 5232.72 | 17.72 | **0.2970** |
| | $KL_{train}$ | **0.9910** | **5530.50** | 19.19 | **0.3271** | **0.9872** | 5605.21 | 19.42 | 0.3592 | **0.9790** | 5297.85 | 17.89 | 0.2976 |
| | $QP_{test}$ | **0.9550** | 5908.97 | 20.34 | **0.3486** | 0.9374 | 6498.39 | 22.19 | 0.3992 | 0.9100 | 6059.03 | 20.99 | 0.4103 |
| | $QP_{train}$ | **0.9990** | 6103.75 | 21.12 | 0.4159 | **0.9952** | 6565.26 | 22.37 | 0.4023 | **0.9900** | 6912.39 | 23.65 | 0.4211 |

**ResNet-50 | DenseNet121 | ViT-B | DEiT-S**

| Protocol | Attack Method | Best ASR↑ | $\ell_1\downarrow$ | $\ell_2\downarrow$ | $\ell_\infty\downarrow$ | Mean ASR↑ | $\ell_1\downarrow$ | $\ell_2\downarrow$ | $\ell_\infty\downarrow$ | Worst ASR↑ | $\ell_1\downarrow$ | $\ell_2\downarrow$ | $\ell_\infty\downarrow$ |
|---|---|---|---|---|---|---|---|---|---|---|---|---|---|
| Top-6 | $KL_{test}$ | 0.0000 | - | - | - | 0.0000 | - | - | - | **0.0000** | - | - | - |
| | $KL_{train}$ | 0.0000 | - | - | - | 0.0000 | - | - | - | **0.0000** | - | - | - |
| | $QP_{test}$ | **0.0060** | **17336.61** | 56.03 | 0.7074 | **0.0014** | **18560.54** | 59.83 | 0.6914 | **0.0000** | - | - | - |
| | $QP_{train}$ | 0.0180 | 17009.50 | 55.21 | 0.7027 | 0.0044 | 19591.40 | 63.24 | 0.6923 | **0.0000** | - | - | - |
| Top-5 | $KL_{test}$ | 0.0000 | - | - | - | 0.0000 | - | - | - | 0.0000 | - | - | - |
| | $KL_{train}$ | 0.0000 | - | - | - | 0.0000 | - | - | - | 0.0000 | - | - | - |
| | $QP_{test}$ | 0.0520 | 20731.71 | 66.35 | 0.7726 | 0.0322 | 21769.04 | 69.65 | 0.7793 | 0.0050 | 22758.10 | 72.59 | 0.8032 |
| | $QP_{train}$ | **0.1060** | 19371.15 | 62.88 | 0.7337 | **0.0604** | 21708.92 | 69.63 | 0.7698 | **0.0080** | 23377.91 | 74.21 | 0.7608 |
| Top-4 | $KL_{test}$ | 0.0010 | 8221.32 | 29.23 | 0.6635 | 0.0002 | 8221.32 | 29.23 | 0.6635 | 0.0000 | - | - | - |
| | $KL_{train}$ | 0.0000 | - | - | - | 0.0000 | - | - | - | 0.0000 | - | - | - |
| | $QP_{test}$ | **0.1540** | 16601.63 | 54.34 | 0.6983 | **0.1012** | 17423.69 | 56.58 | 0.7051 | **0.0320** | 18550.70 | 59.86 | 0.6865 |
| | $QP_{train}$ | 0.3770 | 15173.79 | 49.93 | 0.6892 | 0.2258 | 17362.30 | 56.54 | 0.7063 | 0.0810 | 18634.20 | 60.05 | 0.6941 |
| Top-3 | $KL_{test}$ | 0.0010 | 5483.35 | 19.72 | 0.5098 | 0.0002 | 5483.35 | 19.72 | 0.5098 | 0.0000 | - | - | - |
| | $KL_{train}$ | 0.0040 | **5452.43** | 19.61 | 0.4976 | 0.0010 | **5500.95** | 19.64 | 0.4636 | 0.0000 | - | - | - |
| | $QP_{test}$ | 0.2610 | 12237.89 | 41.09 | 0.6227 | 0.2102 | 12886.82 | 43.00 | 0.6612 | 0.1650 | 13393.16 | 44.74 | 0.6865 |
| | $QP_{train}$ | **0.5390** | 11825.12 | 39.82 | 0.5864 | **0.4316** | 12886.71 | 43.03 | 0.6617 | **0.3320** | 14420.31 | 48.00 | 0.6750 |
| Top-2 | $KL_{test}$ | 0.1470 | 8023.93 | 27.98 | 0.6367 | 0.1030 | 8000.37 | 28.05 | 0.6309 | 0.0610 | 7856.85 | 27.70 | 0.5808 |
| | $KL_{train}$ | 0.3120 | 8036.94 | 28.14 | 0.6382 | 0.1972 | 8053.00 | 28.24 | 0.6354 | 0.0820 | **7962.50** | 27.94 | 0.5875 |
| | $QP_{test}$ | 0.3580 | 8215.26 | 28.22 | 0.6251 | 0.3282 | 8567.16 | 29.60 | 0.5876 | 0.2760 | 8940.55 | 30.70 | 0.6406 |
| | $QP_{train}$ | **0.6600** | **7910.85** | 27.69 | 0.5018 | **0.5888** | 8584.08 | 29.66 | 0.5871 | **0.5210** | 8973.04 | 30.79 | 0.6443 |
| Top-1 | $KL_{test}$ | 0.7430 | **6616.65** | 23.57 | 0.5230 | 0.7142 | **6941.49** | 24.08 | 0.5202 | 0.6810 | **7340.90** | 25.55 | 0.5562 |
| | $KL_{train}$ | 0.9690 | 6664.76 | 23.73 | 0.5238 | 0.9494 | 6982.48 | 24.21 | 0.5223 | 0.9210 | 6886.12 | 23.29 | 0.5242 |
| | $QP_{test}$ | 0.8200 | 8509.04 | 28.54 | **0.5084** | 0.7570 | 8160.65 | 27.64 | **0.4988** | 0.6980 | 7872.09 | 26.91 | 0.5107 |
| | $QP_{train}$ | **0.9730** | 8609.30 | 28.82 | 0.5122 | **0.9538** | 8223.69 | 27.83 | **0.4996** | **0.9310** | 7930.71 | 27.07 | **0.5099** |

Table 18: Full results of the 5 ResNets in the 18-model ensemble for Table 1 in the main paper.

**ResNet-18**

| Protocol | Attack Method | Best ASR↑ | ℓ1↓ | ℓ2↓ | ℓ∞↓ | Mean ASR↑ | ℓ1↓ | ℓ2↓ | ℓ∞↓ | Worst ASR↑ | ℓ1↓ | ℓ2↓ | ℓ∞↓ |
|---|---|---|---|---|---|---|---|---|---|---|---|---|---|
| Top-3 | $KL_{test}$ | 0.4118 | 43424.02 | 136.44 | 0.9870 | 0.1076 | 16861.33 | 55.21 | 0.7161 | 0.0089 | **10485.64** | 35.72 | 0.6911 |
| | $KL_{train}$ | 0.5580 | 43999.30 | 136.90 | 0.9860 | 0.1568 | 16940.45 | 55.28 | 0.7134 | 0.0190 | **10436.65** | 35.81 | 0.6738 |
| | $QP_{test}$ | **0.9107** | 35913.60 | **114.89** | **0.9733** | **0.2525** | 15421.16 | 51.42 | **0.6866** | **0.0357** | 10679.11 | 36.95 | **0.6222** |
| | $QP_{train}$ | **0.9940** | 36669.07 | **116.07** | **0.9709** | **0.3336** | 15604.25 | 51.80 | **0.6851** | **0.0660** | 10635.25 | 36.96 | **0.6205** |
| Top-2 | $KL_{test}$ | **0.9944** | 41464.60 | 132.13 | 0.9886 | 0.3987 | 15964.81 | 53.03 | 0.7342 | 0.1763 | 9539.21 | 33.11 | **0.5806** |
| | $KL_{train}$ | **0.9970** | 42338.41 | 132.99 | 0.9856 | 0.4792 | 16181.23 | 53.36 | 0.7335 | 0.2450 | 9671.39 | 33.53 | 0.5805 |
| | $QP_{test}$ | 0.9375 | 31589.33 | **101.97** | **0.9664** | 0.4531 | 13808.98 | 46.36 | **0.6936** | 0.2467 | 9433.12 | 32.68 | 0.6909 |
| | $QP_{train}$ | 0.9960 | 32282.84 | **103.16** | **0.9627** | **0.5970** | 13975.49 | 46.71 | **0.6912** | **0.3830** | 9252.20 | 32.41 | **0.5658** |
| Top-1 | $KL_{test}$ | 0.8783 | 9570.48 | 33.29 | 0.5938 | **0.8520** | 9096.83 | 31.59 | 0.6240 | **0.8170** | 8962.19 | 31.31 | 0.6716 |
| | $KL_{train}$ | **0.9840** | 9683.44 | 33.59 | 0.5908 | **0.9644** | 9193.77 | 31.88 | 0.6258 | **0.9530** | 9038.70 | 31.57 | 0.6771 |
| | $QP_{test}$ | 0.7299 | 7066.08 | 24.73 | **0.4559** | 0.6592 | 7245.04 | 25.07 | **0.4931** | 0.5480 | 7549.05 | 25.81 | **0.4553** |
| | $QP_{train}$ | 0.8850 | 7123.41 | 24.88 | **0.4541** | 0.8254 | 7287.84 | 25.20 | **0.4931** | 0.7850 | 7281.04 | 25.23 | **0.4871** |

**ResNet-34**

| Protocol | Attack Method | Best ASR↑ | ℓ1↓ | ℓ2↓ | ℓ∞↓ | Mean ASR↑ | ℓ1↓ | ℓ2↓ | ℓ∞↓ | Worst ASR↑ | ℓ1↓ | ℓ2↓ | ℓ∞↓ |
|---|---|---|---|---|---|---|---|---|---|---|---|---|---|
| Top-3 | $KL_{test}$ | 0.0837 | **9628.84** | 32.71 | **0.6444** | 0.0507 | 16166.52 | 54.49 | 0.7231 | 0.0156 | 10261.54 | 35.12 | 0.6478 |
| | $KL_{train}$ | 0.1260 | **9719.70** | 33.05 | **0.6425** | 0.0792 | 16217.00 | 54.71 | 0.7216 | 0.0270 | 10273.97 | 35.13 | 0.6472 |
| | $QP_{test}$ | **0.9431** | 35866.28 | 114.83 | 0.9739 | **0.2911** | 15413.17 | 51.42 | **0.6879** | **0.0580** | 10499.24 | 35.72 | **0.6123** |
| | $QP_{train}$ | **0.9950** | 36657.31 | 116.05 | 0.9709 | **0.3936** | 15580.67 | 51.76 | **0.6865** | **0.1270** | 10365.32 | 35.54 | **0.6068** |
| Top-2 | $KL_{test}$ | 0.5391 | 40957.37 | 131.74 | 0.9916 | 0.3547 | 15888.77 | 53.02 | 0.7347 | 0.2310 | 9709.71 | 33.54 | 0.5817 |
| | $KL_{train}$ | 0.5670 | 41949.08 | 132.70 | 0.9872 | 0.4884 | 16135.37 | 53.38 | 0.7340 | 0.4130 | 9728.55 | 33.65 | 0.5817 |
| | $QP_{test}$ | **0.9431** | 31574.64 | **101.96** | **0.9664** | 0.4596 | 13787.04 | 46.32 | **0.6936** | 0.3181 | 9197.24 | 32.21 | **0.5644** |
| | $QP_{train}$ | **0.9980** | 32273.97 | **103.15** | **0.9627** | **0.6236** | 13968.65 | 46.69 | **0.6923** | **0.4500** | 9243.07 | 32.38 | **0.5669** |
| Top-1 | $KL_{test}$ | 0.8683 | 8676.72 | 30.18 | 0.6375 | 0.8411 | 9098.94 | 31.60 | 0.6243 | 0.8203 | 8967.47 | 31.32 | 0.6727 |
| | $KL_{train}$ | **0.9850** | 8426.15 | 29.66 | 0.6242 | **0.9670** | 9189.54 | 31.87 | 0.6261 | **0.9550** | 10008.52 | 34.03 | 0.5974 |
| | $QP_{test}$ | 0.7478 | 7074.05 | 24.84 | **0.5439** | 0.6792 | 7230.03 | 25.03 | **0.4933** | 0.5357 | 7517.93 | 25.73 | **0.4553** |
| | $QP_{train}$ | **0.9430** | 7123.41 | 25.00 | **0.5397** | 0.8582 | 7284.29 | 25.19 | **0.4933** | 0.7880 | 7566.40 | 25.90 | **0.4553** |

**ResNet-50**

| Protocol | Attack Method | Best ASR↑ | ℓ1↓ | ℓ2↓ | ℓ∞↓ | Mean ASR↑ | ℓ1↓ | ℓ2↓ | ℓ∞↓ | Worst ASR↑ | ℓ1↓ | ℓ2↓ | ℓ∞↓ |
|---|---|---|---|---|---|---|---|---|---|---|---|---|---|
| Top-3 | $KL_{test}$ | 0.2478 | 42058.93 | 135.47 | 0.9914 | 0.1096 | 16468.79 | 54.70 | 0.7204 | 0.0335 | **10177.76** | **34.90** | 0.6401 |
| | $KL_{train}$ | 0.2490 | 42807.82 | 136.05 | 0.9907 | 0.1494 | 16632.95 | 54.93 | 0.7202 | 0.0390 | **10249.42** | 35.27 | 0.6418 |
| | $QP_{test}$ | **0.9152** | 35910.64 | **114.88** | **0.9735** | **0.2797** | 15450.61 | 51.54 | **0.6847** | **0.0592** | 10459.88 | 35.72 | **0.6056** |
| | $QP_{train}$ | **0.9940** | 36664.97 | **116.06** | **0.9709** | **0.3746** | 15605.31 | 51.83 | **0.6856** | **0.1370** | 10631.63 | 37.04 | **0.6232** |
| Top-2 | $KL_{test}$ | 0.6942 | 41508.02 | 132.15 | 0.9878 | 0.4277 | 15982.01 | 53.07 | 0.7327 | 0.2422 | 9677.96 | 33.47 | 0.6934 |
| | $KL_{train}$ | 0.6730 | 42553.44 | 133.16 | 0.9846 | 0.5454 | 16216.55 | 53.38 | 0.7329 | 0.4120 | 9729.21 | 33.67 | 0.6941 |
| | $QP_{test}$ | **0.9230** | 31618.51 | **102.03** | **0.9663** | 0.4844 | 13803.22 | 46.34 | **0.6926** | 0.2835 | 9540.52 | 32.88 | **0.6575** |
| | $QP_{train}$ | **0.9920** | 32287.04 | **103.18** | **0.9627** | **0.6368** | 13965.67 | 46.67 | **0.6914** | **0.4500** | 9562.55 | 32.93 | **0.6553** |
| Top-1 | $KL_{test}$ | **0.8806** | 9898.85 | 33.71 | 0.5919 | **0.8295** | 9102.70 | 31.61 | 0.6242 | **0.7411** | 8377.73 | 29.50 | 0.6247 |
| | $KL_{train}$ | **0.9910** | 9682.78 | 33.59 | 0.5909 | **0.9746** | 9189.40 | 31.87 | 0.6260 | **0.9610** | 8429.89 | 29.67 | 0.6240 |
| | $QP_{test}$ | 0.7176 | 7215.88 | 25.04 | **0.4861** | 0.6408 | 7241.52 | 25.07 | **0.4930** | 0.5257 | 7537.87 | 25.78 | **0.4548** |
| | $QP_{train}$ | 0.8820 | 7121.04 | 25.00 | **0.5390** | 0.8496 | 7281.86 | 25.19 | **0.4932** | 0.8170 | 7557.57 | 25.87 | **0.4549** |

**ResNet-50$_2$**

| Protocol | Attack Method | Best ASR↑ | ℓ1↓ | ℓ2↓ | ℓ∞↓ | Mean ASR↑ | ℓ1↓ | ℓ2↓ | ℓ∞↓ | Worst ASR↑ | ℓ1↓ | ℓ2↓ | ℓ∞↓ |
|---|---|---|---|---|---|---|---|---|---|---|---|---|---|
| Top-3 | $KL_{test}$ | 0.4922 | 43314.78 | 136.38 | 0.9865 | 0.1888 | 16622.63 | **54.63** | 0.7257 | 0.0223 | **10241.99** | **35.11** | 0.7067 |
| | $KL_{train}$ | 0.6340 | 43989.06 | 136.86 | 0.9852 | 0.2400 | 16792.10 | 54.91 | 0.7219 | 0.0540 | **10224.57** | 35.28 | 0.6919 |
| | $QP_{test}$ | **0.8482** | 36015.86 | 115.02 | 0.9733 | 0.2214 | 16501.96 | 54.81 | **0.7077** | 0.0000 | - | - | - |
| | $QP_{train}$ | **0.9330** | 36728.70 | 116.16 | 0.9704 | **0.2752** | 16783.98 | 55.35 | **0.7016** | 0.0000 | - | - | - |
| Top-2 | $KL_{test}$ | 0.5737 | **9654.96** | 33.43 | **0.6958** | 0.3768 | 15881.10 | 52.91 | 0.7350 | 0.1373 | 9799.31 | 34.27 | 0.6715 |
| | $KL_{train}$ | 0.6230 | **9752.39** | 33.69 | **0.6926** | 0.4476 | 16112.29 | 53.25 | 0.7337 | **0.2230** | 9811.92 | 34.37 | 0.6717 |
| | $QP_{test}$ | **0.8248** | 31640.58 | 102.09 | **0.9665** | 0.3725 | **13746.04** | 46.20 | **0.6955** | 0.1283 | 9229.26 | 31.88 | **0.5870** |
| | $QP_{train}$ | **0.9180** | 32338.13 | 103.27 | **0.9620** | **0.4882** | 13928.84 | 46.56 | **0.6930** | 0.2090 | 9262.76 | 32.01 | **0.5883** |
| Top-1 | $KL_{test}$ | **0.9330** | 8326.29 | 29.36 | 0.6256 | **0.8908** | 9069.21 | 31.53 | 0.6240 | **0.8304** | 8936.87 | 31.25 | 0.6714 |
| | $KL_{train}$ | **0.9910** | 9678.14 | 33.58 | 0.5913 | **0.9700** | 9186.14 | 31.86 | 0.6259 | **0.9320** | 9025.93 | 31.53 | 0.6765 |
| | $QP_{test}$ | 0.6551 | 7044.36 | 24.77 | **0.5433** | 0.5292 | 7221.52 | 25.02 | **0.4928** | 0.3873 | 7491.44 | 25.66 | **0.4547** |
| | $QP_{train}$ | 0.7800 | 7117.37 | 24.97 | **0.5372** | 0.6464 | 7277.07 | 25.18 | **0.4923** | 0.5230 | 7541.82 | 25.84 | **0.4541** |

**ResNet-101**

| Protocol | Attack Method | Best ASR↑ | ℓ1↓ | ℓ2↓ | ℓ∞↓ | Mean ASR↑ | ℓ1↓ | ℓ2↓ | ℓ∞↓ | Worst ASR↑ | ℓ1↓ | ℓ2↓ | ℓ∞↓ |
|---|---|---|---|---|---|---|---|---|---|---|---|---|---|
| Top-3 | $KL_{test}$ | 0.4699 | 42901.18 | 135.98 | 0.9878 | 0.1489 | 16658.74 | 54.84 | 0.7196 | 0.0435 | **10249.30** | **35.07** | 0.6395 |
| | $KL_{train}$ | 0.6370 | 43734.60 | 136.65 | 0.9855 | 0.2170 | 16848.79 | 55.11 | 0.7202 | 0.0740 | **10257.94** | 35.25 | 0.6420 |
| | $QP_{test}$ | **0.9129** | 35967.69 | 114.97 | 0.9736 | **0.2839** | 15386.84 | 51.36 | **0.6869** | **0.0670** | 10394.04 | 36.29 | **0.6253** |
| | $QP_{train}$ | **0.9960** | 36650.70 | 116.04 | 0.9709 | **0.3812** | 15574.71 | 51.73 | **0.6867** | **0.1200** | 10616.62 | 36.89 | **0.6216** |
| Top-2 | $KL_{test}$ | 0.7288 | 41469.52 | 132.16 | 0.9887 | 0.4337 | 15985.99 | 53.08 | 0.7316 | 0.2779 | 9626.07 | 33.33 | 0.6935 |
| | $KL_{train}$ | 0.7680 | 42310.90 | 133.02 | 0.9865 | 0.5496 | 16192.13 | 53.40 | 0.7328 | 0.4230 | **9268.07** | 32.06 | 0.7335 |
| | $QP_{test}$ | **0.9096** | 31595.47 | **102.00** | **0.9662** | 0.4580 | 13808.78 | 46.36 | **0.6934** | 0.3080 | 9400.61 | 32.62 | **0.6910** |
| | $QP_{train}$ | **0.9980** | 32274.16 | **103.15** | **0.9627** | **0.6526** | 13961.57 | 46.67 | **0.6919** | **0.5200** | 9382.35 | 32.64 | **0.6841** |
| Top-1 | $KL_{test}$ | **0.8873** | 9546.63 | 33.23 | 0.5941 | **0.8232** | 9104.84 | 31.61 | 0.6238 | **0.7333** | 8383.35 | 29.51 | 0.6247 |
| | $KL_{train}$ | **0.9920** | 9680.22 | 33.58 | 0.5911 | **0.9644** | 9188.99 | 31.87 | 0.6261 | **0.9420** | 8791.88 | 30.51 | 0.6405 |
| | $QP_{test}$ | 0.7310 | 7221.36 | 25.05 | **0.4859** | 0.6435 | 7243.10 | 25.07 | **0.4934** | 0.5792 | 7305.36 | 24.89 | **0.5267** |
| | $QP_{train}$ | 0.8880 | 7123.64 | 25.00 | **0.5380** | 0.8292 | 7282.06 | 25.19 | **0.4933** | 0.7400 | 7332.77 | 24.97 | **0.5324** |

Table 19: Full results of the 4 DenseNets in the 18-model ensemble for Table 1 in the main paper.

**DenseNet121**

| Protocol | Attack Method | Best | | | | Mean | | | | Worst | | | |
|---|---|---|---|---|---|---|---|---|---|---|---|---|---|
| | | ASR↑ | $\ell_1$↓ | $\ell_2$↓ | $\ell_\infty$↓ | ASR↑ | $\ell_1$↓ | $\ell_2$↓ | $\ell_\infty$↓ | ASR↑ | $\ell_1$↓ | $\ell_2$↓ | $\ell_\infty$↓ |
| Top-3 | $KL_{test}$ | 0.4241 | 42980.81 | 136.04 | 0.9872 | 0.1230 | 16614.73 | 54.72 | 0.7185 | 0.0335 | **9732.79** | **33.83** | 0.6468 |
| | $KL_{train}$ | 0.5770 | 43675.11 | 136.65 | 0.9852 | 0.1854 | 16842.51 | 55.11 | 0.7179 | 0.0450 | **9980.72** | **34.57** | 0.6487 |
| | $QP_{test}$ | 0.9364 | **35888.01** | **114.86** | **0.9738** | 0.3201 | **15433.60** | **51.47** | **0.6858** | 0.0625 | 10515.34 | 35.75 | **0.6149** |
| | $QP_{train}$ | 0.9970 | 36651.15 | 116.04 | 0.9709 | 0.4320 | **15593.72** | **51.78** | **0.6870** | 0.1150 | 10617.15 | 36.92 | **0.6263** |
| Top-2 | $KL_{test}$ | 0.6518 | 41516.16 | 132.27 | 0.9891 | 0.3955 | 15944.04 | 52.98 | 0.7353 | 0.2321 | 9631.21 | 33.37 | 0.6953 |
| | $KL_{train}$ | 0.7620 | 42395.59 | 133.11 | 0.9859 | 0.4980 | 16190.08 | 53.38 | 0.7338 | 0.2650 | 9724.76 | 33.63 | 0.6936 |
| | $QP_{test}$ | 0.9118 | **31606.91** | **102.02** | **0.9660** | 0.5364 | **13760.54** | **46.24** | **0.6941** | 0.2935 | 9188.19 | 32.21 | **0.5643** |
| | $QP_{train}$ | 0.9920 | 32301.29 | 103.20 | 0.9626 | 0.6940 | 13946.59 | 46.62 | 0.6919 | 0.4400 | 9230.05 | 32.34 | **0.5643** |
| Top-1 | $KL_{test}$ | 0.8873 | 8676.47 | 30.18 | 0.6369 | 0.8645 | 9076.95 | 31.55 | 0.6244 | 0.8080 | 9897.05 | 33.71 | 0.5922 |
| | $KL_{train}$ | 0.9900 | 8425.99 | 29.66 | 0.6243 | 0.9752 | 9188.67 | 31.87 | 0.6261 | 0.9400 | 10016.17 | 34.05 | 0.5973 |
| | $QP_{test}$ | 0.7467 | **7052.40** | **24.80** | **0.5438** | 0.6643 | **7221.68** | **25.01** | **0.4932** | 0.6127 | 7053.45 | 24.71 | 0.4563 |
| | $QP_{train}$ | 0.9150 | **7118.62** | **24.99** | **0.5394** | 0.8392 | **7278.52** | **25.18** | **0.4931** | 0.7850 | 7278.90 | 25.23 | 0.4873 |

**DenseNet161**

| Protocol | Attack Method | Best | | | | Mean | | | | Worst | | | |
|---|---|---|---|---|---|---|---|---|---|---|---|---|---|
| | | ASR↑ | $\ell_1$↓ | $\ell_2$↓ | $\ell_\infty$↓ | ASR↑ | $\ell_1$↓ | $\ell_2$↓ | $\ell_\infty$↓ | ASR↑ | $\ell_1$↓ | $\ell_2$↓ | $\ell_\infty$↓ |
| Top-3 | $KL_{test}$ | 0.4487 | 42839.08 | 135.92 | 0.9880 | 0.1464 | 16530.50 | 54.54 | 0.7226 | 0.0525 | **10154.03** | **34.91** | 0.6920 |
| | $KL_{train}$ | 0.4880 | 43558.41 | 136.50 | 0.9862 | 0.1894 | 16758.92 | 54.94 | 0.7198 | 0.1040 | **10240.53** | **35.23** | 0.6406 |
| | $QP_{test}$ | **0.9297** | **35838.69** | **114.81** | **0.9739** | 0.2938 | **15362.39** | **51.32** | **0.6870** | 0.0692 | 10549.87 | 36.71 | **0.6204** |
| | $QP_{train}$ | **1.0000** | 36646.21 | 116.03 | 0.9710 | 0.4044 | **15568.96** | **51.72** | **0.6855** | 0.1590 | 10582.79 | 36.90 | **0.6250** |
| Top-2 | $KL_{test}$ | 0.9654 | 41516.79 | 132.17 | 0.9884 | 0.5056 | 15969.56 | 53.03 | 0.7331 | **0.3326** | 9843.51 | 34.37 | 0.6680 |
| | $KL_{train}$ | 0.9880 | 42350.34 | 133.01 | 0.9856 | 0.6282 | 16185.79 | 53.36 | 0.7329 | **0.5040** | 9247.14 | **32.00** | 0.7364 |
| | $QP_{test}$ | 0.9475 | **31535.84** | **101.90** | **0.9666** | 0.5107 | **13763.33** | **46.25** | **0.6937** | 0.2835 | 9175.11 | 32.15 | **0.5623** |
| | $QP_{train}$ | 0.9960 | 32277.96 | 103.16 | 0.9627 | 0.6954 | 13950.96 | 46.64 | 0.6917 | 0.4640 | 9239.06 | 32.37 | **0.5666** |
| Top-1 | $KL_{test}$ | **0.8817** | 8683.09 | 30.19 | 0.6370 | **0.8672** | 9079.22 | 31.55 | 0.6243 | **0.8571** | 8940.17 | 31.26 | 0.6728 |
| | $KL_{train}$ | **0.9910** | 9028.33 | 31.54 | 0.6779 | **0.9802** | 9186.75 | 31.86 | 0.6261 | **0.9690** | 8792.63 | 30.51 | 0.6400 |
| | $QP_{test}$ | 0.7210 | **7217.14** | **25.05** | **0.4861** | 0.6368 | **7233.16** | **25.04** | **0.4930** | 0.5714 | 7513.88 | 25.72 | 0.4552 |
| | $QP_{train}$ | 0.9000 | **7126.82** | **25.00** | **0.5389** | 0.8314 | **7281.12** | **25.18** | **0.4931** | 0.8050 | 7119.53 | 24.87 | 0.4539 |

**DenseNet169**

| Protocol | Attack Method | Best | | | | Mean | | | | Worst | | | |
|---|---|---|---|---|---|---|---|---|---|---|---|---|---|
| | | ASR↑ | $\ell_1$↓ | $\ell_2$↓ | $\ell_\infty$↓ | ASR↑ | $\ell_1$↓ | $\ell_2$↓ | $\ell_\infty$↓ | ASR↑ | $\ell_1$↓ | $\ell_2$↓ | $\ell_\infty$↓ |
| Top-3 | $KL_{test}$ | 0.6663 | 43221.14 | 136.28 | 0.9872 | 0.2304 | 16693.91 | 54.84 | 0.7211 | 0.0658 | 10254.43 | 35.10 | **0.6454** |
| | $KL_{train}$ | 0.8320 | 43823.25 | 136.75 | 0.9856 | 0.3268 | 16851.44 | 55.10 | 0.7189 | 0.0920 | **10278.86** | **35.30** | 0.6489 |
| | $QP_{test}$ | **0.9330** | **35842.76** | **114.79** | **0.9737** | 0.3181 | **15405.76** | **51.42** | **0.6853** | 0.1295 | 10088.45 | 34.99 | 0.6481 |
| | $QP_{train}$ | 0.9960 | 36655.30 | 116.05 | 0.9709 | 0.4198 | **15573.06** | **51.73** | **0.6866** | 0.2310 | 10524.42 | 36.70 | **0.6287** |
| Top-2 | $KL_{test}$ | 0.8315 | 41623.82 | 132.28 | 0.9875 | 0.4714 | 15965.24 | 52.98 | 0.7342 | 0.2634 | 9590.63 | 33.22 | 0.6957 |
| | $KL_{train}$ | 0.9310 | 42402.13 | 133.07 | 0.9849 | 0.5842 | 16199.60 | 53.37 | 0.7347 | 0.3510 | 9768.37 | 33.70 | 0.6938 |
| | $QP_{test}$ | 0.9219 | **31609.59** | **102.01** | **0.9661** | 0.5522 | **13753.34** | **46.25** | **0.6948** | 0.4118 | 9472.02 | 32.69 | **0.6623** |
| | $QP_{train}$ | 0.9960 | 32283.11 | 103.16 | 0.9627 | 0.7430 | 13940.92 | 46.61 | 0.6926 | 0.6110 | 9553.49 | 32.93 | 0.6583 |
| Top-1 | $KL_{test}$ | **0.8984** | 9544.75 | 33.23 | 0.5943 | **0.8766** | 9082.49 | 31.56 | 0.6243 | **0.8460** | 8347.16 | 29.41 | 0.6257 |
| | $KL_{train}$ | **0.9860** | 9030.65 | 31.55 | 0.6777 | **0.9786** | 9187.60 | 31.86 | 0.6261 | **0.9650** | 10010.82 | 34.04 | 0.5973 |
| | $QP_{test}$ | 0.7522 | **7048.35** | **24.69** | **0.4560** | 0.7121 | **7220.10** | **25.01** | **0.4930** | 0.6529 | 7283.95 | 24.83 | 0.5240 |
| | $QP_{train}$ | 0.8920 | **7114.45** | **24.86** | **0.4543** | 0.8614 | **7278.10** | **25.18** | **0.4933** | 0.8110 | 7273.19 | 25.22 | 0.4871 |

**DenseNet201**

| Protocol | Attack Method | Best | | | | Mean | | | | Worst | | | |
|---|---|---|---|---|---|---|---|---|---|---|---|---|---|
| | | ASR↑ | $\ell_1$↓ | $\ell_2$↓ | $\ell_\infty$↓ | ASR↑ | $\ell_1$↓ | $\ell_2$↓ | $\ell_\infty$↓ | ASR↑ | $\ell_1$↓ | $\ell_2$↓ | $\ell_\infty$↓ |
| Top-3 | $KL_{test}$ | 0.3717 | 43063.55 | 136.00 | 0.9863 | 0.1272 | 16662.74 | 54.80 | 0.7225 | 0.0391 | **10035.16** | **34.65** | 0.6429 |
| | $KL_{train}$ | 0.5150 | 43756.76 | 136.62 | 0.9853 | 0.1792 | 16804.21 | 55.01 | 0.7188 | 0.0460 | **10260.34** | **35.36** | 0.6421 |
| | $QP_{test}$ | **0.9364** | **35869.21** | **114.84** | **0.9739** | 0.3123 | **15372.93** | **51.33** | **0.6850** | 0.0714 | 10531.07 | 36.58 | **0.6184** |
| | $QP_{train}$ | 0.9950 | 36649.87 | 116.04 | 0.9710 | 0.4222 | **15542.45** | **51.65** | **0.6855** | 0.1350 | 10485.59 | 36.59 | **0.6242** |
| Top-2 | $KL_{test}$ | 0.7768 | 41568.42 | 132.19 | 0.9885 | 0.4092 | 15955.73 | 52.97 | 0.7368 | 0.2266 | 9193.70 | **31.82** | 0.7453 |
| | $KL_{train}$ | 0.8300 | 42358.44 | 133.03 | 0.9854 | 0.5152 | 16168.42 | 53.32 | 0.7364 | 0.2720 | **9234.37** | **31.98** | 0.7501 |
| | $QP_{test}$ | 0.9308 | **31575.42** | **101.96** | **0.9663** | 0.5359 | **13765.89** | **46.25** | **0.6938** | 0.3940 | 9165.40 | 32.15 | **0.5633** |
| | $QP_{train}$ | 0.9980 | 32279.66 | 103.16 | 0.9627 | 0.7068 | 13943.57 | 46.63 | 0.6912 | 0.5740 | 9560.68 | 32.96 | 0.6539 |
| Top-1 | $KL_{test}$ | **0.9062** | 9541.95 | 33.22 | 0.5942 | **0.8692** | 9080.10 | 31.56 | 0.6243 | **0.8304** | 9900.18 | 33.72 | 0.5920 |
| | $KL_{train}$ | **0.9910** | 9679.35 | 33.58 | 0.5910 | **0.9710** | 9186.99 | 31.86 | 0.6259 | **0.9440** | 10009.61 | 34.04 | 0.5972 |
| | $QP_{test}$ | 0.6920 | **7062.60** | **24.83** | **0.5430** | 0.6609 | **7231.54** | **25.04** | **0.4933** | 0.5714 | 7313.70 | 24.90 | 0.5266 |
| | $QP_{train}$ | 0.9040 | **7117.67** | **24.98** | **0.5391** | 0.8208 | **7279.18** | **25.18** | **0.4933** | 0.7400 | 7277.26 | 25.22 | 0.4871 |

Table 20: Full results of the HRNet-W18 and 3 ConvNeXts in the 18-model ensemble for Table 1 in the main paper.

**HRNet-W18**

| Protocol | Attack Method | Best | | | | Mean | | | | Worst | | | |
|---|---|---|---|---|---|---|---|---|---|---|---|---|---|
| | | ASR↑ | $\ell_1\downarrow$ | $\ell_2\downarrow$ | $\ell_\infty\downarrow$ | ASR↑ | $\ell_1\downarrow$ | $\ell_2\downarrow$ | $\ell_\infty\downarrow$ | ASR↑ | $\ell_1\downarrow$ | $\ell_2\downarrow$ | $\ell_\infty\downarrow$ |
| Top-3 | $KL_{test}$ | 0.0033 | 10420.89 | 35.56 | 0.6851 | 0.0009 | 10324.34 | 35.31 | 0.6759 | **0.0000** | - | - | - |
| | $KL_{train}$ | 0.0020 | 10514.89 | 35.80 | 0.6103 | 0.0004 | 10514.89 | 35.80 | 0.6103 | **0.0000** | - | - | - |
| | $QP_{test}$ | **0.0201** | **10109.50** | **35.04** | **0.5743** | 0.0040 | **10109.50** | **35.04** | **0.5743** | **0.0000** | - | - | - |
| | $QP_{train}$ | **0.0400** | **10308.71** | **35.52** | **0.5728** | **0.0080** | **10308.71** | **35.52** | **0.5728** | **0.0000** | - | - | - |
| Top-2 | $KL_{test}$ | 0.1138 | 9888.54 | 34.40 | 0.6686 | 0.0806 | 9584.90 | 33.22 | 0.6636 | **0.0000** | - | - | - |
| | $KL_{train}$ | 0.1840 | 9943.49 | 34.68 | 0.6675 | 0.1142 | 9700.18 | 33.57 | 0.6660 | **0.0000** | - | - | - |
| | $QP_{test}$ | **0.2946** | **9276.55** | **32.06** | **0.5867** | **0.1371** | **9367.11** | **32.45** | 0.6256 | **0.0000** | - | - | - |
| | $QP_{train}$ | **0.4330** | **9337.91** | **32.22** | **0.5878** | **0.2172** | **9417.57** | **32.64** | **0.6220** | **0.0000** | - | - | - |
| Top-1 | $KL_{test}$ | **0.8047** | 8701.76 | 30.23 | 0.6378 | **0.6556** | 9117.55 | 31.64 | 0.6227 | **0.4375** | 8985.29 | 31.34 | 0.6655 |
| | $KL_{train}$ | **0.9350** | 8429.00 | 29.66 | 0.6240 | **0.8332** | 9196.55 | 31.89 | 0.6245 | **0.6580** | 9032.92 | 31.56 | 0.6721 |
| | $QP_{test}$ | 0.5290 | **7210.77** | **25.04** | **0.4855** | 0.4464 | **7239.52** | **25.06** | **0.4920** | 0.2690 | **7507.79** | **25.71** | **0.4544** |
| | $QP_{train}$ | 0.7540 | **7135.19** | **25.02** | **0.5360** | 0.6360 | **7279.95** | **25.18** | **0.4920** | 0.4780 | **7527.42** | **25.81** | **0.4536** |

**ConvNeXt-$T$**

| Protocol | Attack Method | Best | | | | Mean | | | | Worst | | | |
|---|---|---|---|---|---|---|---|---|---|---|---|---|---|
| | | ASR↑ | $\ell_1\downarrow$ | $\ell_2\downarrow$ | $\ell_\infty\downarrow$ | ASR↑ | $\ell_1\downarrow$ | $\ell_2\downarrow$ | $\ell_\infty\downarrow$ | ASR↑ | $\ell_1\downarrow$ | $\ell_2\downarrow$ | $\ell_\infty\downarrow$ |
| Top-3 | $KL_{test}$ | 0.8471 | 42835.37 | 135.97 | 0.9884 | 0.3404 | 16593.09 | 54.73 | 0.7216 | 0.1228 | 10150.04 | 34.86 | 0.6412 |
| | $KL_{train}$ | 0.8860 | 43641.31 | 136.59 | 0.9862 | **0.4240** | 16762.49 | 54.93 | 0.7227 | **0.1830** | 10155.68 | **35.02** | 0.6448 |
| | $QP_{test}$ | **0.9944** | **35710.10** | **114.64** | **0.9748** | 0.2703 | **15350.56** | **51.25** | **0.6871** | 0.0123 | 10509.20 | 36.18 | **0.6276** |
| | $QP_{train}$ | **1.0000** | **36646.22** | **116.03** | **0.9710** | 0.3390 | **15595.78** | **51.78** | **0.6870** | 0.0220 | 10583.16 | 36.75 | **0.6345** |
| Top-2 | $KL_{test}$ | 0.8940 | 41559.50 | 132.24 | 0.9882 | **0.5768** | 15945.11 | 52.96 | 0.7336 | **0.4342** | 9770.34 | 34.17 | 0.6712 |
| | $KL_{train}$ | 0.9590 | 42359.72 | 133.03 | 0.9853 | **0.6914** | 16182.04 | 53.38 | 0.7330 | **0.4720** | 9752.06 | 33.66 | 0.6927 |
| | $QP_{test}$ | **0.9967** | **31435.69** | **101.75** | **0.9676** | 0.5109 | **13758.97** | **46.27** | **0.6936** | 0.3192 | 9226.28 | 32.30 | **0.5629** |
| | $QP_{train}$ | **1.0000** | **32269.97** | **103.14** | **0.9628** | 0.6340 | **13966.56** | **46.70** | **0.6912** | 0.4680 | 9273.99 | 32.50 | **0.5653** |
| Top-1 | $KL_{test}$ | **0.9498** | 8648.65 | 30.11 | 0.6373 | **0.9192** | 9067.07 | 31.52 | 0.6245 | **0.8661** | 9899.85 | 33.72 | 0.5923 |
| | $KL_{train}$ | **0.9930** | 8789.17 | 30.50 | 0.6403 | **0.9732** | 9188.63 | 31.87 | 0.6261 | **0.9370** | 10011.70 | 34.04 | 0.5974 |
| | $QP_{test}$ | 0.5915 | 7490.26 | 25.67 | **0.4552** | 0.4759 | **7253.81** | **25.10** | 0.4934 | 0.3058 | 7278.80 | 25.22 | 0.4871 |
| | $QP_{train}$ | 0.7640 | **7550.62** | **25.85** | **0.4547** | 0.5818 | **7305.54** | **25.27** | 0.4940 | 0.3570 | 7310.52 | 25.33 | 0.4875 |

**ConvNeXt-$S$**

| Protocol | Attack Method | Best | | | | Mean | | | | Worst | | | |
|---|---|---|---|---|---|---|---|---|---|---|---|---|---|
| | | ASR↑ | $\ell_1\downarrow$ | $\ell_2\downarrow$ | $\ell_\infty\downarrow$ | ASR↑ | $\ell_1\downarrow$ | $\ell_2\downarrow$ | $\ell_\infty\downarrow$ | ASR↑ | $\ell_1\downarrow$ | $\ell_2\downarrow$ | $\ell_\infty\downarrow$ |
| Top-3 | $KL_{test}$ | 0.8672 | 42974.20 | 136.06 | 0.9883 | 0.3214 | 16622.99 | 54.76 | 0.7200 | 0.1406 | 10168.16 | 34.92 | 0.6392 |
| | $KL_{train}$ | 0.9430 | 43641.98 | 136.60 | 0.9863 | **0.4028** | 16772.54 | 54.96 | 0.7206 | **0.2030** | 9677.32 | 32.93 | 0.6414 |
| | $QP_{test}$ | **0.9933** | **35720.24** | **114.65** | **0.9748** | 0.2408 | **15389.55** | **51.44** | **0.6849** | 0.0067 | 10758.47 | 37.26 | **0.6232** |
| | $QP_{train}$ | **1.0000** | **36646.21** | **116.03** | **0.9710** | 0.2884 | **15585.74** | **51.74** | **0.6881** | 0.0200 | 10526.73 | 36.58 | **0.6302** |
| Top-2 | $KL_{test}$ | 0.8996 | 41620.68 | 132.23 | 0.9880 | **0.5437** | 15970.53 | 52.99 | 0.7336 | **0.3973** | 9185.46 | 31.79 | 0.7333 |
| | $KL_{train}$ | 0.9330 | 42394.31 | 133.03 | 0.9856 | **0.6292** | 16198.86 | 53.38 | 0.7335 | **0.4090** | 9702.29 | 33.55 | 0.6914 |
| | $QP_{test}$ | **0.9978** | **31431.94** | **101.74** | **0.9676** | 0.4795 | **13748.97** | **46.25** | **0.6941** | 0.2589 | 9245.51 | 31.96 | **0.5854** |
| | $QP_{train}$ | **1.0000** | **32269.98** | **103.14** | **0.9628** | 0.5496 | **13970.45** | **46.71** | **0.6905** | 0.2430 | 9314.50 | 32.22 | **0.5875** |
| Top-1 | $KL_{test}$ | **0.9375** | 8918.89 | 31.21 | 0.6738 | **0.9143** | 9072.65 | 31.52 | 0.6247 | **0.8828** | 8342.73 | 29.41 | 0.6256 |
| | $KL_{train}$ | **0.9880** | 8789.57 | 30.50 | 0.6402 | **0.9732** | 9187.32 | 31.87 | 0.6260 | **0.9460** | 8430.27 | 29.67 | 0.6240 |
| | $QP_{test}$ | 0.6272 | **7075.12** | **24.77** | **0.4561** | 0.4933 | **7256.15** | **25.11** | 0.4938 | 0.3627 | 7303.93 | 25.28 | 0.4865 |
| | $QP_{train}$ | 0.7720 | **7124.11** | **24.89** | **0.4548** | 0.6142 | **7304.81** | **25.26** | 0.4941 | 0.4930 | 7337.11 | 25.00 | 0.5326 |

**ConvNeXt-$B$**

| Protocol | Attack Method | Best | | | | Mean | | | | Worst | | | |
|---|---|---|---|---|---|---|---|---|---|---|---|---|---|
| | | ASR↑ | $\ell_1\downarrow$ | $\ell_2\downarrow$ | $\ell_\infty\downarrow$ | ASR↑ | $\ell_1\downarrow$ | $\ell_2\downarrow$ | $\ell_\infty\downarrow$ | ASR↑ | $\ell_1\downarrow$ | $\ell_2\downarrow$ | $\ell_\infty\downarrow$ |
| Top-3 | $KL_{test}$ | 0.8438 | 43061.05 | 136.12 | 0.9879 | 0.3513 | 16628.36 | 54.77 | 0.7195 | 0.1641 | 10112.01 | 34.83 | 0.6391 |
| | $KL_{train}$ | 0.9350 | 43657.10 | 136.61 | 0.9859 | **0.4272** | 16762.22 | 54.95 | 0.7206 | **0.2250** | 10194.74 | 35.11 | 0.6465 |
| | $QP_{test}$ | **0.9866** | **35730.24** | **114.66** | **0.9747** | 0.2504 | **15376.51** | **51.41** | **0.6868** | 0.0335 | 10600.57 | 36.91 | **0.6233** |
| | $QP_{train}$ | **1.0000** | **36646.22** | **116.03** | **0.9710** | 0.3000 | **15568.98** | **51.75** | **0.6839** | 0.0720 | 10527.04 | 36.70 | **0.6246** |
| Top-2 | $KL_{test}$ | 0.9085 | 41744.18 | 132.31 | 0.9878 | **0.5417** | 15988.36 | 52.99 | 0.7344 | **0.3728** | 9637.31 | 33.37 | 0.6956 |
| | $KL_{train}$ | 0.9510 | 42486.12 | 133.10 | 0.9851 | **0.6558** | 16216.38 | 53.40 | 0.7326 | **0.3970** | 9746.37 | 33.67 | 0.6916 |
| | $QP_{test}$ | **0.9933** | **31442.47** | **101.76** | **0.9676** | 0.4373 | **13752.34** | **46.26** | **0.6940** | 0.2600 | 9267.04 | 32.00 | **0.5862** |
| | $QP_{train}$ | **1.0000** | **32269.98** | **103.14** | **0.9628** | 0.5028 | **13959.06** | **46.68** | **0.6910** | 0.2920 | 9297.12 | 32.16 | **0.5875** |
| Top-1 | $KL_{test}$ | **0.9520** | 9518.96 | 33.16 | 0.5942 | **0.9194** | 9062.52 | 31.51 | 0.6245 | **0.8683** | 8336.12 | 29.38 | 0.6252 |
| | $KL_{train}$ | **0.9930** | 9676.56 | 33.57 | 0.5910 | **0.9714** | 9187.14 | 31.87 | 0.6260 | **0.9300** | 8430.28 | 29.67 | 0.6241 |
| | $QP_{test}$ | 0.5346 | **7535.08** | **25.79** | **0.4551** | 0.4741 | **7253.90** | **25.11** | 0.4941 | 0.3750 | 7287.91 | 24.86 | 0.5272 |
| | $QP_{train}$ | 0.6730 | **7140.85** | **24.95** | **0.4552** | 0.6064 | **7308.35** | **25.28** | 0.4943 | 0.5170 | 7355.06 | 25.04 | 0.5341 |

Table 21: Full results of the 3 DEiTs in the 18-model ensemble for Table 1 in the main paper.

**DEiT-S**

| Protocol | Attack Method | Best ASR↑ | $\ell_1$↓ | $\ell_2$↓ | $\ell_\infty$↓ | Mean ASR↑ | $\ell_1$↓ | $\ell_2$↓ | $\ell_\infty$↓ | Worst ASR↑ | $\ell_1$↓ | $\ell_2$↓ | $\ell_\infty$↓ |
|---|---|---|---|---|---|---|---|---|---|---|---|---|---|
| Top-3 | $KL_{test}$ | 0.7489 | 43018.77 | 136.10 | 0.9882 | 0.2462 | 16668.20 | 54.85 | 0.7170 | 0.0413 | 10287.89 | 35.19 | 0.6308 |
| | $KL_{train}$ | 0.8160 | 43682.62 | 136.66 | 0.9854 | 0.3372 | 16786.58 | 55.00 | 0.7168 | 0.0820 | 10180.55 | 35.06 | 0.6283 |
| | $QP_{test}$ | 0.9888 | 35696.10 | 114.62 | 0.9748 | 0.4810 | 15310.89 | 51.23 | 0.6869 | 0.2824 | 10121.96 | 35.12 | 0.5763 |
| | $QP_{train}$ | 0.9990 | 36647.35 | 116.04 | 0.9710 | 0.6634 | 15538.81 | 51.65 | 0.6867 | 0.4810 | 10292.76 | 35.33 | 0.6083 |
| Top-2 | $KL_{test}$ | 0.9554 | 41545.30 | 132.20 | 0.9883 | 0.5225 | 15964.92 | 53.02 | 0.7336 | 0.2991 | 9163.59 | 31.76 | 0.7330 |
| | $KL_{train}$ | 0.9930 | 42348.47 | 133.00 | 0.9855 | 0.6484 | 16199.13 | 53.42 | 0.7327 | 0.3330 | 9290.64 | 32.14 | 0.7368 |
| | $QP_{test}$ | 0.9955 | 31434.24 | 101.74 | 0.9676 | 0.6839 | 13705.50 | 46.13 | 0.6948 | 0.5045 | 9199.50 | 31.86 | 0.5862 |
| | $QP_{train}$ | 1.0000 | 32269.98 | 103.14 | 0.9628 | 0.8332 | 13942.33 | 46.62 | 0.6926 | 0.6390 | 9297.81 | 32.13 | 0.5878 |
| Top-1 | $KL_{test}$ | 0.8281 | 8927.29 | 31.23 | 0.6718 | 0.7853 | 9100.77 | 31.60 | 0.6245 | 0.7254 | 9608.39 | 33.37 | 0.5957 |
| | $KL_{train}$ | 0.9530 | 8436.33 | 29.68 | 0.6242 | 0.8970 | 9197.78 | 31.89 | 0.6254 | 0.8010 | 9717.01 | 33.68 | 0.5898 |
| | $QP_{test}$ | 0.7121 | 7088.34 | 24.88 | 0.5429 | 0.6556 | 7215.05 | 25.00 | 0.4922 | 0.6094 | 7235.95 | 24.72 | 0.5202 |
| | $QP_{train}$ | 0.9390 | 7121.76 | 25.00 | 0.5392 | 0.7850 | 7282.78 | 25.19 | 0.4928 | 0.6740 | 7340.99 | 24.98 | 0.5286 |

**DeiT3-S**

| Protocol | Attack Method | Best ASR↑ | $\ell_1$↓ | $\ell_2$↓ | $\ell_\infty$↓ | Mean ASR↑ | $\ell_1$↓ | $\ell_2$↓ | $\ell_\infty$↓ | Worst ASR↑ | $\ell_1$↓ | $\ell_2$↓ | $\ell_\infty$↓ |
|---|---|---|---|---|---|---|---|---|---|---|---|---|---|
| Top-3 | $KL_{test}$ | 0.4509 | 42491.13 | 135.70 | 0.9901 | 0.1545 | 18144.89 | 59.67 | 0.7276 | 0.0000 | - | - | - |
| | $KL_{train}$ | 0.4590 | 43138.54 | 136.21 | 0.9880 | 0.1778 | 16776.99 | 55.17 | 0.6950 | 0.0010 | 10512.81 | 36.04 | 0.5725 |
| | $QP_{test}$ | 0.9978 | 35689.85 | 114.61 | 0.9749 | 0.5886 | 15301.01 | 51.19 | 0.6861 | 0.4040 | 10262.21 | 35.17 | 0.6080 |
| | $QP_{train}$ | 1.0000 | 36646.22 | 116.03 | 0.9710 | 0.7218 | 15555.13 | 51.67 | 0.6864 | 0.5520 | 10333.47 | 35.37 | 0.6076 |
| Top-2 | $KL_{test}$ | 0.9565 | 41531.36 | 132.18 | 0.9883 | 0.4857 | 16002.11 | 53.10 | 0.7324 | 0.2277 | 9963.28 | 34.69 | 0.6665 |
| | $KL_{train}$ | 0.9660 | 42394.58 | 133.05 | 0.9853 | 0.5758 | 16224.20 | 53.46 | 0.7323 | 0.3110 | 9977.45 | 34.84 | 0.6664 |
| | $QP_{test}$ | 1.0000 | 31426.25 | 101.73 | 0.9677 | 0.7417 | 13717.81 | 46.15 | 0.6943 | 0.5134 | 9368.94 | 32.50 | 0.6930 |
| | $QP_{train}$ | 1.0000 | 32269.98 | 103.14 | 0.9628 | 0.8516 | 13955.15 | 46.65 | 0.6923 | 0.6610 | 9406.71 | 32.68 | 0.6876 |
| Top-1 | $KL_{test}$ | 0.9475 | 8331.28 | 29.37 | 0.6255 | 0.9103 | 9072.65 | 31.59 | 0.6247 | 0.8761 | 8931.07 | 31.24 | 0.6738 |
| | $KL_{train}$ | 0.9710 | 10017.37 | 34.06 | 0.5974 | 0.9566 | 9193.09 | 31.88 | 0.6259 | 0.9390 | 8790.45 | 30.50 | 0.6399 |
| | $QP_{test}$ | 0.8895 | 7191.41 | 24.98 | 0.4860 | 0.8145 | 7207.28 | 24.98 | 0.4938 | 0.7243 | 7261.30 | 24.78 | 0.5262 |
| | $QP_{train}$ | 0.9390 | 7266.79 | 25.21 | 0.4873 | 0.8964 | 7279.57 | 25.18 | 0.4934 | 0.8320 | 7333.50 | 24.96 | 0.5315 |

**DeiT3-M**

| Protocol | Attack Method | Best ASR↑ | $\ell_1$↓ | $\ell_2$↓ | $\ell_\infty$↓ | Mean ASR↑ | $\ell_1$↓ | $\ell_2$↓ | $\ell_\infty$↓ | Worst ASR↑ | $\ell_1$↓ | $\ell_2$↓ | $\ell_\infty$↓ |
|---|---|---|---|---|---|---|---|---|---|---|---|---|---|
| Top-3 | $KL_{test}$ | 0.2768 | 41202.53 | 134.76 | 0.9921 | 0.0554 | 41202.53 | 134.76 | 0.9921 | 0.0000 | - | - | - |
| | $KL_{train}$ | 0.2730 | 42455.26 | 135.74 | 0.9918 | 0.0552 | 26018.27 | 84.10 | 0.8208 | 0.0000 | - | - | - |
| | $QP_{test}$ | 1.0000 | 35685.10 | 114.60 | 0.9749 | 0.2025 | 22984.78 | 75.11 | 0.7744 | 0.0000 | - | - | - |
| | $QP_{train}$ | 1.0000 | 36646.21 | 116.03 | 0.9710 | 0.2046 | 23465.52 | 75.78 | 0.7744 | 0.0000 | - | - | - |
| Top-2 | $KL_{test}$ | 0.6283 | 41371.44 | 131.94 | 0.9885 | 0.1632 | 19397.14 | 64.69 | 0.7870 | 0.0000 | - | - | - |
| | $KL_{train}$ | 0.5060 | 42469.34 | 133.03 | 0.9861 | 0.1528 | 17582.40 | 57.43 | 0.7808 | 0.0000 | - | - | - |
| | $QP_{test}$ | 1.0000 | 31426.25 | 101.73 | 0.9677 | 0.4174 | 13725.73 | 46.18 | 0.6937 | 0.1094 | 9120.65 | 32.01 | 0.5593 |
| | $QP_{train}$ | 1.0000 | 32269.98 | 103.14 | 0.9628 | 0.5188 | 14001.64 | 46.78 | 0.6913 | 0.0780 | 9406.47 | 32.82 | 0.5629 |
| Top-1 | $KL_{test}$ | 0.9275 | 8641.64 | 30.10 | 0.6375 | 0.7116 | 9090.80 | 31.59 | 0.6220 | 0.3783 | 9004.43 | 31.41 | 0.6623 |
| | $KL_{train}$ | 0.9720 | 8790.32 | 30.50 | 0.6400 | 0.8448 | 9192.51 | 31.88 | 0.6246 | 0.6360 | 9031.65 | 31.55 | 0.6726 |
| | $QP_{test}$ | 0.7087 | 7057.89 | 24.82 | 0.5449 | 0.4743 | 7204.25 | 24.98 | 0.4929 | 0.1775 | 7493.28 | 25.67 | 0.4539 |
| | $QP_{train}$ | 0.8830 | 7123.51 | 25.00 | 0.5387 | 0.6268 | 7281.08 | 25.19 | 0.4927 | 0.3610 | 7552.86 | 25.87 | 0.4532 |

Table 22: Full results of the ViT-B and MlpMixer-B in the 18-model ensemble for Table 1 in the main paper.

**ViT-B**

| Protocol | Attack Method | Best ASR↑ | $\ell_1$↓ | $\ell_2$↓ | $\ell_\infty$↓ | Mean ASR↑ | $\ell_1$↓ | $\ell_2$↓ | $\ell_\infty$↓ | Worst ASR↑ | $\ell_1$↓ | $\ell_2$↓ | $\ell_\infty$↓ |
|---|---|---|---|---|---|---|---|---|---|---|---|---|---|
| Top-3 | $KL_{test}$ | 0.8482 | 42903.41 | 136.04 | 0.9884 | 0.2192 | 16577.01 | 54.71 | 0.7212 | 0.0335 | 10017.73 | 34.68 | 0.6433 |
| | $KL_{train}$ | 0.9110 | 43652.71 | 136.62 | 0.9858 | 0.2506 | 16760.20 | 54.97 | 0.7226 | 0.0400 | 9990.42 | 34.76 | 0.6436 |
| | $QP_{test}$ | 0.9933 | 35699.80 | 114.62 | 0.9748 | 0.4991 | 15315.13 | 51.23 | 0.6870 | 0.3158 | 10496.03 | 36.54 | 0.6228 |
| | $QP_{train}$ | 0.9990 | 36645.57 | 116.03 | 0.9709 | 0.6838 | 15552.83 | 51.68 | 0.6868 | 0.5120 | 10579.38 | 36.84 | 0.6267 |
| Top-2 | $KL_{test}$ | 0.9654 | 41535.71 | 132.19 | 0.9884 | 0.3790 | 15964.40 | 53.02 | 0.7333 | 0.1562 | 9187.78 | 31.83 | 0.7315 |
| | $KL_{train}$ | 0.9910 | 42349.49 | 133.00 | 0.9855 | 0.4492 | 16196.67 | 53.40 | 0.7322 | 0.2470 | 9689.99 | 33.55 | 0.5822 |
| | $QP_{test}$ | 0.9955 | 31440.41 | 101.75 | 0.9676 | 0.6732 | 13708.89 | 46.13 | 0.6951 | 0.5536 | 9169.76 | 31.78 | 0.5866 |
| | $QP_{train}$ | 1.0000 | 32269.98 | 103.14 | 0.9628 | 0.8278 | 13949.34 | 46.64 | 0.6926 | 0.6470 | 9307.80 | 32.16 | 0.5882 |
| Top-1 | $KL_{test}$ | 0.7891 | 9578.80 | 33.30 | 0.5950 | 0.7183 | 9103.36 | 31.62 | 0.6238 | 0.5569 | 8385.41 | 29.53 | 0.6260 |
| | $KL_{train}$ | 0.9110 | 10018.33 | 34.06 | 0.5972 | 0.8658 | 9199.93 | 31.90 | 0.6251 | 0.8000 | 8802.26 | 30.55 | 0.6390 |
| | $QP_{test}$ | 0.7400 | 7219.84 | 25.05 | 0.4853 | 0.7154 | 7219.63 | 25.01 | 0.4932 | 0.6786 | 7075.21 | 24.86 | 0.5450 |
| | $QP_{train}$ | 0.9170 | 7265.84 | 25.20 | 0.4872 | 0.8814 | 7280.67 | 25.19 | 0.4931 | 0.8620 | 7115.35 | 24.86 | 0.4544 |

**MlpMixer-B**

| Protocol | Attack Method | Best ASR↑ | $\ell_1$↓ | $\ell_2$↓ | $\ell_\infty$↓ | Mean ASR↑ | $\ell_1$↓ | $\ell_2$↓ | $\ell_\infty$↓ | Worst ASR↑ | $\ell_1$↓ | $\ell_2$↓ | $\ell_\infty$↓ |
|---|---|---|---|---|---|---|---|---|---|---|---|---|---|
| Top-3 | $KL_{test}$ | 0.4643 | 43219.71 | 136.19 | 0.9875 | 0.1033 | 21074.36 | 68.15 | 0.7545 | 0.0000 | - | - | - |
| | $KL_{train}$ | 0.4520 | 44003.84 | 136.81 | 0.9851 | 0.1056 | 16910.44 | 55.26 | 0.7034 | 0.0010 | 10680.91 | 36.49 | 0.6510 |
| | $QP_{test}$ | 0.0435 | 10039.03 | 34.88 | 0.6521 | 0.0179 | 10453.45 | 36.09 | 0.6002 | 0.0000 | - | - | - |
| | $QP_{train}$ | 0.0840 | 10215.04 | 35.35 | 0.6465 | 0.0336 | 10360.78 | 35.64 | 0.6147 | 0.0000 | - | - | - |
| Top-2 | $KL_{test}$ | 0.1496 | 9983.38 | 34.78 | 0.6642 | 0.0879 | 9682.25 | 33.55 | 0.6624 | 0.0000 | - | - | - |
| | $KL_{train}$ | 0.2000 | 9925.96 | 34.75 | 0.6638 | 0.1114 | 9723.57 | 33.73 | 0.6636 | 0.0000 | - | - | - |
| | $QP_{test}$ | 0.2667 | 9607.19 | 33.05 | 0.6563 | 0.1547 | 12993.21 | 45.23 | 0.6955 | 0.0335 | 27291.52 | 95.71 | 0.9880 |
| | $QP_{train}$ | 0.3860 | 9638.89 | 33.18 | 0.6544 | 0.2100 | 13411.17 | 45.85 | 0.6931 | 0.0330 | 29372.07 | 98.47 | 0.9813 |
| Top-1 | $KL_{test}$ | 0.5658 | 9030.95 | 31.49 | 0.6698 | 0.4614 | 9182.98 | 31.84 | 0.6229 | 0.2824 | 10006.06 | 34.02 | 0.5919 |
| | $KL_{train}$ | 0.6760 | 8494.16 | 29.86 | 0.6240 | 0.5086 | 9245.34 | 32.05 | 0.6240 | 0.3640 | 9793.77 | 33.92 | 0.5873 |
| | $QP_{test}$ | 0.2768 | 7281.86 | 25.22 | 0.4848 | 0.2127 | 7299.70 | 25.23 | 0.4932 | 0.1384 | 7347.16 | 24.98 | 0.5221 |
| | $QP_{train}$ | 0.3360 | 7177.63 | 25.20 | 0.5368 | 0.2486 | 7308.03 | 25.31 | 0.4923 | 0.1040 | 7342.19 | 25.09 | 0.5304 |

Table 23: Full results of the three **unseen** testing DNNs (ConvMixer-768, SWin-B and HRNet-30) in the 18-model ensemble for Table 2 in the main paper.

**ConvMixer-768**

| Protocol | Attack Method | Best | | | | Mean | | | | Worst | | | |
|---|---|---|---|---|---|---|---|---|---|---|---|---|---|
| | | ASR↑ | $\ell_1$↓ | $\ell_2$↓ | $\ell_\infty$↓ | ASR↑ | $\ell_1$↓ | $\ell_2$↓ | $\ell_\infty$↓ | ASR↑ | $\ell_1$↓ | $\ell_2$↓ | $\ell_\infty$↓ |
| Top-3 | $KL_{test}$ | **0.0982** | 10142.63 | 34.94 | 0.6343 | **0.0339** | 21069.72 | 68.20 | 0.7551 | **0.0000** | - | - | - |
| | $KL_{train}$ | **0.1710** | **10106.77** | **34.87** | 0.6432 | **0.0510** | 20969.48 | 67.99 | 0.7570 | **0.0000** | - | - | - |
| | $QP_{test}$ | 0.0636 | **9989.84** | **34.76** | **0.5766** | 0.0150 | **18538.55** | **61.49** | 0.7428 | **0.0000** | - | - | - |
| | $QP_{train}$ | 0.1180 | 10138.39 | 35.13 | **0.5769** | 0.0300 | **19172.00** | **62.27** | 0.7386 | **0.0000** | - | - | - |
| Top-2 | $KL_{test}$ | 0.5558 | 41975.76 | 132.67 | 0.9870 | 0.2308 | 16077.85 | 53.16 | 0.7328 | **0.0960** | 9625.95 | 33.27 | **0.6898** |
| | $KL_{train}$ | 0.6660 | 42651.81 | 133.34 | 0.9834 | 0.3328 | 16202.02 | 53.33 | 0.7333 | **0.1660** | 9575.57 | 33.25 | 0.6934 |
| | $QP_{test}$ | **0.7522** | **31651.29** | **102.06** | **0.9646** | 0.2900 | **13737.18** | **46.13** | **0.6945** | 0.0625 | 9297.55 | 32.23 | 0.7035 |
| | $QP_{train}$ | 0.8310 | 32373.43 | 103.34 | 0.9613 | 0.3730 | 13958.04 | 46.63 | 0.6924 | 0.1460 | 9296.54 | 32.34 | 0.6980 |
| Top-1 | $KL_{test}$ | 0.8125 | 8333.09 | 29.38 | 0.6251 | **0.6703** | 9098.49 | 31.60 | 0.6236 | **0.4955** | 8942.69 | 31.28 | 0.6709 |
| | $KL_{train}$ | **0.9270** | 8428.17 | 29.67 | 0.6240 | **0.7852** | 9194.93 | 31.89 | 0.6244 | **0.5760** | 9026.39 | 31.56 | 0.6717 |
| | $QP_{test}$ | 0.6373 | **7037.36** | **24.75** | **0.5447** | 0.3708 | **7224.56** | **25.03** | **0.4926** | 0.2567 | **7303.90** | **24.90** | **0.5218** |
| | $QP_{train}$ | 0.7340 | **7124.40** | **25.00** | **0.5383** | 0.4524 | **7279.79** | **25.19** | **0.4919** | 0.3280 | 7525.14 | 25.82 | **0.4533** |

**SWin-B**

| Protocol | Attack Method | Best | | | | Mean | | | | Worst | | | |
|---|---|---|---|---|---|---|---|---|---|---|---|---|---|
| | | ASR↑ | $\ell_1$↓ | $\ell_2$↓ | $\ell_\infty$↓ | ASR↑ | $\ell_1$↓ | $\ell_2$↓ | $\ell_\infty$↓ | ASR↑ | $\ell_1$↓ | $\ell_2$↓ | $\ell_\infty$↓ |
| Top-3 | $KL_{test}$ | 0.0502 | 43172.79 | 136.17 | 0.9901 | 0.0228 | **16662.27** | **54.78** | 0.7137 | 0.0022 | 10015.45 | 34.49 | 0.5918 |
| | $KL_{train}$ | 0.0850 | **10115.34** | **35.01** | **0.6469** | 0.0330 | 18422.92 | 59.84 | 0.7481 | 0.0000 | - | - | - |
| | $QP_{test}$ | **0.3002** | 35995.68 | **114.79** | 0.9740 | **0.0679** | 16671.67 | 55.21 | **0.7018** | 0.0000 | - | - | - |
| | $QP_{train}$ | **0.2740** | 36935.15 | 116.20 | 0.9702 | **0.0680** | **15551.76** | **51.54** | **0.6936** | 0.0010 | 10317.88 | 35.73 | 0.6444 |
| Top-2 | $KL_{test}$ | 0.3225 | 42166.83 | 132.72 | 0.9866 | 0.1667 | 16143.42 | 53.27 | 0.7371 | 0.0748 | 9859.18 | 34.43 | 0.6736 |
| | $KL_{train}$ | 0.3350 | 42639.94 | 133.26 | 0.9852 | 0.1906 | 16308.45 | 53.61 | 0.7322 | **0.0940** | 9999.30 | 34.89 | 0.6715 |
| | $QP_{test}$ | **0.9196** | 31568.16 | 101.93 | 0.9662 | 0.2589 | 13809.34 | 46.39 | 0.6945 | 0.0826 | 9596.37 | 33.03 | 0.6614 |
| | $QP_{train}$ | 0.8830 | 32450.70 | 103.41 | 0.9611 | 0.2738 | 14038.73 | 46.90 | 0.6908 | 0.0920 | 9332.18 | 32.71 | 0.5666 |
| Top-1 | $KL_{test}$ | 0.7824 | 9930.26 | 33.79 | 0.5909 | 0.5935 | 9148.25 | 31.74 | 0.6226 | 0.3940 | 8408.84 | 29.61 | 0.6239 |
| | $KL_{train}$ | **0.8760** | 10037.36 | 34.10 | 0.5970 | **0.7098** | 9228.72 | 32.00 | 0.6246 | **0.5980** | 8463.16 | 29.78 | 0.6234 |
| | $QP_{test}$ | 0.3203 | **7280.77** | **25.25** | **0.4851** | 0.2243 | **7292.25** | **25.25** | **0.4933** | 0.1574 | **7361.81** | **25.07** | 0.5260 |
| | $QP_{train}$ | 0.3840 | **7325.08** | **25.38** | **0.4870** | 0.2416 | **7327.70** | **25.37** | **0.4931** | 0.1360 | 7565.33 | 25.97 | **0.4523** |

**HRNet-W30**

| Protocol | Attack Method | Best | | | | Mean | | | | Worst | | | |
|---|---|---|---|---|---|---|---|---|---|---|---|---|---|
| | | ASR↑ | $\ell_1$↓ | $\ell_2$↓ | $\ell_\infty$↓ | ASR↑ | $\ell_1$↓ | $\ell_2$↓ | $\ell_\infty$↓ | ASR↑ | $\ell_1$↓ | $\ell_2$↓ | $\ell_\infty$↓ |
| Top-3 | $KL_{test}$ | **0.0045** | 10276.44 | 35.28 | 0.6617 | **0.0016** | 10128.63 | 34.55 | 0.6938 | **0.0000** | - | - | - |
| | $KL_{train}$ | 0.0050 | 10607.65 | 36.15 | 0.6607 | 0.0018 | 10322.33 | 35.45 | 0.6612 | **0.0000** | - | - | - |
| | $QP_{test}$ | 0.0022 | **9844.63** | **34.10** | **0.5799** | 0.0004 | **9844.63** | **34.10** | **0.5799** | **0.0000** | - | - | - |
| | $QP_{train}$ | 0.0130 | 10204.23 | 35.18 | **0.5773** | 0.0026 | 10204.23 | 35.18 | **0.5773** | **0.0000** | - | - | - |
| Top-2 | $KL_{test}$ | **0.1931** | 9725.59 | 33.59 | **0.5821** | 0.0654 | 9487.07 | 32.90 | 0.6591 | **0.0000** | - | - | - |
| | $KL_{train}$ | **0.2400** | 9821.72 | 33.90 | **0.5811** | 0.0834 | 9616.60 | 33.37 | 0.6671 | **0.0000** | - | - | - |
| | $QP_{test}$ | 0.1518 | **9370.52** | **32.28** | 0.5868 | 0.0942 | **9409.03** | **32.59** | **0.6227** | **0.0000** | - | - | - |
| | $QP_{train}$ | 0.2270 | **9401.19** | **32.36** | 0.5875 | 0.1526 | **9428.82** | **32.68** | **0.6223** | **0.0000** | - | - | - |
| Top-1 | $KL_{test}$ | **0.8359** | 8678.02 | 30.18 | 0.6372 | **0.6362** | 9124.14 | 31.67 | 0.6228 | **0.4319** | 9639.00 | 33.48 | 0.5927 |
| | $KL_{train}$ | **0.9210** | 8791.17 | 30.50 | 0.6400 | **0.8126** | 9195.47 | 31.89 | 0.6245 | **0.6770** | 9026.14 | 31.55 | 0.6731 |
| | $QP_{test}$ | 0.6183 | **7085.78** | **24.88** | **0.5452** | 0.4109 | **7241.31** | **25.07** | **0.4934** | 0.2444 | **7511.37** | **25.72** | **0.4540** |
| | $QP_{train}$ | 0.7750 | **7129.44** | **25.02** | **0.5376** | 0.5774 | **7278.13** | **25.18** | **0.4923** | 0.4090 | 7525.87 | 25.81 | **0.4528** |

Table 24: Full results of the three **unseen** testing DNNs *at the foundation model level* (ConvNeXtV2-H, CLIP-ViT-B and EVA2-ViT-B) in the 18-model ensemble for Table 2 in the main paper.

**ConvNeXtV2-$H$**

| Protocol | Attack Method | Best ASR↑ | $\ell_1\downarrow$ | $\ell_2\downarrow$ | $\ell_\infty\downarrow$ | Mean ASR↑ | $\ell_1\downarrow$ | $\ell_2\downarrow$ | $\ell_\infty\downarrow$ | Worst ASR↑ | $\ell_1\downarrow$ | $\ell_2\downarrow$ | $\ell_\infty\downarrow$ |
|---|---|---|---|---|---|---|---|---|---|---|---|---|---|
| Top-3 | $KL_{test}$ | 0.0424 | **10103.21** | **34.90** | **0.6415** | 0.0100 | 17285.70 | 55.46 | 0.7209 | 0.0011 | 10372.17 | 35.24 | 0.6745 |
| | $KL_{train}$ | 0.0769 | **10209.38** | **34.89** | **0.6547** | 0.0154 | **10209.38** | **34.89** | **0.6547** | 0.0000 | - | - | - |
| | $QP_{test}$ | **0.2176** | 35394.05 | 114.45 | 0.9804 | **0.0493** | 16623.10 | 55.23 | **0.7119** | 0.0000 | - | - | - |
| | $QP_{train}$ | **0.3077** | 36324.36 | 115.92 | 0.9759 | **0.0667** | 23240.93 | 75.25 | 0.8227 | 0.0000 | - | - | - |
| Top-2 | $KL_{test}$ | 0.4464 | 41899.93 | 132.42 | 0.9865 | 0.1384 | 16086.00 | 53.17 | 0.7335 | 0.0279 | 9119.14 | 31.63 | 0.7333 |
| | $KL_{train}$ | 0.4615 | 43572.84 | 133.61 | 0.9702 | 0.1692 | 18115.51 | 58.48 | 0.7302 | 0.0000 | - | - | - |
| | $QP_{test}$ | **0.7545** | **31744.83** | **102.21** | **0.9645** | **0.1944** | **13875.24** | **46.49** | **0.6900** | 0.0045 | 9476.35 | 32.54 | **0.5723** |
| | $QP_{train}$ | **0.6154** | **31929.99** | **102.41** | **0.9563** | 0.1846 | 14904.23 | 49.87 | 0.7209 | 0.0000 | - | - | - |
| Top-1 | $KL_{test}$ | **0.2991** | 9680.39 | 33.58 | 0.5944 | **0.2330** | 9201.79 | 31.88 | 0.6194 | **0.1853** | 9038.93 | 31.48 | 0.6589 |
| | $KL_{train}$ | **0.3077** | 8578.32 | 29.97 | 0.6120 | **0.2359** | 9350.13 | 32.26 | 0.6050 | 0.1282 | 9119.61 | 31.87 | 0.5994 |
| | $QP_{test}$ | 0.1496 | **7121.98** | **24.92** | **0.4596** | 0.1009 | **7319.13** | **25.28** | **0.4935** | 0.0580 | **7390.53** | **25.10** | **0.5143** |
| | $QP_{train}$ | 0.1282 | **7467.81** | **25.73** | **0.4898** | 0.0974 | **7393.86** | **25.53** | **0.5018** | 0.0256 | **7780.77** | **26.49** | **0.4625** |

**CLIP-ViT-B**

| Protocol | Attack Method | Best ASR↑ | $\ell_1\downarrow$ | $\ell_2\downarrow$ | $\ell_\infty\downarrow$ | Mean ASR↑ | $\ell_1\downarrow$ | $\ell_2\downarrow$ | $\ell_\infty\downarrow$ | Worst ASR↑ | $\ell_1\downarrow$ | $\ell_2\downarrow$ | $\ell_\infty\downarrow$ |
|---|---|---|---|---|---|---|---|---|---|---|---|---|---|
| Top-3 | $KL_{test}$ | 0.0067 | **9927.98** | **33.63** | **0.6158** | 0.0018 | 10174.50 | 34.86 | 0.6276 | **0.0000** | - | - | - |
| | $KL_{train}$ | 0.0000 | - | - | - | 0.0000 | - | - | - | 0.0000 | - | - | - |
| | $QP_{test}$ | 0.0056 | 36719.51 | 116.23 | 0.9765 | 0.0018 | 23385.06 | 75.63 | 0.7780 | **0.0000** | - | - | - |
| | $QP_{train}$ | 0.0000 | - | - | - | 0.0000 | - | - | - | 0.0000 | - | - | - |
| Top-2 | $KL_{test}$ | 0.0379 | **9988.84** | **34.70** | **0.6646** | 0.0210 | 15886.55 | 52.98 | 0.7312 | 0.0100 | 9988.20 | 34.28 | 0.5806 |
| | $KL_{train}$ | 0.0513 | **10015.69** | **34.37** | **0.5898** | 0.0308 | **9909.85** | **34.14** | 0.6777 | 0.0000 | - | - | - |
| | $QP_{test}$ | 0.1886 | 32026.21 | 102.55 | 0.9609 | **0.0922** | **13896.60** | **46.49** | **0.6904** | 0.0145 | 9158.52 | 32.13 | **0.5633** |
| | $QP_{train}$ | 0.2564 | 32339.93 | 102.92 | 0.9484 | **0.1487** | 14002.58 | 46.81 | 0.6792 | 0.0513 | 9527.82 | 33.27 | 0.5441 |
| Top-1 | $KL_{test}$ | **0.5100** | 8724.35 | 30.30 | 0.6337 | **0.3100** | 9137.74 | 31.72 | 0.6186 | **0.1842** | 9660.73 | 33.57 | 0.5878 |
| | $KL_{train}$ | **0.7692** | 8791.55 | 30.39 | 0.6357 | **0.4872** | 9215.21 | 31.95 | 0.6184 | **0.2821** | 9754.42 | 33.84 | 0.5745 |
| | $QP_{test}$ | 0.2254 | **7100.00** | **24.93** | **0.5458** | 0.1587 | **7256.02** | **25.11** | **0.4925** | 0.0982 | **7325.37** | **24.94** | **0.5199** |
| | $QP_{train}$ | 0.3846 | **7460.55** | **25.72** | **0.4512** | 0.2615 | **7321.85** | **25.29** | **0.4943** | 0.1282 | **7491.41** | **25.86** | **0.4962** |

**EVA2-ViT-B**

| Protocol | Attack Method | Best ASR↑ | $\ell_1\downarrow$ | $\ell_2\downarrow$ | $\ell_\infty\downarrow$ | Mean ASR↑ | $\ell_1\downarrow$ | $\ell_2\downarrow$ | $\ell_\infty\downarrow$ | Worst ASR↑ | $\ell_1\downarrow$ | $\ell_2\downarrow$ | $\ell_\infty\downarrow$ |
|---|---|---|---|---|---|---|---|---|---|---|---|---|---|
| Top-3 | $KL_{test}$ | **0.0000** | - | - | - | **0.0000** | - | - | - | **0.0000** | - | - | - |
| | $KL_{train}$ | **0.0000** | - | - | - | **0.0000** | - | - | - | **0.0000** | - | - | - |
| | $QP_{test}$ | **0.0000** | - | - | - | **0.0000** | - | - | - | **0.0000** | - | - | - |
| | $QP_{train}$ | **0.0000** | - | - | - | **0.0000** | - | - | - | **0.0000** | - | - | - |
| Top-2 | $KL_{test}$ | **0.0000** | - | - | - | **0.0000** | - | - | - | **0.0000** | - | - | - |
| | $KL_{train}$ | **0.0000** | - | - | - | **0.0000** | - | - | - | **0.0000** | - | - | - |
| | $QP_{test}$ | **0.0000** | - | - | - | **0.0000** | - | - | - | **0.0000** | - | - | - |
| | $QP_{train}$ | **0.0000** | - | - | - | **0.0000** | - | - | - | **0.0000** | - | - | - |
| Top-1 | $KL_{test}$ | **0.1998** | 8762.19 | 30.40 | 0.6340 | 0.0560 | 9039.48 | 31.41 | 0.6145 | **0.0000** | - | - | - |
| | $KL_{train}$ | **0.2051** | 8883.08 | 30.57 | 0.6376 | 0.0872 | 9023.45 | 31.48 | 0.6242 | **0.0000** | - | - | - |
| | $QP_{test}$ | 0.1752 | **7279.96** | **24.85** | **0.5244** | 0.0592 | **7358.06** | **25.36** | **0.5035** | **0.0000** | - | - | - |
| | $QP_{train}$ | **0.2051** | **7388.33** | **25.56** | **0.4440** | **0.0923** | **7318.77** | **25.28** | **0.5000** | **0.0000** | - | - | - |

Table 25: Full results for the 6 models for Table 4 in the main paper.

**Swin-B**

| Protocol | Attack Method | Best ASR↑ | ℓ₁↓ | ℓ₂↓ | ℓ∞↓ | Mean ASR↑ | ℓ₁↓ | ℓ₂↓ | ℓ∞↓ | Worst ASR↑ | ℓ₁↓ | ℓ₂↓ | ℓ∞↓ |
|---|---|---|---|---|---|---|---|---|---|---|---|---|---|
| Top-3 | $QP_{test}$ | 0.9990 | 24540.19 | 79.65 | 0.8952 | 0.9974 | 25110.71 | 81.46 | 0.9118 | 0.9950 | 23007.00 | 75.03 | 0.8986 |
| | $QP_{train}$ | 1.0000 | 24536.30 | 79.64 | 0.8974 | 1.0000 | 25110.62 | 81.45 | 0.9130 | 1.0000 | 24536.30 | 79.64 | 0.8974 |
| | $KL_{test}$ | 0.9720 | 29787.00 | 96.13 | 0.9542 | 0.9350 | 32452.50 | 104.13 | 0.9700 | 0.8980 | 32640.93 | 104.66 | 0.9732 |
| | $KL_{train}$ | 0.9870 | 29820.40 | 96.16 | 0.9541 | 0.9532 | 32411.51 | 104.05 | 0.9702 | 0.9190 | 32579.29 | 104.54 | 0.9728 |
| Top-2 | $QP_{test}$ | 0.9980 | 23136.34 | 76.05 | 0.9286 | 0.9972 | 23314.75 | 76.01 | 0.9084 | 0.9960 | 22281.68 | 72.39 | 0.8869 |
| | $QP_{train}$ | 1.0000 | 22275.55 | 72.31 | 0.8848 | 0.9998 | 23296.71 | 75.97 | 0.9074 | 0.9990 | 23009.64 | 75.82 | 0.9302 |
| | $KL_{test}$ | 0.9960 | 29431.99 | 94.62 | 0.9541 | 0.9798 | 30468.20 | 98.02 | 0.9616 | 0.9350 | 32060.72 | 102.97 | 0.9681 |
| | $KL_{train}$ | 1.0000 | 31233.87 | 100.59 | 0.9666 | 0.9858 | 30435.74 | 97.98 | 0.9615 | 0.9450 | 31977.27 | 102.83 | 0.9678 |
| Top-1 | $QP_{test}$ | 1.0000 | 26403.61 | 85.39 | 0.9125 | 0.9994 | 27434.66 | 88.55 | 0.9256 | 0.9971 | 28208.06 | 91.11 | 0.9412 |
| | $QP_{train}$ | 1.0000 | 26340.53 | 85.33 | 0.9138 | 1.0000 | 27416.17 | 88.56 | 0.9255 | 1.0000 | 26340.53 | 85.33 | 0.9138 |
| | $KL_{test}$ | 1.0000 | 23104.45 | 75.70 | 0.8816 | 0.9996 | 22164.92 | 72.41 | 0.8749 | 0.9990 | 19679.64 | 65.03 | 0.8373 |
| | $KL_{train}$ | 1.0000 | 19694.63 | 65.03 | 0.8362 | 1.0000 | 22156.36 | 72.39 | 0.8744 | 1.0000 | 19694.63 | 65.03 | 0.8362 |

**HRNet-W30**

| Protocol | Attack Method | Best ASR↑ | ℓ₁↓ | ℓ₂↓ | ℓ∞↓ | Mean ASR↑ | ℓ₁↓ | ℓ₂↓ | ℓ∞↓ | Worst ASR↑ | ℓ₁↓ | ℓ₂↓ | ℓ∞↓ |
|---|---|---|---|---|---|---|---|---|---|---|---|---|---|
| Top-3 | $QP_{test}$ | 0.9880 | 23168.02 | 75.45 | 0.9293 | 0.9810 | 21138.58 | 69.36 | 0.9316 | 0.9660 | 21865.56 | 71.53 | 0.9362 |
| | $QP_{train}$ | 0.9990 | 21950.48 | 72.15 | 0.9240 | 0.9978 | 21116.67 | 69.33 | 0.9324 | 0.9970 | 19429.58 | 64.21 | 0.9302 |
| | $KL_{test}$ | 0.7380 | 28115.32 | 90.56 | 0.9676 | 0.6282 | 27281.20 | 87.64 | 0.9562 | 0.5220 | 26862.44 | 86.68 | 0.9547 |
| | $KL_{train}$ | 0.8300 | 28205.96 | 90.70 | 0.9675 | 0.7888 | 27313.62 | 87.69 | 0.9580 | 0.7180 | 27603.06 | 88.17 | 0.9603 |
| Top-2 | $QP_{test}$ | 0.9912 | 19796.01 | 64.54 | 0.9136 | 0.9851 | 19159.13 | 62.43 | 0.8903 | 0.9722 | 15574.23 | 51.32 | 0.8651 |
| | $QP_{train}$ | 1.0000 | 15481.31 | 51.16 | 0.8700 | 0.9996 | 19112.94 | 62.38 | 0.8916 | 0.9979 | 21513.55 | 69.81 | 0.8845 |
| | $KL_{test}$ | 0.9210 | 24272.99 | 78.93 | 0.9472 | 0.8694 | 26127.12 | 84.02 | 0.9507 | 0.7830 | 25212.21 | 80.87 | 0.9401 |
| | $KL_{train}$ | 0.9280 | 24268.70 | 78.89 | 0.9481 | 0.8810 | 26133.35 | 84.03 | 0.9505 | 0.8210 | 25273.17 | 80.97 | 0.9393 |
| Top-1 | $QP_{test}$ | 0.9985 | 18562.94 | 60.49 | 0.8946 | 0.9953 | 18590.65 | 60.32 | 0.8541 | 0.9927 | 21124.19 | 68.02 | 0.8749 |
| | $QP_{train}$ | 1.0000 | 15454.53 | 51.04 | 0.8367 | 0.9984 | 18558.31 | 60.30 | 0.8567 | 1.0000 | 15454.53 | 51.04 | 0.8367 |
| | $KL_{test}$ | 0.9990 | 16755.41 | 55.35 | 0.8885 | 0.9984 | 16784.70 | 55.46 | 0.8765 | 0.9970 | 15941.66 | 53.22 | 0.8801 |
| | $KL_{train}$ | 1.0000 | 15956.82 | 53.27 | 0.8753 | 1.0000 | 16772.27 | 55.45 | 0.8764 | 1.0000 | 15956.82 | 53.27 | 0.8753 |

**ConvMixer-768**

| Protocol | Attack Method | Best ASR↑ | ℓ₁↓ | ℓ₂↓ | ℓ∞↓ | Mean ASR↑ | ℓ₁↓ | ℓ₂↓ | ℓ∞↓ | Worst ASR↑ | ℓ₁↓ | ℓ₂↓ | ℓ∞↓ |
|---|---|---|---|---|---|---|---|---|---|---|---|---|---|
| Top-3 | $QP_{test}$ | 0.9956 | 27666.59 | 90.09 | 0.9533 | 0.9933 | 26984.55 | 87.62 | 0.9427 | 0.9912 | 25219.01 | 82.74 | 0.9415 |
| | $QP_{train}$ | 1.0000 | 28116.82 | 90.43 | 0.9447 | 0.9990 | 26952.49 | 87.61 | 0.9431 | 0.9959 | 25199.53 | 82.72 | 0.9380 |
| | $KL_{test}$ | 0.8620 | 28086.21 | 90.09 | 0.9237 | 0.7522 | 29261.82 | 94.18 | 0.9511 | 0.6490 | 29377.40 | 94.58 | 0.9531 |
| | $KL_{train}$ | 0.9120 | 28138.44 | 90.16 | 0.9223 | 0.8282 | 29283.39 | 94.21 | 0.9513 | 0.7540 | 29345.09 | 94.49 | 0.9551 |
| Top-2 | $QP_{test}$ | 0.9971 | 21134.29 | 69.68 | 0.8719 | 0.9947 | 24096.42 | 78.38 | 0.9073 | 0.9912 | 28750.18 | 92.31 | 0.9367 |
| | $QP_{train}$ | 1.0000 | 20292.20 | 66.82 | 0.8808 | 0.9990 | 24069.73 | 78.36 | 0.9066 | 0.9959 | 26091.54 | 84.71 | 0.9406 |
| | $KL_{test}$ | 0.9540 | 26778.85 | 86.45 | 0.9436 | 0.9238 | 27857.19 | 89.67 | 0.9437 | 0.9060 | 27665.11 | 89.29 | 0.9508 |
| | $KL_{train}$ | 0.9470 | 26759.37 | 86.39 | 0.9459 | 0.9144 | 27883.61 | 89.71 | 0.9435 | 0.8870 | 30007.15 | 96.24 | 0.9579 |
| Top-1 | $QP_{test}$ | 1.0000 | 22449.09 | 72.89 | 0.8889 | 0.9985 | 21496.57 | 70.00 | 0.8644 | 0.9971 | 20309.03 | 66.53 | 0.8643 |
| | $QP_{train}$ | 1.0000 | 20157.37 | 65.77 | 0.8436 | 1.0000 | 21467.20 | 69.98 | 0.8651 | 1.0000 | 20157.37 | 65.77 | 0.8436 |
| | $KL_{test}$ | 1.0000 | 18248.97 | 59.50 | 0.8337 | 0.9994 | 17103.97 | 56.55 | 0.8173 | 0.9980 | 14218.59 | 47.78 | 0.7766 |
| | $KL_{train}$ | 1.0000 | 14223.16 | 47.77 | 0.7754 | 1.0000 | 17089.81 | 56.52 | 0.8157 | 1.0000 | 14223.16 | 47.77 | 0.7754 |

**CLIP-ViT-B**

| Protocol | Attack Method | Best ASR↑ | ℓ₁↓ | ℓ₂↓ | ℓ∞↓ | Mean ASR↑ | ℓ₁↓ | ℓ₂↓ | ℓ∞↓ | Worst ASR↑ | ℓ₁↓ | ℓ₂↓ | ℓ∞↓ |
|---|---|---|---|---|---|---|---|---|---|---|---|---|---|
| Top-3 | $QP_{test}$ | 1.0000 | 34654.59 | 114.18 | 0.9842 | 0.9997 | 35112.00 | 115.42 | 0.9842 | 0.9985 | 34728.69 | 115.56 | 0.9857 |
| | $QP_{train}$ | 1.0000 | 34578.89 | 114.16 | 0.9858 | 0.9996 | 35090.07 | 115.42 | 0.9843 | 0.9979 | 34769.59 | 115.62 | 0.9860 |
| | $KL_{test}$ | 0.9020 | 49293.40 | 154.16 | 0.9946 | 0.4080 | 46355.60 | 152.15 | 0.9967 | 0.0010 | 49052.12 | 154.53 | 1.0000 |
| | $KL_{train}$ | 0.9110 | 49259.36 | 154.20 | 0.9947 | 0.4068 | 46254.19 | 151.42 | 0.9936 | 0.0000 | - | - | - |
| Top-2 | $QP_{test}$ | 1.0000 | 26357.61 | 89.12 | 0.9614 | 0.9991 | 28446.28 | 95.32 | 0.9719 | 0.9985 | 27916.95 | 93.95 | 0.9686 |
| | $QP_{train}$ | 1.0000 | 26284.29 | 89.06 | 0.9624 | 0.9994 | 28422.88 | 95.35 | 0.9721 | 0.9990 | 27947.58 | 94.10 | 0.9701 |
| | $KL_{test}$ | 0.9980 | 44208.61 | 140.39 | 0.9918 | 0.6426 | 44164.83 | 141.67 | 0.9951 | 0.0180 | 43464.04 | 140.91 | 0.9989 |
| | $KL_{train}$ | 1.0000 | 44281.64 | 140.47 | 0.9917 | 0.6366 | 44018.60 | 141.76 | 0.9950 | 0.0090 | 42664.03 | 141.20 | 0.9991 |
| Top-1 | $QP_{test}$ | 1.0000 | 27269.24 | 88.93 | 0.9514 | 0.9992 | 26705.81 | 86.93 | 0.9342 | 0.9980 | 22323.44 | 72.22 | 0.8475 |
| | $QP_{train}$ | 1.0000 | 27282.61 | 88.96 | 0.9496 | 0.9996 | 26681.01 | 86.88 | 0.9335 | 0.9980 | 22368.83 | 72.29 | 0.8443 |
| | $KL_{test}$ | 1.0000 | 25907.21 | 85.20 | 0.9533 | 1.0000 | 26343.30 | 86.47 | 0.9446 | 1.0000 | 25907.21 | 85.20 | 0.9533 |
| | $KL_{train}$ | 1.0000 | 25945.80 | 85.28 | 0.9513 | 1.0000 | 26335.81 | 86.45 | 0.9436 | 1.0000 | 25945.80 | 85.28 | 0.9513 |

**EVA2-ViT-B**

| Protocol | Attack Method | Best ASR↑ | ℓ₁↓ | ℓ₂↓ | ℓ∞↓ | Mean ASR↑ | ℓ₁↓ | ℓ₂↓ | ℓ∞↓ | Worst ASR↑ | ℓ₁↓ | ℓ₂↓ | ℓ∞↓ |
|---|---|---|---|---|---|---|---|---|---|---|---|---|---|
| Top-3 | $QP_{test}$ | 0.9990 | 21314.92 | 69.67 | 0.9225 | 0.9954 | 19896.05 | 65.16 | 0.8793 | 0.9920 | 18538.02 | 61.01 | 0.8512 |
| | $QP_{train}$ | 1.0000 | 21309.14 | 69.66 | 0.9218 | 0.9986 | 19878.14 | 65.12 | 0.8798 | 0.9950 | 19260.00 | 62.87 | 0.8457 |
| | $KL_{test}$ | 0.8170 | 27769.76 | 88.80 | 0.9440 | 0.7714 | 30104.63 | 96.12 | 0.9593 | 0.7350 | 27555.40 | 88.37 | 0.9419 |
| | $KL_{train}$ | 0.9300 | 27676.84 | 88.59 | 0.9439 | 0.8798 | 30139.58 | 96.14 | 0.9581 | 0.8490 | 27771.35 | 88.66 | 0.9382 |
| Top-2 | $QP_{test}$ | 0.9956 | 16364.13 | 53.88 | 0.7936 | 0.9942 | 16712.75 | 54.84 | 0.8048 | 0.9927 | 16992.77 | 55.60 | 0.8109 |
| | $QP_{train}$ | 1.0000 | 16917.59 | 55.49 | 0.8146 | 0.9990 | 16678.32 | 54.81 | 0.8052 | 0.9959 | 16347.58 | 53.88 | 0.7964 |
| | $KL_{test}$ | 0.9780 | 27044.71 | 86.89 | 0.9488 | 0.9594 | 25446.25 | 81.90 | 0.9236 | 0.9370 | 24641.59 | 79.20 | 0.9185 |
| | $KL_{train}$ | 0.9910 | 24015.74 | 77.20 | 0.8814 | 0.9732 | 25446.83 | 81.89 | 0.9237 | 0.9650 | 24598.83 | 79.07 | 0.9196 |
| Top-1 | $QP_{test}$ | 1.0000 | 18853.34 | 62.10 | 0.8375 | 0.9994 | 16412.60 | 54.32 | 0.7796 | 0.9985 | 13582.94 | 44.77 | 0.6327 |
| | $QP_{train}$ | 1.0000 | 18792.10 | 62.01 | 0.8329 | 1.0000 | 16382.95 | 54.30 | 0.7805 | 1.0000 | 18792.10 | 62.01 | 0.8329 |
| | $KL_{test}$ | 0.9970 | 13449.05 | 44.75 | 0.6954 | 0.9940 | 10806.82 | 35.79 | 0.5983 | 0.9890 | 9571.74 | 31.47 | 0.4424 |
| | $KL_{train}$ | 1.0000 | 11484.26 | 37.94 | 0.5829 | 1.0000 | 10797.96 | 35.77 | 0.5969 | 1.0000 | 11484.26 | 37.94 | 0.5829 |

**ConvNeXtV2-H**

| Protocol | Attack Method | Best ASR↑ | ℓ₁↓ | ℓ₂↓ | ℓ∞↓ | Mean ASR↑ | ℓ₁↓ | ℓ₂↓ | ℓ∞↓ | Worst ASR↑ | ℓ₁↓ | ℓ₂↓ | ℓ∞↓ |
|---|---|---|---|---|---|---|---|---|---|---|---|---|---|
| Top-3 | $QP_{test}$ | 0.9980 | 24945.29 | 81.07 | 0.9354 | 0.9932 | 27163.90 | 88.36 | 0.9520 | 0.9770 | 22306.51 | 74.64 | 0.9473 |
| | $QP_{train}$ | 1.0000 | 22155.59 | 74.44 | 0.9519 | 0.9996 | 27137.74 | 88.32 | 0.9528 | 0.9990 | 24936.93 | 81.06 | 0.9366 |
| | $KL_{test}$ | 0.9670 | 31722.42 | 101.31 | 0.9534 | 0.9454 | 29654.27 | 95.36 | 0.9540 | 0.9110 | 31060.00 | 99.65 | 0.9638 |
| | $KL_{train}$ | 0.9980 | 31715.49 | 101.27 | 0.9511 | 0.9898 | 29587.92 | 95.24 | 0.9540 | 0.9740 | 31056.11 | 99.58 | 0.9621 |
| Top-2 | $QP_{test}$ | 0.9990 | 29030.98 | 93.98 | 0.9649 | 0.5974 | 27490.60 | 88.93 | 0.9452 | 0.0000 | - | - | - |
| | $QP_{train}$ | 1.0000 | 28284.06 | 90.78 | 0.9484 | 0.5998 | 27446.39 | 88.88 | 0.9456 | 0.0000 | - | - | - |
| | $KL_{test}$ | 0.9920 | 34999.71 | 110.47 | 0.9734 | 0.9548 | 27633.93 | 88.80 | 0.9306 | 0.8530 | 27379.13 | 89.63 | 0.9418 |
| | $KL_{train}$ | 1.0000 | 34976.64 | 110.32 | 0.9727 | 0.9712 | 27608.80 | 88.72 | 0.9305 | 0.8670 | 27445.30 | 89.72 | 0.9421 |
| Top-1 | $QP_{test}$ | 1.0000 | 35972.03 | 115.01 | 0.9846 | 0.9982 | 31676.12 | 101.80 | 0.9694 | 0.9927 | 27356.92 | 88.33 | 0.9538 |
| | $QP_{train}$ | 1.0000 | 27287.05 | 88.28 | 0.9556 | 1.0000 | 31669.78 | 101.83 | 0.9696 | 1.0000 | 27287.05 | 88.28 | 0.9556 |
| | $KL_{test}$ | 1.0000 | 26136.14 | 84.45 | 0.9252 | 0.8000 | 23033.74 | 74.43 | 0.8774 | 0.0000 | - | - | - |
| | $KL_{train}$ | 1.0000 | 26154.14 | 84.44 | 0.9246 | 0.8000 | 23008.02 | 74.37 | 0.8766 | 0.0000 | - | - | - |

