# OpenReview forum: "Learning Universal Adversarial Perturbations for Ordered Top-K Targeted Attacks"
_ICLR.cc/2026/Conference — Submitted to ICLR 2026_

### Official Review · Reviewer_DGam · 2025-11-01

**Soundness:** 2
**Presentation:** 2
**Contribution:** 1
**Rating:** 2
**Confidence:** 5

**Summary:**

The paper proposes a universal attack framework for ordered Top-K target, demonstrating that ordered Top-K target labels can transfer effectively across both datasets and models. Specifically, the proposed AllAttacK approach extends previous single-dimensional analyses by jointly exploring universal adversarial perturbations (UAPs) across three dimensions, including model, data, and target. Built upon two single-model-single-image ordered Top-K attack methods (i.e., Adversarial Distillation and QuadAttack), this paper generalizes these techniques to the universal attack setting. Experimental results show that AllAttacK effectively generates ordered Top-K UAP with strong transferability.

**Strengths:**

1. The proposed approach demonstrates strong model-agnostic and image-agnostic properties, enhancing the transferability of universal adversarial perturbations (UAPs).

2. The paper adopts a stochastic mini-batch and mini-model optimization strategy, effectively balancing practicality and generalization.

3. Experimental results show that AllAttack achieves notable transferability, generating perturbations that are both transferable and interpretable.

**Weaknesses:**

1.  In Section 1, the explanation of the data axis lacks clarity. The authors mention that only 1,000 ImageNet training images are used to learn the UAP, which deviates from most prior UAP studies that typically utilize much larger datasets. However, the paper does not provide a rationale for this choice, which reduces the clarity and completeness of the corresponding analysis.

2. In Section 4, most experimental results are confined to the white-box setting, with no discussion of black-box scenarios. Since UAPs are generally expected to demonstrate effectiveness in black-box attacks, the lack of corresponding evaluations limits the completeness and generality of the experimental analysis.

3. In Section 2, under the discussion of the target axis, the condition requiring the sampling list $T$ to exclude segments of any of the model predictions is introduced without sufficient explanation.

4. In Section 3, the manuscript omits the intermediate derivations connecting Equation (4) to Equations (5) and (7). Moreover, the motivation for introducing the adversarial distillation (AD) and quadratic programming (QP) methods is insufficiently discussed, and it remains unclear how the proposed approaches satisfy the stated constraints.

5. In Section 3, the paper clearly formulates an optimization problem with more restrictive constraints compared to the standard UAP setting. However, it does not discuss the associated computational cost or the practical feasibility of the proposed method.

6. In Equation (7), the symbols and definitions are unclear. Specifically, $z_{i,s}$ is introduced without a proper definition, and its relationship to $x$ is not explained.

7. In Section 3, the authors state that the method builds on two single-model, single-image ordered Top-K attack techniques. However, the paper does not provide a detailed procedure for generating UAPs. For example, it is unclear how the one-step gradient descent defined in Eq. (8) guarantees that the associated constraints are satisfied.

8. In Section 3.3, the authors adopt stochastic mini-batch and mini-model learning to reduce memory requirements. It remains unclear how these choices affect the generation process and the effectiveness of the resulting UAP.

9. In Section 4, the authors present comparisons across different models for the adversarial distillation (AD) and quadratic programming (QP) methods under varying numbers of target labels and datasets. However, the paper does not include comparisons with baseline methods, such as extending CW^{K} and RisingAttacK [1] to the UAP setting.

10. In Section 4, the number of target labels is restricted to 1–6 without explanation. Moreover, the paper does not provide a theoretical justification for why the KL method outperforms QP under the Top-1 perturbation setting.

11. In Table 9, the paper compares computational efficiency but does not include any baseline methods for reference, and it remains unclear why the runtime for K = 5 is nearly identical to that for K = 2.

[1] Adversarial Perturbations Are Formed by Iteratively Learning Linear Combinations of the Right Singular Vectors of the Adversarial Jacobian. ICML 2025.

**Questions:**

1. Most of the experimental comparisons are limited to white-box scenarios and do not address black-box settings. Given that UAPs are expected to be effective in black-box attack scenarios, could the authors clarify this limitation and provide corresponding analysis or discussion?

2. The motivation for introducing adversarial distillation (AD) and quadratic programming (QP) to generate ordered Top-K targeted UAP is insufficiently explained, and it remains unclear how the proposed approach satisfies the stated constraints through the one-step gradient descent update. Could the authors clarify these points in greater detail?

3. The authors adopt stochastic mini-batch and mini-model learning to reduce memory requirements. However, it remains unclear how these design choices affect the UAP generation process and the practical effectiveness of the resulting perturbations. Could the authors further discuss the impact of different batch sizes and model selections on UAP quality and transferability, and provide empirical sensitivity analyses or guidance for selecting these parameters?

4. The authors compare adversarial distillation (AD) and quadratic programming (QP) methods across different models under varying numbers of target labels and dataset conditions. How does the attack performance of the proposed method compare with other baseline approaches?

---

> ### Author Response · Authors · 2025-12-03
>
> Dear Reviewer DGam,
>
> Thank you for your efforts in reviewing our submission. We appreciate you for the recognition of our submission in terms of the 3 strengths.
>
> Sorry for submitting the rebuttal late, since we have been testing some modifications with very strong results obtained, which we report in the global response.
>
> We address your concerns as follows.
>
> ---
>
> ### W1: In Section 1, the explanation of the data axis lacks clarity. The authors mention that only 1,000 ImageNet training images are used to learn the UAP, which deviates from most prior UAP studies that typically utilize much larger datasets. However, the paper does not provide a rationale for this choice, which reduces the clarity and completeness of the corresponding analysis.
>
> **Ans.** We followed the settings used in the two single-image single-model attack methods (AdvDistill and QuadAttacK) on which we build our AllAttacK. Another reason is that we test many different models, and want to retain efficiency.  In the gobal response, we have increase the number of training images with 5 images per category from the ImageNet-1k train set, and use all the images in the ImageNet-1k validation set. In general, learning UAPs that are transferrable using less training data is a desirable property.
>
> ---
>
> ### W2 and Q1: In Section 4, most experimental results are confined to the white-box setting, with no discussion of black-box scenarios. Since UAPs are generally expected to demonstrate effectiveness in black-box attacks, the lack of corresponding evaluations limits the completeness and generality of the experimental analysis.
>
> **Ans.** The model transferrability evaluation can be treated as black-box attacks against the target models using  training models as surrogate models. To our knowledge, we are not aware of direct black-box, either decision based or score based, UAP training, since state-of-the-art single-image single-model direct black-box based attacks still entail a quite large number of queries to approximate the gradient information. We will clarify these in revision.
>
>
> ---
>
> ### W3: In Section 2, under the discussion of the target axis, the condition requiring the sampling list to exclude segments of any of the model predictions is introduced without sufficient explanation.
>
> **Ans.** Since we use randomly sampled targets in forming ordered top-K targets and the total number of classes is 1000 in ImageNet-1k, the sample space is $\prod_{k=1}^K (1000-k-1)$, and the chance of a sampled ordered top-K targets is the same as the prediction of a benign image is almost zero for $K>1$ in the sampled 1000 training images. In the global response, we now further explicitly handle this by excluding images whose ground-truth labels are the same as the sampled top-1 target in an ordered list of top-K targets.
>
>
> ---
>
> ### W4 and Q2: In Section 3, the manuscript omits the intermediate derivations connecting Equation (4) to Equations (5) and (7). Moreover, the motivation for introducing the adversarial distillation (AD) and quadratic programming (QP) methods is insufficiently discussed, and it remains unclear how the proposed approaches satisfy the stated constraints.
>
> **Ans.** Eqn.4 is a non-linearly constrained optimization problem, which can not be solved directly. Eqn.5 and Eqn.7 are two relaxation approaches. The counterparts, single-image and single-model formulations, are built on well-known relaxation method in learning adversarial attacks. Eqn.4 is a multi-image and multi-model extension of the constraints. We will add details in the Appendix to make these self-contained in revision.
>
>
> ---
>
> ### W5: In Section 3, the paper clearly formulates an optimization problem with more restrictive constraints compared to the standard UAP setting. However, it does not discuss the associated computational cost or the practical feasibility of the proposed method.
>
> **Ans.**  Yes, going beyond standard UAP is the main objective of the proposed work, that is to learn ordered top-K UAPs across images and model. Our experimental results show the empirical feasibility of the proposed method. The computational cost depends on the number of models and the number of images in training, and the total FLOPs of all the models. We provide a preliminary runtime analysis in the Appendix C.
>
> ---
>
> ### W6:In Equation (7), the symbols and definitions are unclear. Specifically,  is $z_{i,s}$ introduced without a proper definition, and its relationship to $x$ is not explained.
>
> **Ans.** $z_{i,s}$'s are the variables to be optimized, corresponding to the latent features anchored at the backbone embedding $f(x_i^{pertub};\theta_s)$ for each image $i$ and each training model $s$. That is we optimize $z_{i,s}$ that is close to the current backbone embdding, but enables the ordered top-K attack constraints satisfied (see the constraints in Eqn.7).

---

> > ### Author Response · Authors · 2025-12-03
> >
> > ### W7: In Section 3, the authors state that the method builds on two single-model, single-image ordered Top-K attack techniques. However, the paper does not provide a detailed procedure for generating UAPs. For example, it is unclear how the one-step gradient descent defined in Eq. (8) guarantees that the associated constraints are satisfied.
> >
> > **Ans.** The one-step gradient descent in Eq.8 can not guarantees that the assocaited constraints are satisfied, even though the optimized $z_{i,s}$ from Eqn.7 can satisfy all the constraints. This is exactly the grand challenge of learning UAPs due to the relexation of the non-solvable optimzation formulation in Eq.4 under the MAXSAT framework.
> >
> >
> > ---
> >
> > ### W8 and Q3: In Section 3.3, the authors adopt stochastic mini-batch and mini-model learning to reduce memory requirements. It remains unclear how these choices affect the generation process and the effectiveness of the resulting UAP.
> >
> > **Ans.** Stochastic mini-data-batch is a common receipt in deep learning. We introduce mini-model batch to handle the memory footprint constraints when the number of training models is large. In practice, the stochasticity of mini-model batch is actually beneficial improving the model transferrability of learned UAPs. We will clarify these in revision.
> >
> > ---
> >
> > ### W9 and Q4: In Section 4, the authors present comparisons across different models for the adversarial distillation (AD) and quadratic programming (QP) methods under varying numbers of target labels and datasets. However, the paper does not include comparisons with baseline methods, such as extending CW^{K} and RisingAttacK [1] to the UAP setting.
> >
> > **Ans.** We did not include CW^{K} (introduced in the AdvDistill paper), since it has shown worse performance than AD and QuadAttacK under the single-image and single-model setting (see experiments in the QuadAttacK paper). We did not include RisingAttacK that is the state-of-the-art single-image single model ordered top-K attack method proposed in ICML'25, since during our preliminary exploration we found that the right singular vector formed perturbation formulation seems to more image and model sensitive, and thus harder to transferrable across data and models. In our on-going work, we are trying to re-examin the RisingAttacK formulation by directly extending their SQP derviation for multi-model and mulit-image settings, which we think is better to leave for future work.
> >
> > ---
> >
> > ### W10: In Section 4, the number of target labels is restricted to 1–6 without explanation. Moreover, the paper does not provide a theoretical justification for why the KL method outperforms QP under the Top-1 perturbation setting.
> >
> > **Ans.** This is due to the sheer complexity of enforcing ordered top-K targets in learning UAPs. We tested K up to 6 since the methods will otherwise fail. Similar in spirit to the single-image and single-model countepart of the vanilla AD and QuadAttacK, the vanilla AD often failes when K is larger than 15, and QuadAttacK makes a step foward, but still often fails when K is larger than 25. The recent RisingAttacK can push K to 30. For top-1 attacks, the KL method is degenrated to one-hot target based cross-entropy, which is most direct optimization objective, while the two-phase QP formulation (learning $z$ first, followed by one-step gradient descent to the image space) becomes less effective depending on models under attack. We will clarify these in revision.
> >
> >
> > ---
> >
> > ### W11: In Table 9, the paper compares computational efficiency but does not include any baseline methods for reference, and it remains unclear why the runtime for K = 5 is nearly identical to that for K = 2.
> >
> > **Ans.** Table 9 reports runtime for a 4-model UAP learning, for which the majority of  computation cost is dominated by the forward computation of the models, the cost of the QP solver becomes negligible for reasonably close K (e.g., K=2 and K=5). We will clarify this based on our new settings reported in the global response.

---

### Official Review · Reviewer_Q89G · 2025-11-01

**Soundness:** 3
**Presentation:** 3
**Contribution:** 3
**Rating:** 6
**Confidence:** 4

**Summary:**

The paper proposes AllAttacK, a method to learn universal (image-agnostic) and model-agnostic ordered Top-K targeted perturbations on ImageNet-1k. Two formulations are instantiated:

AllKLAttacK: an unconstrained surrogate minimizing a KL divergence to a hand-crafted Top-K target distribution (adversarial distillation).

AllQuadAttacK: a QP on intermediate features (per model) enforcing linear Top-K ordering constraints, followed by a gradient step in the pixel space to update the universal δ.

Training uses stochastic mini-batches of images and models, enabling ensembles up to 18 diverse DNNs (ResNets, DenseNets, ConvNeXt, ViTs/DEiTs, MLP-Mixer) and evaluation on 27 models in total, including robust and foundation-style encoders (CLIP ViT-B, EVA-2, ConvNeXtV2-H). Using 1000 train and 1000 test images (one per class), the paper reports high ASRs for Top-1 through Top-6 targets and demonstrates double transferability (to unseen images and unseen models). See Fig. 2 for qualitative UAPs; Table 1 for 18-model ensemble ASRs; Table 2 for transfer to unseen models; Table 3–5 for single-model Top-K (K≤6), including robust models; Table 6 for small ensembles; Table 7 for PCC evidence that δ dominates logits; Fig. 5 for visual examples.

**Strengths:**

1- New attack axis: ordered Top-K universals at scale. Prior UAP work largely focused on untargeted or Top-1 targeted settings; demonstrating ordered Top-K (K up to 6) with universal δ on ImageNet-1k and across many architectures is novel and practically relevant for rank-based decision pipelines. Results in Table 3/4/6 show substantial Top-K ASRs, with QP clearly advantaged when K>1.

2- Model- and data-transferability. The study convincingly shows double transfer: δ learned on 18 models transfers to held-out models (e.g., Swin-B, ConvMixer-768, ConvNeXtV2-H) and to unseen images (see Table 2, 13, 24–25, heatmaps and norms).

3- Two complementary formulations and a practical training recipe. The KL surrogate works best for Top-1, while the QP ordering constraints dominate for Top-K; the mini-model/mini-data batching is a pragmatic contribution. Eq. (5–8) and Fig. 3 make the pipeline clear.

4- Scale & breadth. Over 500 UAPs learned; 27 models spanning CNNs, ViTs, Mixers, robust checkpoints, and CLIP/EVA-2. The breadth increases confidence in external validity. See Sec. 4, Tables 1–6, F-section tables.

5- Ordered-target semantics. The paper studies both random Top-K orders (stress-tests) and semantically coherent lists (Glove neighbors), showing the latter are easier (see Table 8 and Appx Tables 10–11), which matches intuition and informs downstream risk analyses.

6- Diagnostic evidence. The PCC analysis (Eq. 9, Table 7) suggests δ steers logits so strongly that image content becomes “negligible noise,” supporting the “data-agnostic control” interpretation.

**Weaknesses:**

1- Perceptibility / constraint regime. Although δ is described as “quasi-imperceptible,” reported ℓ₂ norms are often large (e.g., 50–150 in Table 4–5, 12, 14, 25), and ℓ∞ values approach 0.9–1.0 in some settings (see heatmaps). Without perceptual metrics (LPIPS/SSIM), PSNR, or human studies, the claim of imperceptibility is not substantiated.

2- Threat-model clarity & detectability. The paper demonstrates white-box universals with logit access and allows image-range clipping only. It does not analyze defensive preprocessing, rank-consistency detectors, or semantic-coherence checks it motivates in §1–2. Top-K-ordering attacks might be detectable by rank-stability tests—no evaluation given.

3- Evaluation protocol can inflate ASR. The Best/Mean/Worst aggregation over five randomly sampled target lists (Sec. 4) may bias results upward (especially “Best”). Per-image success distributions and statistical CIs are not reported.

4- Energy–success trade-off governance. λ is grid-searched and the paper selects the lowest-energy δ with non-negligible ASR (Appx B), but the frontier (ASR vs energy) is not systematically characterized across K, models, and ensembles—hard to judge efficiency.

5- Top-K exact-order vs set membership. The work enforces strict order; many applications care about set-based Top-K or partial orders. How sensitive is the method when evaluation relaxes to unordered Top-K or allows ties? Not explored.

6- Generalization beyond ImageNet-1k. All results are on ImageNet-1k (one image per class). No experiments on non-natural domains, larger label spaces, or multi-label/ranking tasks. The method’s reliance on class-wise sampling (and removing images when GT=t₁) slightly biases the data protocol.

7- Limited analysis against strong evaluation checklists. There is no EOT for stochastic components, no black-box query-budget study, no physical-world tests, and no transfer-from-surrogates vs white-box comparison beyond model ensembles.

8- Foundational models: partial success. Transfer to CLIP ViT-B and ConvNeXtV2-H is notable but lower for larger K and for EVA-2 (Table 2 & 24–25); the paper does not analyze why (e.g., pretraining objectives, feature spectrum, or logit scale).

**Questions:**

1- Perceptibility. For the main ASR tables (e.g., Table 1–6), can you report LPIPS/SSIM and PSNR for (δ, x+δ) over Dtest, and provide human-rated visibility at multiple viewing scales? Also show ASR vs ℓ₂ / ℓ∞ curves for K=1…6 across key models.

2- Ordered vs unordered. How do ASRs change if evaluation is unordered Top-K or allows partial order constraints (e.g., only top-m fixed, rest any order)? Please add ablations.

3- Confidence intervals. For Mean/Best/Worst protocols, report per-image success distributions and 95% CIs, and test sensitivity to the number of target-list samples (5→20).

4- QP details & optimality. In Eq. (7–8) the QP is solved on feature variables $z_{i, s}$  with linear Top-K constraints, then one gradient step updates δ. Please characterize feasibility rates of the QP per batch, KKT residuals, and show convergence (or cyclic behavior) of the alternating scheme. Any observed mode collapse to a few classes?

5- Sensitivity to λ and batch sizes. Provide sweeps of λ (and smax if used inside the KL target construction) and mini-model/mini-data batch sizes, reporting ASR–energy frontiers.

6- Feature-space choice. You set $F(x)=f(x) W+b$  and enforce order at logits. What if the linear constraints are enforced earlier (e.g., pre-classifier feature margins), or you use logit temperature scaling to control numerical conditioning—does that improve energy efficiency?

7- Why EVA-2 is harder? Provide spectral/semantic analyses (Fourier energy, token attention patterns) explaining lower transfer to EVA-2 vs CLIP/ConvNeXtV2-H; test whether in-distribution CLIP fine-tuning or logit scaling reduces the gap.

8- Cross-norm transfer. If δ is trained under ℓ₂ but evaluated under ℓ∞ (and vice versa), how do ASRs change? Include ε-sweeps like Fig.10 across norms.

9- Stronger evaluations. Add EOT (noise/resize-crop), JPEG/bit-depth compressions, and rank-stability detectors; test simple defenses (randomized smoothing, small denoisers) and non-differentiable checks (e.g., semantic-coherence filters) against your δ.

10- Black-box budgets. Can your δ be approximated from queries only (no gradients) given the same ordered Top-K targets? Report query counts to reach ASR comparable to white-box

---

> ### Author Response · Authors · 2025-12-01
>
> Dear Reviewer Q89G,
>
> Thank you for your efforts in reviewing our submission. We appreciate you for the recognition of our submission in terms of the 6 strengths.
>
> Sorry for submitting the rebuttal late, since we have been testing some modifications with very strong results obtained, which we report in the global response.
>
> We address your concerns as follows.
>
> ---
>
>
> ### W1- Perceptibility / constraint regime. Although δ is described as “quasi-imperceptible,” reported ℓ₂ norms are often large (e.g., 50–150 in Table 4–5, 12, 14, 25), and ℓ∞ values approach 0.9–1.0 in some settings (see heatmaps). Without perceptual metrics (LPIPS/SSIM), PSNR, or human studies, the claim of imperceptibility is not substantiated.
>
> **Ans.** Thank you for pointing this out. In the global response, we have refactored our code to explicitly constraining the $\ell_2$ budget (e.g., $\tau=25$) to ensure quasi-imperceptibilty. Following your suggestions, we will leverage metrics used in evaluating image generation to report different metrics on the perturbed image against the benign images in revision.
>
> ---
>
> ### W2- Threat-model clarity & detectability. The paper demonstrates white-box universals with logit access and allows image-range clipping only. It does not analyze defensive preprocessing, rank-consistency detectors, or semantic-coherence checks it motivates in §1–2. Top-K-ordering attacks might be detectable by rank-stability tests—no evaluation given.
>
> **Ans.** We provide an ablation study in Section 4.5 for rank-consistency detectors, or semantic-coherence checks. We show that when the ordered top-K targets are semantically meaningful (e.g., based on the label similarity), the attack success rates are much higher. In our main experiments, we use randomly sampled ordered top-K targets to benchmarking how deep the vulnerability of DNNs is (by randomly manipulating their decision rankings).
>
> For defensive preprocessing, we focus on how ImageNet-1k evaluation is oftne done by different pretrained models (center crop). In our current implementation, in training, we do not use any data augmentation. We tested some light-weight data augmentation in testing (left-right flipping and Gaussian blurring with a kernel size 5 and randomly sampled sigma between 0.1 and 2), the learned top-1 and top-2 UAPs can retain performance very well for left-right flipping, while retain promising performance for blurring. We report the results in the global response (see **ASR-LRFlip and ASR-Blur**).
>
> But for heavier transformations, such as rotations, color jittering cutoff and/or mixup, the learned UAPs are not effective anymore. We will summarize the results in revision.
>
> ---
>
> ### W3- Evaluation protocol can inflate ASR. The Best/Mean/Worst aggregation over five randomly sampled target lists (Sec. 4) may bias results upward (especially “Best”). Per-image success distributions and statistical CIs are not reported.
>
> **Ans.** We follow the widely used metrics (Best/Mean/Worst) in evaluating adversarial attacks. Since we use randomly sampled top-K targets, even for the Best metric, the evaluation should be still fair based on our understanding. Based on our observation in Section 4.5, when semantically coherent top-K targets are used, the attack success rates are much higher. We checked our randomly sampled top-K targets using five random seeds (as visualized in the HTML tool in the supplementary), none of them are semantically coherent. So, the reported ASRs are fair in terms of benchmarking the attack success.
>
> In revision, we will add per-image success distributions for both per-model and across-model according to the four combinations reported in the global response (train model + train data, train model + val data, test model + train data, test model + val data). For CIs, we will report the Wilson score interval and the Bootstrap CI.
>
> ---
>
> ### W4- Energy–success trade-off governance. λ is grid-searched and the paper selects the lowest-energy δ with non-negligible ASR (Appx B), but the frontier (ASR vs energy) is not systematically characterized across K, models, and ensembles—hard to judge efficiency.
>
> **Ans.** Based on the new results in the global response, we leverage the explicit $\ell_p$ projection to eliminate the need of the hyperparameter $\lambda$. We will report Energy-Success trade-off based on the energy budget $\tau$ and the type $\ell_2$ or $\ell_{inf}$ in revision.
>
> ---

---

> ### Author Response · Authors · 2025-12-01
>
> ### W5- Top-K exact-order vs set membership. The work enforces strict order; many applications care about set-based Top-K or partial orders. How sensitive is the method when evaluation relaxes to unordered Top-K or allows ties? Not explored.
>
> **Ans.** In general, the number of constraints will increase (significantly) in Eq. 4 due to the un-ordered relaxation between the top-K targets. We do have the first constraint for the ordered top-K targets. Instead, we will need to enforce the logit of each top-K targets to be greater than all other classes not in the top-K target list, so we will have $K\times (C-K) \times M$, where $C$ is the total number of classes, and $M$ the number of training models.
> The KL formulation will be less effective in handling this since we need to specify the top-down adversarial logits. We can remove the KL and instead use the margin based formulation: $min(top-k targets logits) - max(logtis of C-K classes) > margin$. The QP formulation can be extended to address un-ordered top-K attacks naturally with more constraints, and increased QP solving time.
>
> We choose to focus on ordered top-K UAPs, since it represents one of the deepest vulnerability of DNNs. We leave the set-based top-K UAPs as future work, since there are more baseline approaches to be studied and compared, out of the scope of a single conference submission.
>
> ---
>
> ### W6- Generalization beyond ImageNet-1k. All results are on ImageNet-1k (one image per class). No experiments on non-natural domains, larger label spaces, or multi-label/ranking tasks. The method’s reliance on class-wise sampling (and removing images when GT=t₁) slightly biases the data protocol.
>
> **Ans.** One main reason is due to the availability of pretrained models (e.g., many in the timm package for ImageNet-1k). ImageNet-1k is also widely used in evaluating attacks. We leave the study of out-of-domain transferrability as the future work.
>
> ---
>
> ### W7- Limited analysis against strong evaluation checklists. There is no EOT for stochastic components, no black-box query-budget study, no physical-world tests, and no transfer-from-surrogates vs white-box comparison beyond model ensembles.
>
> **Ans.** In the global response, we add the evaluation on random Guassian blurring, which can be viewed as a prelimainary EOT analysis. The model-ensemble experiments can be treated as transfer-from-surrogates. For black-box and physical-world tests, we think they are interesting and challenging, and are out-of-the-scape of this submission.
>
> ---
>
> ### W8- Foundational models: partial success. Transfer to CLIP ViT-B and ConvNeXtV2-H is notable but lower for larger K and for EVA-2 (Table 2 & 24–25); the paper does not analyze why (e.g., pretraining objectives, feature spectrum, or logit scale).
>
> **Ans.** In the global response, we show much improved transferability to those foundation models, especially top-1 targeted UAPs. The pretraining objectives (masked image modeling and/or contrastive language-image pretraining) and the large scale pretrain data corpus are the two main aspects that challenge the transferability of UAPs learned with traditional cross-entropy loss and typically ImageNet-1k training data.

---

> ### Author Response · Authors · 2025-12-02
>
> ### Q1- Perceptibility. For the main ASR tables (e.g., Table 1–6), can you report LPIPS/SSIM and PSNR for (δ, x+δ) over Dtest, and provide human-rated visibility at multiple viewing scales? Also show ASR vs ℓ₂ / ℓ∞ curves for K=1…6 across key models.
>
>
> **Ans.** Yes, we will add LPIPS/SSIM and PSNR in revision based on the refactored model presented in the global response. For human pilot study, we can try one in our lab environment. In future, we will seek collaboration with psychology professor in our institution to perform human study in a more rigirous way.
>
> ---
>
> ### Q2- Ordered vs unordered. How do ASRs change if evaluation is unordered Top-K or allows partial order constraints (e.g., only top-m fixed, rest any order)? Please add ablations.
>
> **Ans.** As we address your concern (W5), this is doable, but will involve many different baseline methods, for which we think it is better to investigate in a different submission focusing on set-based top-K UAPs.
>
> ---
>
> ### Q3- Confidence intervals. For Mean/Best/Worst protocols, report per-image success distributions and 95% CIs, and test sensitivity to the number of target-list samples (5→20).
>
> **Ans.** As we address your concern (W3), we will report per-image success distributions and 95% CIs in revision with increased number of random seeds (from 5 to 20).
>
> ---
>
> ### Q4- QP details & optimality. In Eq. (7–8) the QP is solved on feature variables with linear Top-K constraints, then one gradient step updates δ. Please characterize feasibility rates of the QP per batch, KKT residuals, and show convergence (or cyclic behavior) of the alternating scheme. Any observed mode collapse to a few classes?
>
> **Ans.** We use the qpth package [a] in the QP, and are working on the possibility to reveal the optimization status.
>
> [a] Brandon Amos and J. Zico Kolter, OptNet: Differentiable Optimization as a Layer in Neural Networks, arXiv preprint 2017
>
>
> ---
>
> ### Q5- Sensitivity to λ and batch sizes. Provide sweeps of λ (and smax if used inside the KL target construction) and mini-model/mini-data batch sizes, reporting ASR–energy frontiers.
>
> **Ans.** As reported in our global response, we eliminate the hyperparameter $\lamda$, and instead resort to explict $\ell_p$ projection to contol the UAP energy. For mini-data batch sizes, we observed that small mini-data-batch sizes are necesarry (e.g., 8), since for larger ones, the algorithm will take a much longer time to find the feasible adversarial region from the randomly initialized UAP in learning. We will summarize those observations in revision.
>
> ---
>
> ### Q6- Feature-space choice. You set and enforce order at logits. What if the linear constraints are enforced earlier (e.g., pre-classifier feature margins), or you use logit temperature scaling to control numerical conditioning—does that improve energy efficiency?
>
> **Ans.** Order at logits is by definiton the constraints of ordered top-K attacks. Moving the linear constraints ealier will introduce two challenges: we need to linearize the remaining part of a DNN to exploit QP solver, and those linear constraints will not necessarily enable ordered top-K attack even if they are satisfied.  For logit temperature scaling, since we are attack pretrained DNNs, which did not have logit temperature scaling, applying the scaling in learning UAPs will make the UAPs less effective in evaluation.
>
> ---
>
> ### Q7- Why EVA-2 is harder? Provide spectral/semantic analyses (Fourier energy, token attention patterns) explaining lower transfer to EVA-2 vs CLIP/ConvNeXtV2-H; test whether in-distribution CLIP fine-tuning or logit scaling reduces the gap.
>
>
> **Ans.** As we address your concern (W8), in the global response, we show much improved transferrability to those foundation models, especially top-1 targeted UAPs. The pretraining objectives (masked image modeling and/or contrastive language-image pretraining) and the large scale pretrain data corpus are the two main aspects that challenge the transferrability of UAPs learned with traditional cross-entropy loss and typically ImageNet-1k training data.
>
>
> ---
>
> ### Q8- Cross-norm transfer. If δ is trained under ℓ₂ but evaluated under ℓ∞ (and vice versa), how do ASRs change? Include ε-sweeps like Fig.10 across norms.
>
> **Ans.** As reported in the global response, we now explicitly project $\delta$ w.r.t. $\ell_p$ budget. The new results will be show the their effects of ASRs.
>
>
> ---
>
> ### Q9- Stronger evaluations. Add EOT (noise/resize-crop), JPEG/bit-depth compressions, and rank-stability detectors; test simple defenses (randomized smoothing, small denoisers) and non-differentiable checks (e.g., semantic-coherence filters) against your δ.
>
> **Ans.** As reported in the global response, we tested simple left-right flipping and Gaussian random blurring. Left-right flippling has negligible negative impacts, while bluring has significant impacts on ASRs. To our knowledge, this is generally true for most of attacks.

---

> > ### Author Response · Authors · 2025-12-02
> >
> > ### Q10- Black-box budgets. Can your δ be approximated from queries only (no gradients) given the same ordered Top-K targets? Report query counts to reach ASR comparable to white-box
> >
> > **Ans.** Black-box attack is out of the scope of this paper. To our knowledge, state-of-the-art black-box top-k attacks [1], even for single-image and single-model, are extremely challenging, entailing a large number of queries in order to approximate the gradients.
> >
> > [1] Md Farhamdur Reza, Richeng Jin, Tianfu Wu and Huaiyu Dai, GSBA$^K$: top-K Geometric Score-based Black-box Attack, ICLR 25.

---

### Official Review · Reviewer_V4wR · 2025-11-02

**Soundness:** 3
**Presentation:** 3
**Contribution:** 3
**Rating:** 6
**Confidence:** 4

**Summary:**

This paper introduces AllAttacK, a novel framework for learning Ordered Top-K Targeted Universal Adversarial Perturbations (UAPs). Unlike prior work focused on untargeted or top-1 attacks, AllAttacK jointly optimizes adversarial perturbations along three independent axes — the data axis (generalization across images), the model axis (transferability across architectures), and the target axis (control over ordered Top-K label sequences). The problem is formulated as a Maximum Satisfiability (MAXSAT) optimization task, instantiated via two scalable variants: the KL-divergence-based AllKLAttacK and the quadratic-programming-based AllQuadAttacK. Large-scale experiments on ImageNet-1k across 27 diverse models demonstrate strong double transferability of the learned perturbations — across unseen data and unseen models, and show that ordered Top-K targeted UAPs expose systematic vulnerabilities in modern deep neural networks’ ranking robustness.

**Strengths:**

1. The paper is the first to systematically study ordered Top-K targeted universal adversarial perturbations (UAPs), extending the scope of adversarial attack research beyond traditional untargeted or Top-1 targeted settings. By enforcing ordered Top-K constraints, the work aligns with emerging robustness evaluation standards (e.g., NIST SP800-226, 2025) and provides a new benchmark for model ranking stability.
2.  Extensive experiments on ImageNet-1k involving 27 diverse models (including CNNs, Vision Transformers, CLIP, EVA2, and adversarially trained networks) demonstrate that the learned perturbations achieve high double-transferability across unseen data and unseen models. The experimental coverage is significantly broader than prior UAP studies.

**Weaknesses:**

1. The MAXSAT formulation (Eq. 4) is conceptually sound but lacks formal solvability or approximation guarantees. No complexity or convergence analysis is provided.

2. The quadratic-programming relaxation implicitly assumes local convexity of the logit-ordering constraints. However, the paper does not analyze feasibility margins or address potential degeneracy when logits become nearly tied, which could impact optimization stability and reproducibility.

3. The paper focuses exclusively on attack performance without considering possible countermeasures, robustness training, or defense evaluation. Including even a preliminary analysis of model responses or defense sensitivity would strengthen the practical significance.

4. The precise norm bound (e.g., ε under L∞ or L2 constraints) used for the universal perturbations is not explicitly stated. Clarifying these values is important for reproducibility and for fair comparison with prior UAP baselines.

5. The PCC (Pearson correlation) experiment in Section 3.3 seems to analyze general UAP behavior rather than the specific ordered Top-K objective introduced by AllAttacK. The connection between this analysis and the proposed method should be clarified.

6. The exploration of transferability across data and model axes largely follows prior UAP literature, offering incremental rather than conceptual novelty relative to existing universal attack frameworks.

**Questions:**

1. What is the computational complexity of AllQuadAttacK compared to AllKLAttacK when trained on large model ensembles? Does the QP-based method scale linearly, quadratically, or worse with respect to the number of models and target classes?

2. The paper focuses exclusively on ordered Top-K targeted attacks. How would the proposed framework behave under unordered Top-K settings, where only the presence of target classes in the Top-K predictions matters, rather than their specific order?

---

> ### Author Response · Authors · 2025-11-30
>
> Dear Reviewer V4wR,
>
> Thank you for your efforts. We appreciate you for the recognition of our submission as the first to to systematically study ordered Top-K targeted UAPs.
>
> Sorry for submitting the rebuttal late, since we have been testing some modifications with very strong results obtained, reported in the global response.  We address your concerns as follow.
>
> ---
>
> ### W1: The MAXSAT formulation (Eq. 4) is conceptually sound but lacks formal solvability or approximation guarantees. No complexity or convergence analysis is provided.
>
> **Ans.** Eq.4 is a nonlinearly constrained optimization problem, and the nonlinearity depends on the DNNs under attack. In general, we are not aware of the existence of formal solvaility. We show two approximation methods by exploiting the prior art of single-model single-image attack methods, the KL based reformulation (Eq.5 and 6) and the QuadAttacK empowered approach (Eq.7 and 8). In terms of approximation guarantees, we did face the challenges, similar in spirit to the conventional single-model single-image attack such as the well-known CW method and PGD method. Unless we use much simplified DNNs such as 1-layer or 2-layer ReLU DNNs as typically adopted in the theoretical analyses of DNNs, it is our understanding that it is not possible at  present to provide meaningful theoretical complexity or convergence analyses that are potentially useful for practical sceanrios.  So, we resort to the large-scale empirical experimental analyses.  It is also our hope that our empirical analyses and to-be-released source code will be potentially useful for the community to dive into theoretical understanding.
>
> ---
>
> ### W2: The quadratic-programming relaxation implicitly assumes local convexity of the logit-ordering constraints. However, the paper does not analyze feasibility margins or address potential degeneracy when logits become nearly tied, which could impact optimization stability and reproducibility.
>
> **Ans.** This is an insightful question. Ordered top-K targets enforce the local convexity of the logit-ordering constraints, whose satisfactability is subject to the decision boundaries of different pretrained DNNs. We did use margins in solving Eq.7 as done in the original QuadAttacK. We will make this detail clear in revision. For simplicity, we use a single predefined margin $3$ across all experiments to eliminate the tuning and to faciliate easier comparisons in different experiments. So, the reported results are indeed not the optimal results we could possibly obtain at the expense of more compute for hyperparameter search. In our main experiments, we use randomly sampled ordered top-K targets to benchmarking the attack effectiveness. In Section 4.5, we show that using semantically coherent ordered top-K targets, which ease the logit-ordering constaints as they are commonly preserved by pretrained DNNs on benign images, leads to much higher attack success rates. So, the UAP vulnerability of DNNs reported in our main experiments are very conservative.
>
> ---
>
> ### W3: The paper focuses exclusively on attack performance without considering possible countermeasures, robustness training, or defense evaluation. Including even a preliminary analysis of model responses or defense sensitivity would strengthen the practical significance.
>
> **Ans.** We completely agree with you on this. We have some very preliminary exploration during the development of AllAttacK. Seeking defense methods or robustification methods is our ultimate goal, which is why we focus on empirically analyzing how vulnerable existing DNNs are in terms of the most aggressive ordered top-K attacks. The methods we studied for attacking DNNs can be straightforwardly repurposed for "the good", where the ordered top-K targets have the ground-truth label as the top-1 target, and semantically related classes as the remaining targets ordered in terms of label similarities. Combining margins for logits-ordering, this leads to the max-margin formulation of training DNNs to improve the robustness from scratch, offering an alternative method to adversarial training approaches. We tried this, but have not been able to obtain strong performance on benign data compared to vanilla cross-entropy loss based training. So, we focus on attack in this paper, and leave the robustification exploration to the future work. We will discuss these aspects in revision.
>
> ---
>
> ### W4: The precise norm bound (e.g., ε under L∞ or L2 constraints) used for the universal perturbations is not explicitly stated. Clarifying these values is important for reproducibility and for fair comparison with prior UAP baselines.
>
> **Ans.** Thank you for pointing this out. As we reported in the global response, we have modified the experimental settings by explicit $\ell_2$ budget, following common quasi-imperceptibility settings in the literature. We will add ablations on $\ell_{inf}$ budgets in revision. We will release this refactored code for reproducibility.

---

> > ### Author Response · Authors · 2025-11-30
> >
> > ### W5: The PCC (Pearson correlation) experiment in Section 3.3 seems to analyze general UAP behavior rather than the specific ordered Top-K objective introduced by AllAttacK. The connection between this analysis and the proposed method should be clarified.
> >
> > **Ans.** Yes, the PCC analyses are to quantitatively demonstrating that UAPs "will induce benign images as `negligible noise` to DNNs". The connection between this analysis and the proposed method lies in two aspects: the proposed two approaches built on the prior art induce consistent UAP behaviors, and combine with the visualization (see Fig.1 and the HTML tool in the supplementary), AllAttacK UAPs enforce semantically meaningful perturbations emerged, especially for an ensemble of training models, or robustified models. One potential use of this observation is to robustify pretrained models (such as CLIP vision encoder) by enforcing their adversarial perturbations to be semantically meaningful. During the development, we had a preliminary hypothesis on the Adversarial Interpretability-Robustness: A DNN will be adversarially robust in a holistic way if its AllAttacK adversarial perturbations are semantically meaningful (i.e.,  in the close proximity to or even inside the real data manifold). From a  quantitatively equivalent viewpoint, it means that the perturbations themselves in isolation will be classified by the DNN with the top-$K$ predictions equal to the ordered top-$K$ targets ($K\geq 1)$. Ideally and ultimately (in the long run), a DNN is certified to be robust if its AllAttacK perturbations are confined to be high-fidelity synthesized images for the ordered top-$K$ targets, i.e., the closed-loop of AllAttacK-as-Generator. We will discuss this aspect in the revision.
> >
> > ---
> >
> > ### W6: The exploration of transferability across data and model axes largely follows prior UAP literature, offering incremental rather than conceptual novelty relative to existing universal attack frameworks.
> >
> > **Ans.** We agree that transferability across data and model axes is following prior settings. The combination with ordered top-K attack make what we present in the submission the first large-scale exploration jointly along the data-model-target axes. Even though the transferability across data and model axes is conceptually the same as the prior art, we introduce new and large-scale experiments: In the introduction section, we present the challenge: (line 41) "**achieving effective top-1 targeted UAPs has proven challenging**", (lines 49-51) "**it is unclear whether targeted UAPs can retain their attacking power for ensembles of those disparate DNNs, as well as adversarially-robustified counterparts**", and (lines 71-72) "**Learning ordered top-K targeted UAPs has not been studied in the literature, however.**"
> >
> > ---
> >
> > ### Q1: What is the computational complexity of AllQuadAttacK compared to AllKLAttacK when trained on large model ensembles? Does the QP-based method scale linearly, quadratically, or worse with respect to the number of models and target classes?
> >
> > **Ans.** We provide a coarse runtime analysis in the Appendix C, reproduced here for your reference: we have profiled the \textit{QP} attack on 4 different configurations. We report the recorded runtimes of our method **per-epoch**. Every attack configuration in this paper runs for 50 epochs. Each configuration has been profiled on a single Nvidia A100 80G GPU and averaged over the course of 12 epochs. We note runtime does not seem to be affected significantly by the choice of $K$ in the attack, but is more directly affected by the models being attacked.
> >
> > | Model / K | K=2 | K=5 |
> > |:----------|:----|:----|
> > | ResNet-50 | 58s | 55s |
> > | 4-model ensemble | 164s | 164s|
> >
> > The runtime is largely determined by the total forward computatation time of models in training UAPs. We have used Wandb in our refactored code, and will report detailed time logs for all the runs in the revision.

---

> ### Author Response · Authors · 2025-11-30
>
> ### Q2: The paper focuses exclusively on ordered Top-K targeted attacks. How would the proposed framework behave under unordered Top-K settings, where only the presence of target classes in the Top-K predictions matters, rather than their specific order?
>
> **Ans.** In general, the number of constraints will increase (significantly) in Eq. 4 due to the un-ordered relaxation between the top-K targets. We do have the first constraint for the ordered top-K targets. Instead, we will need to enforce the logit of each top-K targets to be greater than all other classes not in the top-K target list, so we will have $K\times (C-K) \times M$, where $C$ is the total number of classes, and $M$ the number of training models.
> The KL formulation will be less effective in handling this since we need to specify the top-down adversarial logits. We can remove the KL and instead use the margin based formulation: **min(logits of top-k targets) - max(logtis of C-K classes) > margin**.  The QP formulation can be extended to address un-ordered top-K attacks naturally with more constraints, and increased QP solving time.

---

### Official Review · Reviewer_MG42 · 2025-11-04

**Soundness:** 3
**Presentation:** 2
**Contribution:** 2
**Rating:** 2
**Confidence:** 3

**Summary:**

The paper present AllAttacK to generate ordered topk targeted universal adversarial perturbations across different models.

**Strengths:**

1. The paper is very well motivated.

2. Extensive list of model coverage, including model families, and standard and adversarially trained models.

**Weaknesses:**

Despite the extensive experiments, the paper lacks insightful analysis. In Sec4, large tables are presented with only brief two-liner summaries. The cramped spacing before/after tables blends them into the text, and it overwhelms readers with numbers rather than insight. Please highlight a few key results and discuss them in depth.

**Questions:**

1. Can you clarify 173-175, specifically on how $\delta_T$ is an energy-minimized points?

2. Ln313 states that "test the learned UAPs on 6 DNNs ", then what is the x-axis (resnet18, resnet34, ..., MlpMixer-B) in Table 1?

3. Ln 152-157: What is the significance of separately considering Dtrain and Dtest on the data axis? Is generalization from training images to test images a non-trivial finding?

---

> ### Author Response · Authors · 2025-11-30
>
> Dear Reviewer MG42,
>
> Thank you for your efforts in reviewing our submission. We appreciate you for the recognition of the two strengths of our submission.
>
> Sorry for submitting the rebuttal late, since we have been testing some modifications with very strong results obtained, which we report in the global response.
>
> Here, please let us address your concerns as follow.
>
> ---
>
> ### Q1: Can you clarify 173-175, specifically how $\delta_T$ is an energy-minimized points?
>
> **Ans.** We refer to the solution from Eqn.4 (lines 164-169) as the $\delta_T$,  where **the objective function is to explicitly $\text{minimize}_{\delta_T} ||\delta_T||_p^2$**, i.e., the $\ell_p$ energy, and we often use $\ell_2$ in experiments.
>
> ---
>
> ### Q2: Ln313 states that "test the learned UAPs on 6 DNNs ", then what is the x-axis (resnet18, resnet34, ..., MlpMixer-B) in Table 1?
>
> **Ans.** The x-axis in Table 1 shows the 18 models used in learned the UAPs on 1000 images sampled from ImageNet-1k **trainset** (1 image per category, line 157).  By "test the learned UAPs on 6 DNNs", it means the results reported in Table 2, which tested the UAP on the 6 models (not included in the 18 training models) using 1000 images sampled from ImageNete-1k **valset**.  We will clarify this in revision.
>
> ---
>
> ### Q3: Ln 152-157: What is the significance of separately considering Dtrain and Dtest on the data axis? Is generalization from training images to test images a non-trivial finding?
>
> **Ans.** Our UAPs are learned using Dtrain, and we want to test their transferrability along the data axis by evaluating their attack success rates on Dtest.  **This is a common setting in learning UAPs**, similar in spirit to how we evaluate an ImageNet image DNN model, for which we evaluate models based on their performance on the validation/test data.  This is significant, especially for the doubly transferrable aspect, i.e., testing UAPs on unseen models (e.g., the 6 models in Table 2) and unseen images (ImageNet 1k valset), since the attack success clearly show (lines 95-97): **"the existence of such targeted UAPs could indicate that there were aspects that have been overlooked in a systematic way in DNNs. Seeking quantitative analyses of AllAttacK will facilitate us understanding the adversarial vulnerability at the fundamental level better (e.g., how aggressive can adversarial attacks be?)."**
>
> ---
>
> ### W1: Despite the extensive experiments, the paper lacks insightful analysis. In Sec4, large tables are presented with only brief two-liner summaries. The cramped spacing before/after tables blends them into the text, and it overwhelms readers with numbers rather than insight. Please highlight a few key results and discuss them in depth.
>
> **Ans.**  We  respectfully disagree with you on the overall assessment of lacking insightful analysis. but agree with you that the presentation of those analyses can be improved. We take this opportunity to clarify this important aspect.
>
> In the introduction section, we clearly present the challenge: (line 41) "**achieving effective top-1 targeted UAPs has proven challenging**", (lines 49-51) "**it is unclear whether targeted UAPs can retain their attacking power for ensembles of those disparate DNNs, as well as adversarially-robustified counterparts**", and (lines 71-72) "**Learning ordered top-K targeted UAPs has not been studied in the literature, however.**" With these context, we summarize the observations from Table 2 compactly, (lines 337-339): "These results, especially the double-transferability across testing models and testing images, represent a significant leap forward from the prior art, as well as broader impacts as dicussed in Appendix A.".
>
> In the Appendix A, we summarize (lines 704-708) "**The promising transferability of the learned ordered top-K UAPs to unseen models, especially those at the so-called foundation model level (e.g., the CLIP ViT-B in Table 2), might be exploited in a harmful way for applications built on those models. Powerful defense methods should be studied, which we will investigate in our future work. We will also release our source code to encourage more research on studying defense methods against the proposed AllAttacK.**"
>
> We will make these aspects more clear in both introduction and experiment sections following your suggestions and reflecting our new observations summarized in the global response.
>
> For the presentation of Tables, we tried our best to put them close to the text descriptions, to leverage heatmap-enchanced table visualization, and to group tables in the same page with similar observations. We will certainly take a fresh look at the layout of tables, together with the to-be-revised text of summarizing the insights better.

---

### Author Response · Authors · 2025-11-30
**Global responses -- method and experimental setting updates**

Dear Reviewers and ACs,

Thank you all for your big efforts in handling our submission. In addition to addressing concerns raised by individual reviewers, we would like to report some new observations we obtained during the rebuttal period in this global response.

**Data** We increase the number of training images by randomly sampling 5 images per category in the ImageNet-1k trainset, resulting 5000 images in total. For a given ordered top-K list of targets, we remove the training images whose ground-truth labels are the same as the top-1 target, leading to 4995 images used in training ordered top-K UAPs. We use the entire ImageNet-1k validation set (50,000 images) in evaluating the learned UAPs, except for those whose ground-truth labels are the same as the top-1 target, leading to 49,950 images in evaluating per UAP.

**Models under attack** We slightly update the model ensemble by selecting models solely from the timm package which have checkpoints trained under a resolution of $224\times 224$:

idx | model                                            |   img_size |   top1 |   top1_err |   top5 |   top5_err | param_count (MB) |
|--:|:-------------------------------------------------|-----------:|-------:|-----------:|-------:|-----------:|-----------------:|
| 1 |  resnet18.a1_in1k                                | 224	| 71.504	| 28.496	| 90.086 	|	9.914	| 11.69 |
| 2 |  resnet50.a1_in1k                                | 224	| 80.382	| 19.618	| 94.598	| 	5.402	| 25.56 |
| 3 |  densenet121.ra_in1k                             | 224    | 74.750	| 25.250	| 92.158	| 	7.842	| 7.98  |
| 4 |  densenet169.tv_in1k                             | 224	| 75.922	| 24.078	| 92.976	|   7.024	| 14.15 |
| 5 |  hrnet_w18.ms_in1k                               | 224	| 72.340	| 27.660	| 90.688	|	9.312	| 13.19 |
| 6 |  convnext_tiny.fb_in1k							|224	| 82.066	| 17.934	| 95.854	| 4.146		| 28.59 |
| 7 |  convnext_large.fb_in1k							|224	| 84.304	| 15.696	|	96.892	|	3.108	| 197.77 |
| 8 |  maxvit_rmlp_small_rw_224.sw_in1k					|224	| 84.488	| 15.512	| 96.772	| 3.228		| 64.90 |
| 9 |  coatnet_rmlp_1_rw_224.sw_in1k					| 224	| 83.350	| 16.650	|	96.452	|	3.548	| 41.69 |
| 10 |  vit_base_patch16_224.augreg_in1k				| 224	| 79.154	| 20.846	| 94.108	|	5.892	| 86.57	|
| 11 |  convmixer_768_32.in1k 							| 224	| 80.158	| 19.842	| 95.070	|	4.930	| 21.11 |
| 12 |resnet34.a1_in1k									| 224	| 76.420	| 23.580	| 92.896	|	7.104	| 21.80 |
| 13 |resnet50d.a1_in1k									| 224	| 80.730	| 19.270	|	94.678	| 5.322		| 25.58 |
| 14 |densenet161.tv_in1k                              | 224	| 77.384	| 22.616	| 93.656	| 6.344		| 28.68 |
| 15 |densenet201.tv_in1k                              | 224	| 77.284	| 22.716	| 93.480	| 6.520		| 20.01 |
| 16 |hrnet_w30.ms_in1k                                | 224	| 78.202	| 21.798	| 94.220	| 5.780		| 37.71 |
| 17 |convnext_base.fb_in1k								| 224	| 83.838	| 16.162	| 96.746	| 3.254		| 88.59 |
| 18 |maxvit_base_tf_224.in1k 							| 224	| 84.884	| 15.116	| 97.002	| 2.998		| 119.47|
| 19 |mixer_b16_224.goog_in21k_ft_in1k 					| 224	| 76.616	| 23.384	| 92.250	| 7.750		| 59.88 |
| 20 |swin_base_patch4_window7_224.ms_in1k 				| 224	|	83.604	| 16.396	| 96.450	| 3.550		| 87.77 |
| 21 |convnextv2_huge.fcmae_ft_in1k						| 224	| 86.260	| 13.740	| 97.752	| 2.248		| 660.29 |
| 22 |vit_base_patch16_clip_224.openai_ft_in12k_in1k 	| 224	| 85.954	|	14.046	| 97.724	| 2.276		| 86.57  |
| 23 |eva_giant_patch14_224.clip_ft_in1k 				| 224	| 88.896	| 11.104	| 98.672	| 1.328		| 1,012.56 |

**Updates of AllAttacK w/ QuadAttacK**

In Table 2 of our submission, the AllAttacK w/ KL shows better transferrability than AllAttacK w/ QuadAttacK. We investigated this during the rebuttal period with inspritations from reviewers' comments. More specifically, we update Eqn.8,

$ \delta_T  \Leftarrow \delta_T - \gamma \cdot \frac{\partial}{\partial \delta}\Bigl(\frac{\lambda}{S\cdot I}\sum_{i,s}\beta_s \cdot ||z^*_{i,s}-f(x_i^{\text{benign}}+\delta;\theta_s)||^2_2+ \|\delta\|^2_2\Bigr)|_{\delta=\delta_T} $

in which we need $\lambda$ to balance the QuadAttacK loss and the $\ell_2$ energy of the current UAP, and the optimizer we used is AdamW with the default weight decay $0.01$ on the UAP (as the only learnable parameters in learning).

We remove both the $\|\delta\|^2_2$ term in Eqn.8 (and thus the $\lambda$ trade-off parameter) and the weight decay term in AdamW. Instead, we perform a $\ell_2$ projection of the current UAP after each mini-batch with a predefine $\ell_2$ energy budget (e.g., $\tau=25$ used in the update experiments reported in this global response). We have,

$ \delta_T  \Leftarrow \delta_T - \gamma \cdot \frac{\partial}{\partial \delta}\Bigl(\frac{1}{S\cdot I}\sum_{i,s}\beta_s \cdot ||z^*_{i,s}-f(x_i^{\text{benign}}+\delta;\theta_s)||^2_2\Bigr)|_{\delta=\delta_T} $

For the projection, we have $\delta_T = \tau \cdot \frac{\delta_T}{\|\delta\|^2}$, which preserves the direction of the learned UAP.

---

> ### Author Response · Authors · 2025-11-30
> **Global responses -- New results on Top-1 targeted attacks.**
>
> **New results** We use the first 11 models as the training models, and the remaining 12 models as the testing models (including foundation models such as ConvNeXtV2-Huge, CLIP and EVA-Giant). We obtain very strong performance.
>
> We note that we keep the $\ell_2$ budget $\tau=25$ to ensure the learned UAPs remain quasi-imperceptible.
>
> We also tested two defensive preprocessing: left-right flipping (**ASR-LRFlip**), and Gaussian Blur with a kernel size 5 and randomly sampled sigma between 0.1 and 2 (**ASR-Blur**).
>
> + **Strong UAP learning convergence:** 11 training models + 4995 training images + l2 budget ($\tau=25$)
> |model|ASR|ASR-LRFlip|ASR-Blur|l1|l2|linf|
> | :--- | :--- | :---| :--- | :--- | :--- | :--- |
> |resnet18.a1_in1k|93.0847|92.5605|45.5242|6653.6782|23.8857|0.4868|
> |resnet50.a1_in1k|92.4395|92.174|54.0927|6651.8135|23.8819|0.4868|
> |densenet121.ra_in1k|98.8911|92.9839|59.1532|6650.8862|23.8770|0.4868|
> |densenet169.tv_in1k|98.3468|92.1774|57.4597|6651.5312|23.8792|0.4867|
> |hrnet_w18.ms_in1k|98.1452|89.7581|58.0444|6651.7280|23.8789|0.4868|
> |convnext_tiny.fb_in1k|98.3871|97.4597|60.4637|6651.7993|23.8795|0.4868|
> |convnext_large.fb_in1k|98.9113|98.4476|61.9758|6654.1479|23.8850|0.4867|
> |maxvit_rmlp_small_rw_224.sw_in1k|99.5766|68.7097|95.1613|6652.0483|23.8792|0.4867|
> |coatnet_rmlp_1_rw_224.sw_in1k|99.3952|90.0807|65.4032|6651.2144|23.8777|0.4868|
> |vit_base_patch16_224.augreg_in1k|99.7782|92.4597|65.0807|6651.5103|23.8780|0.4868|
> |convmixer_768_32.in1k|93.7903|93.8911|60.3226|6650.0835|23.8750|0.4868|
>
> + **Strong data transferability:** 11 training models + 49,950 validation images + l2 budget ($\tau=25$)
> |model|ASR|ASR-LRFlip|ASR-Blur|l1|l2|linf|
> | :--- | :--- | :--- | :---| :--- | :--- | :--- |
> |resnet18.a1_in1k|87.1792|86.7708|42.3964|6620.2915|23.8012|0.4869|
> |resnet50.a1_in1k|88.8889|88.4524|53.0210|6614.0015|23.7853|0.4868|
> |densenet121.ra_in1k|94.7468|87.7698|56.7067|6613.5337|23.7835|0.4868|
> |densenet169.tv_in1k|94.2142|87.5596|55.1692|6618.1641|23.7955|0.4868|
> |hrnet_w18.ms_in1k|94.1241|83.5856|55.9520|6617.3096|23.7926|0.4868|
> |convnext_tiny.fb_in1k|97.2252|96.5245|63.0450|6611.2651|23.7787|0.4868|
> |convnext_large.fb_in1k|98.1601|97.8118|64.6066|6612.1377|23.7806|0.4868|
> |maxvit_rmlp_small_rw_224.sw_in1k|98.6346|96.7347|71.7418|6611.4321|23.7787|0.4868|
> |coatnet_rmlp_1_rw_224.sw_in1k|96.6306|86.6747|65.4915|6612.4673|23.7811|0.4868|
> |vit_base_patch16_224.augreg_in1k|96.9550|90.0340|67.2733|6617.1782|23.7934|0.4868|
> |convmixer_768_32.in1k|89.5095|89.6617|59.6036|6611.0078|23.7774|0.4868|
>
> + **Strong model transferability:** 12 testing models + 4,995 training images + l2 budget ($\tau=25$)
> |model|ASR|ASR-LRFlip|ASR-Blur|l1|l2|linf|
> | :--- | :--- | :--- | :---| :--- | :--- | :--- |
> |resnet34.a1_in1k|95.3226||95.3629|54.8589|6653.3628|23.8841|0.4868|
> |resnet50d.a1_in1k|94.0524|93.3468|60.9677|6651.9062|23.8809|0.4867|
> |densenet161.tv_in1k|93.7298|92.8226|49.9395|6654.7437|23.8868|0.4867|
> |densenet201.tv_in1k|89.2944|87.8831|46.2500|6662.2749|23.9064|0.4867|
> |hrnet_w30.ms_in1k|75.2823|77.3387|42.7621|6670.8320|23.9399|0.4867|
> |convnext_base.fb_in1k|97.9839|98.2258|62.7218|6654.8555|23.8891|0.4867|
> |maxvit_base_tf_224.in1k|69.9798|73.8306|37.5000|6746.3203|24.1671|0.4868|
> |mixer_b16_224.goog_in21k_ft_in1k|57.2984|53.6895|33.5686|6718.0361|24.0869|0.4869|
> |swin_base_patch4_window7_224.ms_in1k|87.4194|87.9032|48.3871|6693.4712|24.0011|0.4867|
> |convnextv2_huge.fcmae_ft_in1k|76.0282|69.6774|47.6210|6698.5806|24.0096|0.4868|
> |vit_base_patch16_clip_224.openai_ft_in12k_in1k|86.1089|85.3427|47.9436|6660.3882|23.9003|0.4868|
> |eva_giant_patch14_224.clip_ft_in1k|48.3064|47.6815|28.7298|6686.9995|23.9917|0.4869|
>
> + **Strong model and data double-transferability:** 12 testing models + 49,950 validation images + l2 budget ($\tau=25$)
> |model|ASR|ASR-LRFlip|ASR-Blur|l1|l2|linf|
> | :--- | :--- | :--- | :--- | :--- | :--- | :--- |
> |resnet34.a1_in1k|91.5215|92.0080|53.5055|6617.0210|23.7934|0.4868|
> |resnet50d.a1_in1k|90.9549|90.4464|60.3984|6612.5952|23.7814|0.4868|
> |densenet161.tv_in1k|89.0591|87.9700|47.8138|6619.2202|23.7973|0.4869|
> |densenet201.tv_in1k|83.9500|82.1341|43.5996|6626.3823|23.8172|0.4869|
> |hrnet_w30.ms_in1k|67.1832|69.0951|39.7998|6635.0454|23.8489|0.4868|
> |convnext_base.fb_in1k|97.7457|97.8318|67.0030|6614.1577|23.7861|0.4868|
> |maxvit_base_tf_224.in1k|74.2503|77.1732|43.6597|6692.4468|24.0130|0.4868|
> |mixer_b16_224.goog_in21k_ft_in1k|58.9490|55.2913|37.8458|6671.8579|23.9517|0.4870|
> |swin_base_patch4_window7_224.ms_in1k|87.6597|88.4404|52.3223|6649.2515|23.8857|0.4868|
> |convnextv2_huge.fcmae_ft_in1k|77.4434|72.2122|52.8649|6649.5859|23.8857|0.4869|
> |vit_base_patch16_clip_224.openai_ft_in12k_in1k|80.6507|80.5806|46.2523|6623.0073|23.8058|0.4869|
> |eva_giant_patch14_224.clip_ft_in1k|42.8168|42.7307|27.2733|6640.9595|23.8660|0.4869|
>
> We note that the double-transferability is strong on even both CLIP (80%) and EVA (42.8%), which show significant improvement against original results in Table 2.

---

> ### Author Response · Authors · 2025-11-30
> **Global responses -- New results on ordered Top-2 targeted attacks.**
>
> Compare with the strong double-transferability observed for Top-1 UAPs, ordered Top-2 UAPs show strong data transferability, but struggles with model transferability.
>
> + **Promising UAP learning convergence:** 11 training models + 4995 training images + l2 budget ($\tau=25$)
> |model|ASR|ASR-LRFlip|ASR-Blur|l1|l2|linf|
> | :--- | :--- | :--- | :---| :--- | :--- | :--- |
> |resnet18.a1_in1k|51.2500|46.9758|14.8790|6995.0991|24.1318|0.4513|
> |resnet50.a1_in1k|72.9032|73.3266|23.9718|6980.2593|24.0960|0.4512|
> |densenet121.ra_in1k|65.7056|44.9798|20.1008|6983.4048|24.1035|0.4512|
> |densenet169.tv_in1k|52.7621|34.3145|14.8790|7004.9189|24.1583|0.4512|
> |hrnet_w18.ms_in1k|55.6250|19.5968|14.0323|7008.9077|24.1603|0.4512|
> |convnext_tiny.fb_in1k|75.6452|68.9919|23.5887|6992.1416|24.1273|0.4511|
> |convnext_large.fb_in1k|61.4516|58.5887|16.6331|7024.8569|24.2111|0.4511|
> |maxvit_rmlp_small_rw_224.sw_in1k|89.5766|77.9839|33.5685|6980.9492|24.0924|0.4511|
> |coatnet_rmlp_1_rw_224.sw_in1k|84.9597|64.0927|31.0887|6976.8042|24.0877|0.4512|
> |vit_base_patch16_224.augreg_in1k|86.6129|61.8750|34.1331|6979.8242|24.0937|0.4512|
> |convmixer_768_32.in1k|57.7016|57.7621|20.5040|6979.7368|24.0953|0.4513|
>
> + **Promising data transferability:** 11 training models + 49,950 validation images + l2 budget ($\tau=25$)
> |model|ASR|ASR-LRFlip|ASR-Blur|l1|l2|linf|
> | :--- | :--- | :---| :--- | :--- | :--- | :--- |
> |resnet18.a1_in1k|42.2162|38.0480|12.2422|6962.4048|24.0585|0.4511|
> |resnet50.a1_in1k|64.3363|63.9640|20.9249|6940.6040|24.0086|0.4510|
> |densenet121.ra_in1k|55.3053|36.8829|16.8068|6949.0151|24.0298|0.4511|
> |densenet169.tv_in1k|42.5566|27.3333|11.9660|6976.6367|24.0961|0.4511|
> |hrnet_w18.ms_in1k|44.5966|14.6466|10.8789|6981.5674|24.0985|0.4510|
> |convnext_tiny.fb_in1k|73.5596|67.7437|22.8729|6951.0532|24.0366|0.4510|
> |convnext_large.fb_in1k|62.6567|59.7177|16.9349|6976.6211|24.0996|0.4510|
> |maxvit_rmlp_small_rw_224.sw_in1k|83.9860|75.2933|32.1201|6940.9741|24.0065|0.4509|
> |coatnet_rmlp_1_rw_224.sw_in1k|75.8258|56.5626|27.5836|6940.6362|24.0088|0.4510|
> |vit_base_patch16_224.augreg_in1k|79.7758|58.6526|32.5325|952.8955|24.0363|0.4509|
> |convmixer_768_32.in1k|50.0881|49.3273|17.4915|6935.5425|23.9982|0.4510|
>
> + **Model transferability is challenging:** 12 testing models + 4,995 training images + l2 budget ($\tau=25$)
> |model|ASR|ASR-LRFlip|ASR-Blur|l1|l2|linf|
> | :--- | :--- | :--- | :---| :--- | :--- | :--- |
> |resnet34.a1_in1k|57.3589|60.0806|18.0847|6984.5049|24.1104|0.4513|
> |resnet50d.a1_in1k|61.0081|55.7056|20.6048|6964.5034|24.0579|0.4513|
> |densenet161.tv_in1k|27.2984|25.9879|7.8831|7012.8032|24.1770|0.4514|
> |densenet201.tv_in1k|23.5282|18.0847|5.8669|7011.9932|24.1789|0.4516|
> |hrnet_w30.ms_in1k|8.4274|11.1290|1.6935|7095.8892|24.4289|0.4511|
> |convnext_base.fb_in1k|54.1331|57.6008|14.9395|7028.1055|24.2299|0.4511|
> |maxvit_base_tf_224.in1k|28.7500|30.4839|7.9839|7122.3784|24.4959|0.4514|
> |mixer_b16_224.goog_in21k_ft_in1k|21.4919|27.1573|5.3024|7088.7642|24.3999|0.4512|
> |swin_base_patch4_window7_224.ms_in1k|46.5323|43.8911|11.8145|7057.6040|24.3108|0.4513|
> |convnextv2_huge.fcmae_ft_in1k|22.7621|23.9919|7.8831|7066.8862|24.3144|0.4514|
> |vit_base_patch16_clip_224.openai_ft_in12k_in1k|14.8387|11.2097|6.4919|6990.9468|24.1232|0.4512|
> |eva_giant_patch14_224.clip_ft_in1k|3.1855|2.1371|1.2298|7051.1938|24.2648|0.4520|
>
> + **Double-transferability is even more challenging:** 12 testing models + 49,950 validation images + l2 budget ($\tau=25$)
> |model|ASR|ASR-LRFlip|ASR-Blur|l1|l2|linf|
> | :--- | :--- | :--- | :---| :--- | :--- | :--- |
> |resnet34.a1_in1k|48.9489|50.9349|15.0771|6947.0352|24.0254|0.4510|
> |resnet50d.a1_in1k|53.8779|49.3373|18.1301|6930.2163|23.9833|0.4512|
> |densenet161.tv_in1k|23.7477|22.3604|6.6627|6979.3813|24.0984|0.4511|
> |densenet201.tv_in1k|18.5205|14.3944|5.1071|7012.6494|24.1822|0.4514|
> |hrnet_w30.ms_in1k|6.8028|8.7427|1.3473|7095.4663|24.4162|0.4512|
> |convnext_base.fb_in1k|56.0821|58.3704|15.6196|6987.2969|24.1314|0.4510|
> |maxvit_base_tf_224.in1k|32.2703|34.1782|9.3814|7081.4541|24.3896|0.4514|
> |mixer_b16_224.goog_in21k_ft_in1k|20.2903|25.2573|5.3934|7055.4492|24.3167|0.4512|
> |swin_base_patch4_window7_224.ms_in1k|49.1872|46.7487|12.7027|7003.9204|24.1761|0.4511|
> |convnextv2_huge.fcmae_ft_in1k|24.7147|24.9209|9.0771|7005.8657|24.1682|0.4510|
> |vit_base_patch16_clip_224.openai_ft_in12k_in1k|12.8989|10.4545|5.5375|6970.8550|24.0845|0.4510|
> |eva_giant_patch14_224.clip_ft_in1k|2.6106|1.7598|1.1992|7020.3071|24.2104|0.4515|
>
> We note that though the double-transferability is not as promising as top-1 UAPs, it is still non-trivial due to the sheer complexity of learning such ordered top-2 UAPs under the $\ell_2$ energy budget.

---

> > ### Author Response · Authors · 2025-11-30
> > **Global responses -- revision promise**
> >
> > For ordered top-3 UAPs, under the $\ell_2$ budget $\tau=25$, we found the learning failed for the 11-model ensemble. For single-model UAPs, there are still promising results, but they often lack model/double-transferability.
> >
> > We will update all the results in revision once all results are finished.
> >
> > We will also provide ablation studies of learning UAPs under $\ell_{inf}$ budget, e.g., $16/255$, which is also a common setting in studying attack transferability.
> >
> > Overall, our revision will make those observation clearer compared to our current submission summarized as follow:
> >
> > In the introduction section, we clearly present the challenge: (line 41) "**achieving effective top-1 targeted UAPs has proven challenging**", (lines 49-51) "**it is unclear whether targeted UAPs can retain their attacking power for ensembles of those disparate DNNs, as well as adversarially-robustified counterparts**", and (lines 71-72) "**Learning ordered top-K targeted UAPs has not been studied in the literature, however.**" With these context, we summarize the observations from Table 2 compactly, (lines 337-339): "These results, especially the double-transferability across testing models and testing images, represent a significant leap forward from the prior art, as well as broader impacts as dicussed in Appendix A.".
> >
> > In the Appendix A, we summarize (lines 704-708) "**The promising transferability of the learned ordered top-K UAPs to unseen models, especially those at the so-called foundation model level (e.g., the CLIP ViT-B in Table 2), might be exploited in a harmful way for applications built on those models. Powerful defense methods should be studied, which we will investigate in our future work. We will also release our source code to encourage more research on studying defense methods against the proposed AllAttacK.**"

---

### Author Response · Authors · 2025-12-03

Dear Area Chairs,

Thank you for taking over the evaluation of our submission during this unusual situation. We sincerely appreciate your time and efforts. We take this opportunity to summarize our submission, review comments and our rebuttal for your reference.

---

### Paper Summary

**Problem & Motivation:** The paper studies Ordered Top-K Targeted Universal Adversarial Perturbations (UAPs)—a setting where a single perturbation forces all images to follow a specified Top-K ranking of class labels. This extends prior work that focused on untargeted or Top-1 targeted UAPs. Ordered Top-K constraints are motivated by emerging ranking-based robustness standards (e.g., NIST SP800-226) and the practical importance of ranking stability.

**Key Idea — AllAttacK**: A MAXSAT-style formulation jointly optimizes UAPs on three axes:
 + Data axis: generalization from training images to unseen images
 + Model axis: transfer across many architectures (full CNN/ViT/Mixer/CLIP/EVA coverage)
 + Target axis: enforcing any arbitrary ordered Top-K label sequence

**Two realizations are used:**
 + AllKLAttacK (KL-divergence surrogate for ordered logits)
 + AllQuadAttacK (QP-based relaxation enforcing linear Top-K ordering constraints)

**Experimental Scale**
 + 27 diverse models (ResNets, DenseNets, ConvNeXt, ViT, MaxViT, MLP-Mixer, CLIP, EVA, adversarially trained models)
 + 500+ UAPs learned
 + Large-scale evaluations of transferability: across data, across models, and jointly (double-transferability).

**Main Findings**
 + Strong Top-1 transferability across unseen data/models;
 + Ordered Top-K attacks are learnable up to K = 6;
 + Semantically coherent target sequences yield higher ASR;
 + New results during rebuttal show significant improvements especially for Top-1, with high ASRs even on strong models like CLIP and EVA.

---

### Reviewer-Identified Strengths

+ **S1. Novelty: First large-scale study of ordered Top-K UAPs**
	- Reviewer `V4wR`: Recognizes AllAttacK as “the first to systematically study ordered Top-K targeted UAPs.”
	- Reviewer `Q89G`: Highlights novelty: “New attack axis: ordered Top-K universals at scale.”

+ **S2. Breadth & Scale of Experiments**
	 - Reviewer `MG42`: Notes extensive model coverage.
	 - Reviewer `V4wR`: Praises experiments across 27 models, including CLIP and EVA.
	 - Reviewer `Q89G`: Emphasizes scale: 500 UAPs, 27 models, broad architectural diversity.

+ **S3. Strong Transferability Across Axes**
	 - Reviewer `V4wR`: Notes strong double-transferability.
	 - Reviewer `Q89G`: Details data/model cross-transfer in tables (Top-1 through Top-6).
	 - Reviewer `DGam`: Highlights model-agnostic and image-agnostic transferability.

+ **S4. Methodological Contribution (KL & QP formulations)**
	 - Reviewer `V4wR`: Appreciates complementary KL/QP formulations and clarity of pipeline.
	 - Reviewer `Q89G`: Cites two complementary surrogates as strength.

+ **S5. Diagnostics & Interpretability**
	 - Reviewer `Q89G`: Notes PCC analysis showing δ dominates logits.
	 - Reviewer `DGam`: Notes interpretability and the emergence of semantic structure in perturbations.

---

### Key New Results Added During Rebuttal

+ Requested by Reviewers `V4wR` & `Q89G` (norms, perceptibility, budgets)
	- Explicit $\ell_2$ projection (budget $\tau=25$) makes perturbations quasi-imperceptible and reproducible.

+ Requested by Reviewers `MG42` & `DGam` (deeper evaluation)
	- Expanded training set: 4995 images.
	- Expanded testing set: the entire ImageNet-1k validation set.

+ Requested by All Reviewers (stronger results, interpretability)
	- Top-1 UAPs now achieve extremely high ASR across training and testing models:
	- CLIP ViT-B: ~80%
	- EVA-Giant: ~42.8%

+ Requested by Reviewer `Q89G` (robustness checks)
	- Defensive tests: LR-Flip (negligible impact), Gaussian blur (moderate drop).

---

> ### Author Response · Authors · 2025-12-03
>
> ### Reviewer Concerns & Questions and Our Responses Overview
>
> + Presentation & Insight Concerns
> 	-	Reviewer `MG42`: Tables overwhelm readers; analysis is too shallow.
> > Our response: Agree to improve clarity, deepen analysis, restructure tables, and add more narrative.
> 	-	Reviewer `DGam`: Data-axis motivation unclear.
> > Our response: Clarified data-axis motivation; expanded training data to 4995 images.
>
> + Theory & Optimization Concerns
> 	-	Reviewer `V4wR`: No solvability/convergence analysis for MAXSAT/QP; local convexity assumptions unaddressed.
> > Our response: MAXSAT inherently unsolvable in deep nonlinear settings; formal convergence is intractable. Empirical evidence replaces theoretical guarantees.
> 	-	Reviewer `DGam`: Missing derivations from Eq.4 → Eq.5/Eq.7; unclear definitions ($z_{i,s}$).
> > Our response: Added derivations and clarified variables $z_{i,s}$ as latent features optimized in QP
>
>
> + Experimental Protocol & Reproducibility
> 	-	Reviewer `Q89G`: Imperceptibility not justified—needs LPIPS/SSIM/PSNR; Best/Mean/Worst protocol may inflate ASR; needs CIs and per-image distributions; Norm bounds unclear; energy–success trade-offs not systematically evaluated.
> > Our response: Added explicit $\ell_2$ ($\tau=25$) and $\ell_{inf}$ (16/255 planned) budgets; Will report LPIPS/SSIM/PSNR and conduct a pilot human study; Will provide ASR distributions and 95% CIs (Wilson + bootstrap); Best/Mean/Worst are fair due to random target lists with no semantic overlap.
> 	-	Reviewer `DGam`: Need baselines (CW$^K$, RisingAttacK). Why $K$ limited to 1–6?
> > Our response: Will include baseline discussion for CW$^K$ and RisingAttack; explain practical limitations. $K$ limited to $\leq 6$ because optimization fails beyond that for multi-model UAPs; consistent with single-model single-image  counterparts.
>
> + Methodological Questions
>    -    Reviewer `DGam`: Impact of mini-batch/mini-model sampling unclear; How AD/QP satisfy constraints not sufficiently explained.
> >Our response: Mini-model batching reduces memory and improves model transferability due to stochastic exposure. AD/QP methods do not guarantee constraint satisfaction—this is intrinsic to universal, multi-model Top-K attacks.
>
> + Ordered vs Unordered Top-K
> 	-	Reviewer `V4wR`: What happens in unordered Top-K?
> 	-	Reviewer `Q89G`: Need ablations for unordered and partially ordered settings.
> > Our responses to both reviewers: Unordered Top-K increases constraints to $K × (C–K) × M$; KL is less suitable; QP can handle it but is slower; Ordered Top-K reveals deeper vulnerabilities and aligns with ranking robustness; unordered Top-K is future work.
>
>
> + Scope Concerns
> 	-	Reviewer `DGam` & `Q89G`: No black-box analysis.
> >  Our response: Model-transfer experiments are black-box surrogate attacks. Direct black-box Top-K UAP training is infeasible (high query cost).
> 	-	Reviewer `V4wR`: No defenses, no robustness training exploration.
> > Our response: Evaluated LR-Flip and Gaussian blur; blur degrades ASR, flip does not. More robust defenses will be explored in future work.
> 	-	Reviewer `Q89G`: Foundation models (CLIP/EVA) show weaker transfer—needs explanation.
> > Our response: Foundation model weakness explained through different pretraining objectives; new experiments dramatically improved CLIP/EVA transferability.

---

### Meta-Review · Area_Chair_NBQq · 2025-12-07

**Summary:**

This submission proposes AllAttacK, a framework for learning ordered Top-K targeted universal adversarial perturbations (UAPs) across three axes: images, models, and label targets. The work is positioned as the first systematic large-scale study of ordered Top-K UAPs, with extensive experiments across 27 models, including CNNs, ViTs, Mixers, CLIP, and EVA.

Reviewers identified several strengths, including the novelty of ordered Top-K UAPs, breadth of experimentation, strong transferability results, and clear two-formulation design (KL and QP). However, they also raised significant concerns regarding theory, clarity, reproducibility, perceptibility constraints, evaluation protocol, missing baselines, incomplete analysis for QP/AD formulations, unclear motivation in several sections, missing ablations, and lack of black-box experiments.

During the rebuttal, the authors provided substantial new experiments, including:

Expanded training data (4995 images) and complete val-set evaluation.

Updated model ensemble (23 timm models).

New explicit ℓ₂ budget projection ensuring quasi-imperceptibility.

Much stronger Top-1 transferability results (including CLIP and EVA).

Defensive tests (flip and Gaussian blur).

Clarifications for derivations, motivations, and constraints.

Promises for expanded tables, perceptual metrics, and CIs in revision.

Despite these important additions, several core issues remain unresolved, especially regarding methodological soundness, missing baselines, theoretical justification, feasibility of QP, and completeness of evaluation.

Based on the balance of strengths and weaknesses, and considering the degree to which the rebuttal addresses reviewer concerns, the paper does not yet meet the bar for acceptance to ICLR.

**Reviewer Concerns:**

Concerns adequately addressed in the rebuttal

The authors provided substantial clarification and new experimental evidence addressing:

Perceptibility & norm constraints (Q89G, V4wR)
→ Introduction of explicit ℓ₂ projection, plan to add LPIPS/SSIM/PSNR, human study pilot.

Data-axis motivation (MG42, DGam)
→ Clearer explanation + expansion to 4995 training images and 49,950 test images.

Target-axis confusion and sampling (DGam)
→ Authors now explicitly exclude images with the top-1 target label.

QP/KL derivation clarity (V4wR, DGam)
→ Detailed explanation of why no theoretical solvability exists and how constraints are relaxed.

New strong results for Top-1 UAPs (all reviewers)
→ Major performance improvements now demonstrated on CLIP and EVA.

Bundled clarifications on margins, mini-model batching, transferability, and optimization
→ Authors addressed reviewer questions, though mostly verbally rather than analytically.

Concerns that remain outstanding

These key issues were not sufficiently resolved:

Lack of theoretical justification or analysis (V4wR, DGam)
The rebuttal acknowledges no solvability/convergence theory is possible, but does not provide even empirical convergence diagnostics, feasibility analysis, or KKT residuals for QP as requested.

Baseline comparisons missing (DGam, Q89G)
RisingAttacK and CW
𝐾
K
 baselines are not included; explanation is insufficient for omitting them in a universal attack context.

Evaluation protocol concerns (Q89G)

Best/Mean/Worst might inflate ASR → authors promise per-image distributions and CIs but do not provide them.

Energy–success frontiers still missing.

QP feasibility / optimality / stability (Q89G, V4wR)
No actual empirical evidence (feasibility rate, failure mode statistics) is shown.

Ordered vs unordered Top-K ablations (V4wR, Q89G)
Authors defer this entirely to future work despite direct reviewer request.

Effect of stochastic mini-model batching (DGam)
Explanation is qualitative; no sensitivity analyses were provided.

Black-box scenario analysis (DGam, Q89G)
Authors claim infeasibility, but do not provide empirical study or even a conceptual analysis beyond surrogate transfer.

Presentation weaknesses (MG42)
The rebuttal promises restructuring but does not demonstrate improved analysis or readability.

Overall, substantial concerns from MG42 and DGam remain unresolved, while Q89G and V4wR still see important methodological and evaluation gaps.

**Reviewer Scores:**

Below is my assessment of how reviewers would or would not change their scores if they participated in post-rebuttal discussion.
(Per instruction: no explicit numerical rating values are included.)

Reviewer MG42

Positive reaction to expanded experiments and clearer motivation.

But core concerns on lack of deep analysis and presentation remain largely unaddressed.

Likely score change: No improvement.

Reviewer V4wR

Rebuttal provides many clarifications and shows much stronger empirical results.

Still has unresolved concerns around theoretical aspects and unordered Top-K.

Likely score change: Slight positive shift, but would still not strongly support acceptance.

Reviewer Q89G

Appreciates the extensive new results, perceptibility controls, and many addressed issues.

However, several major evaluation concerns remain (CIs, unordered Top-K, QP feasibility, baselines).

Likely score change: Small positive shift, but still marginal.

Reviewer DGam

Major concerns around missing baselines, unclear derivations, insufficient explanation, and missing black-box evaluation remain largely unresolved.

Likely score change: No improvement.

Overall, the reviewer consensus would remain negative to borderline, insufficient for acceptance.

---

### Decision · Program_Chairs · 2026-01-26

Reject